# Network two-sample test for block models

**Chung K. Nguen**    **Oscar H. Madrid-Padilla**    **Arash A. Amini**

Department of Statistics
University of California, Los Angeles
California, CA 90024
{cnguen1, oscar.madrid, aaamini}@stat.ucla.edu

## Abstract

We consider the two-sample testing problem for networks, where the goal is to determine whether two sets of networks originated from the same stochastic model. Assuming no vertex correspondence and allowing for different numbers of nodes, we address a fundamental network testing problem that goes beyond simple adjacency matrix comparisons. We adopt the stochastic block model (SBM) for network distributions, due to their interpretability and the potential to approximate more general models. The lack of meaningful node labels and vertex correspondence translate to a graph matching challenge when developing a test for SBMs. We introduce an efficient algorithm to match estimated network parameters, allowing us to properly combine and contrast information within and across samples, leading to a powerful test. We show that the matching algorithm, and the overall test are consistent, under mild conditions on the sparsity of the networks and the sample sizes, and derive a chi-squared asymptotic null distribution for the test. Through a mixture of theoretical insights and empirical validations, including experiments with both synthetic and real-world data, this study advances robust statistical inference for complex network data.

## 1 Introduction

Network data is pervasive across various fields, including transportation [7], trading [5], social networks [11, 41], neuroscience [32, 31, 6, 30], ecology [35], and politics [4], among others. To address the diverse needs of these applications, recent statistical methods have emerged to handle scenarios where multiple networks are observed [22, 28, 19, 3]. In this paper, we consider the problem of testing whether two sets of networks have been generated from the same probability model. Specifically, in network two-sample testing, we are given two collections of graphs $\{G_{11}, \ldots, G_{1N_1}\}$ and $\{G_{21}, \ldots, G_{2N_2}\}$ generated as $G_{rt} \sim F_r$ for $r \in \{1, 2\}$, $t \in [N_r] := \{1, \ldots, N_r\}$, and would like to test

$$H_0 : F_1 = F_2 \quad \text{against} \quad H_1 : F_1 \neq F_2. \tag{1}$$

We approach the two-sample testing problem in (1) without requiring the existence of vertex correspondence. This means that the nodes in different networks are unlabeled, and as a consequence, the $i$th node in the $t$th network of the $r$th sample has no direct correspondence or meaning in any of the other networks. Additionally, it is possible for the number of nodes, denoted as $n_{rt}$, to vary across different networks. As such, the problem we are studying is truly a network testing problem, and not immediately reducible to testing the distributions of a collection of adjacency matrices.

To further elaborate on the subtlety of graph versus matrix testing, consider the adjacency matrices $A_{rt} \in \{0, 1\}^{n_{rt} \times n_{rt}}$ associated with each graph $G_{rt}$ for $r \in \{1, 2\}$ and $t \in [N_r]$, by fixing a particular node labeling for each graph. Then, graph-level distribution $F_1$ induces distributions on the adjacency matrices $A_{1t}, t \in [N_1]$. However, these distributions are not directly comparable since the

dimensions of $\{A_{1t}\}$ could be different. Even if the dimensions are the same (i.e., graphs $\{G_{1t}\}$ all have the same number of nodes), the distributions of $\{A_{1t}\}$ depend on the particular labelings chosen for each graph. Changing the labelings, will change the distributions of $\{A_{1t}\}$, although we want to treat all such distributions as the same. In other words, when testing the equality of distributions of the adjacency matrices, the formulation must account for equality up to relabeling of the nodes, and disregard the potential size mismatches, to truly test at the level of graph distributions.

The absence of vertex correspondence introduces a fundamental challenge that elevates this problem beyond simple matrix comparisons. In fact, the unlabeled two-sample testing problem subsumes the classic *graph isomorphism problem* as a special case. If we consider samples of size $N_1 = N_2 = 1$ where the generating distributions are point masses on two fixed graphs, $G_1$ and $G_2$, the two-sample test reduces to deciding if $G_1$ is isomorphic to $G_2$. Our problem can thus be seen as a "noisy" or statistical version of graph isomorphism, where we must determine if two distributions on graphs are equivalent up to a permutation.

To model the distribution of the adjacency matrices, we adopt the stochastic block model (SBM) framework [17] as a foundational structure. The SBM is one of the most commonly used models in the literature, valued not only for its simplicity and interpretability but also for its role as a universal approximator. It is effectively the network equivalent of a histogram [29]; an SBM with a sufficiently large number of communities, $K$, can form a step-function approximation to an arbitrary graphon in the $L^2$ sense [2, 14].

This capability positions our method as a "bin-then-test" strategy, which is analogous to classic non-parametric approaches like Pearson's chi-squared test, where continuous data is binned to form contingency tables. This perspective makes the SBM a natural and powerful starting point, providing a flexible, non-parametric framework for the two-sample testing of networks.

In this work, we assume that the adjacency matrices are generated from SBM distributions. In particular, an adjacency matrix $A_{rt} \in \{0, 1\}^{n_{rt} \times n_{rt}}$ is generated as follows. First, we draw a vector of random *community* labels $z_{rt} = (z_{rt1}, \ldots, z_{rtn})^\top \in [K]^n$ with entries that are independent draws from a categorical distribution with parameter $\pi_r = (\pi_{r1}, \ldots, \pi_{rK})$, where $\pi_{rl} > 0$ for all $l$ and $\sum_{l=1}^K \pi_{rl} = 1$. Here, $K$ is the number of communities. Then, given the labels $z_{rt}$, the entries of $A_{rt}$, below the diagonal, are independently drawn as

$$(A_{rt})_{ij} \,|\, z_{rt} \sim \mathsf{Ber}((B_r)_{z_{rti} z_{rtj}}), \quad i < j,$$

where $B_r \in [0, 1]^{K \times K}$ is a symmetric matrix of probabilities. We assume the networks to be undirected, with no self-loops, hence $A_{rt}$ is extended to a symmetric matrix with zero diagonal entries. Throughout, we write $A_{rt} \sim \mathsf{SBM}_{n_{rt}}(B_r, \pi_r)$ for $A_{rt}$ generated as described above. We often suppress the dependence on $n_{rt}$ and simply write $A_{rt} \sim \mathsf{SBM}(B_r, \pi_r)$.

Given $A_{rt} \sim \mathsf{SBM}(B_r, \pi_r)$, the network testing problem (1) is equivalent to the following

$$\begin{cases} H_0 : \ \exists P \in \Pi_K, \quad B_1 = PB_2P^\top \quad \text{and} \quad \pi_1 = P\pi_2 \\ H_1 : \ \forall P \in \Pi_K, \quad B_1 \neq PB_2P^\top \quad \text{or} \quad \pi_1 \neq P\pi_2 \end{cases} \tag{2}$$

where $\Pi_K$ is the set of $K \times K$ permutation matrices. The inclusion of permutation matrices $P$ in (2) is related to the subtlety of testing unlabeled network models. The fact that there is no pre-defined node label or vertex correspondence across networks leads to the meaninglessness of community labels in SBM. Consequently, demanding that $B_1 = B_2$ under the null hypothesis is not meaningful. Put another way, SBM parameters $(B_r, \pi_r)$ are only identifiable up a permutation; see (4).

Before proceeding, we generalize the problem slightly. In (2), the null hypothesis is rejected if the class proportions $\pi_r$ differ modulo permutation. Given the SBM labels, this scenario is straightforward to test by estimating and sorting class proportions to remove permutation ambiguity. To avoid this relatively trivial case, we treat the class proportions as nuisance parameters. This choice is also practical, as communities are typically defined by connectivity patterns rather than relative sizes. Additionally, differences in class proportions $\pi_r$ can sometimes be transferred to connectivity matrices $B_r$ via community *refinement* (see Appendix C.1 in the supplementary material). With these considerations in mind, we focus on the following formulation for the remainder of the paper: Given $A_{rt} \sim \mathsf{SBM}(B_r, \pi_{rt})$, we test

$$\begin{cases} H_0 : \ \exists P \in \Pi_K, \quad B_1 = PB_2P^\top \\ H_1 : \ \forall P \in \Pi_K, \quad B_1 \neq PB_2P^\top. \end{cases} \tag{3}$$

Note that here, the class proportions $\pi_{rt}$ are allowed to vary across networks without violating the null hypothesis.

To develop a test for (3), it is essential to address the significant challenge posed by the fact that matrices $B_r$ are only identifiable up to permutations. A potential approach to constructing a test statistic is to estimate the community labels for each network, using any of the many existing approaches (for instance [4]). Subsequently, one can proceed to estimate $B_r$ by computing block averages of the adjacency matrices, based on these labels. However, the challenge lies in how to effectively combine, and compare, the estimates from different networks when vertex correspondence is not assumed. This becomes particularly daunting, as matching estimated labels across different networks boils down to searching over the space of permutations $\Pi_K$, which has $K!$ elements, a non-trivial task even for moderate values of $K$. It is worth noting that this matching challenge exists even if we observe a single network for each of the two samples, since one has to properly match the communities of the two networks to be able to compare the connectivity matrices $B_1$ and $B_2$.

## 1.1 Summary of results

In this paper, we tackle the core challenges of unlabeled network comparison by developing a practical and theoretically-grounded two-sample test. Our main contributions are:

- **An Efficient and Consistent Matching Algorithm.** We introduce a computationally efficient spectral matching algorithm to solve the fundamental problem of aligning estimated SBM parameters. We prove that this algorithm consistently recovers the correct alignment under standard spectral conditions on the underlying connectivity matrices (formally, the $(\eta, \theta)$-friendly property).

- **A Rigorous Characterization of the Test Statistic.** We establish the key theoretical properties of our test. Under the null hypothesis, we prove that our statistic converges to a $\chi^2_{K(K+1)/2}$ distribution under mild conditions on network sparsity (average degree $\nu_n = \Omega(\log n)$) and sample size. This provides a practical foundation for accurate p-value calculation.

- **Proof of Consistency and Power.** Under the alternative hypothesis, we show that the test is consistent and powerful. The test statistic grows at a rate of $\Omega(N\nu_n^2)$, demonstrating that its power increases with both the number of networks ($N$) and their average degree ($\nu_n$).

- **A Scalable and Empirically Validated Test.** We provide a practical, computationally efficient algorithm for the two-sample test, whose scalability stands in contrast to computationally prohibitive methods like subgraph counting. We demonstrate its superior performance and robustness on a wide range of synthetic data, real-world networks, and non-SBM models, including graphons and RDPGs.

## 1.2 Related work

Two-sample testing for labeled networks with known vertex correspondence has been well studied in the literature. Notable efforts in the labeled case include the following: [16], where the authors develop a test based on the geometric characterization of the space of the graph Laplacians. [15] consider the problem from a minimax testing perspective, focusing on the theoretical characterization of the minimax separation with respect to the number of networks and the number of nodes. [25] develop a test based on an omnibus embedding of multiple networks into a single space. [8, 9] develop a spectral-based test statistic that has an asymptotic Tracy–Widom law under null. [18] introduce a novel approach to testing degree corrected stochastic block models based on interlacing balance measure.

Existing literature on testing (1) with unlabeled networks with no vertex correspondence is limited, but there have been notable efforts. For instance, [22] develop a geometric and statistical framework for making inferences about populations of unlabeled networks. [37] introduce a two-sample test for random dot product graphs based on estimating the maximum mean discrepancy between the latent positions. [27] develop a two-sample test, based on subgraph counts, and characterize the null distribution under a graphon model. While not designed for testing, two algorithms described in [28] solve a closely related problem of clustering network-value data.

As discussed earlier, the construction of our test relies on the subroutine that matches the estimated connectivity matrices of two networks. This type of problem is known as weighted graph matching and can be formulated as a quadratic assignment problem (QAP) which is NP-hard. One of the

standard techniques is to relax the constraint set of permutation matrices to its convex hull [26, 39, 1] or the set of orthogonal matrices [13], and then "round" the solution to the set of permutation matrices. [38] proposes to solve the weighted matching by solving a linear assignment problem on a matrix derived from eigenvectors of the adjacency matrices of the two graphs. [12] show exact recovery of the permutation with high probability for the Gaussian Wigner model. The idea of our main matching algorithm is mentioned in passing in [36, Section 39.4] in the context of testing isomorphism of two graphs, and is attributed to [24].

## 2   Matching Methodology

In this section, we begin by describing the challenges associated with the matching problem, specifically the task of aligning two matrices, $B_1$ and $B_2$, by relabeling the communities. Subsequently, we introduce our proposed matching algorithm, which is at the heart of our test construction.

### 2.1   Matching challenge

From a statistical perspective, the challenge in testing samples from $\mathsf{SBM}(B, \pi)$ is that the parameters $(B, \pi)$ are identifiable only up to permutations. That is, for any permutation matrix

$$\mathsf{SBM}(B, \pi) = \mathsf{SBM}(PBP^\top, P\pi) \tag{4}$$

as distributions. To illustrate the challenge more clearly, consider observing the two-sample problem with only two adjacency matrices

$$A_1 \sim \mathsf{SBM}(B_1, \pi_1), \quad A_2 \sim \mathsf{SBM}(B_2, \pi_2).$$

Assume that the null hypothesis holds, where we can assume $B_2 = B_1$. To devise a test, a natural first step is to fit an $\mathsf{SBM}$ to $A_r$, using any consistent community detection algorithm, to obtain the labels, from which we can construct an estimate $\widehat{B}_r = \mathcal{B}(A_r)$ of $B_r$, for $r = 1, 2$. The operator $\mathcal{B}$ is a shorthand for the procedure that produces such estimate, the details of which are discussed in Section 3. Even though $B_1 = B_2$, in general, we will have

$$\widehat{B}_2 \approx P^* \widehat{B}_1 P^{*\top},$$

for some permuation matrix $P^* \in \Pi_K$, since in each case the communities will be estimated in an unknown (arbitrary) order. There is no way a priori to guarantee the same order of estimated communities in the two cases. This is a manifestation of the permutation ambiguity in (4). To simplify future discussions, let us introduce the following notation for *exact matching*:

$$B_1 \xrightarrow{P^*} B_2 \quad \Longleftrightarrow \quad B_2 = P^* B_1 P^{*T}. \tag{5}$$

The above discussion shows that two-sample testing for SBMs requires solving the following subproblem:

**Problem 1.** Assume that $B_1 \xrightarrow{P^*} B_2$ for $B_1, B_2 \in [0, 1]^{K \times K}$. The *noisy graph isomorphism* (or graph matching) problem is to recover a matching permutation $P^* \in \Pi_K$ between $B_1$ and $B_2$, using only noisy observations $\widehat{B}_r \approx B_r$ for $r = 1, 2$.

As alluded to in Section 1.2, this problem is in general hard. In particular, finding an optimal matching in Frobenius norm reduces to solving a quadratic assignment problem (QAP), which is NP-hard in general.

### 2.2   Spectral matching

For a large class of weighted networks, we can avoid solving a QAP to recover a matching. The core idea is that if a connectivity matrix has a unique spectral signature, its eigenvalue decomposition (EVD) contains the information needed to recover any permutation. The following lemma makes this precise.

**Lemma 2.1.** Consider two $K \times K$ symmetric matrices $B_1$ and $B_2$ with EVDs given by $B_r = Q_r \Lambda_r Q_r^\top$ for $r = 1, 2$. If the eigenvalues of $B_1$ are distinct, then a permutation matrix $P^*$ satisfies $B_2 = P^* B_1 P^{*\top}$ if and only if there exists a diagonal sign matrix $S^*$ such that:

$$\Lambda_2 = \Lambda_1, \tag{6}$$
$$Q_2 = P^* Q_1 S^*. \tag{7}$$

---

**Algorithm 1** Spectral Matching: $\mathcal{M}(\widehat{B}_1 \to \widehat{B}_2)$

---

**Input:** Estimated connectivity matrices $\widehat{B}_1$, $\widehat{B}_2$
**Output:** Estimated sign matrix $\widehat{S}$, estimated permutation matrix $\widetilde{P}$
 1: **for** $r = 1, 2$ **do**
 2:     Perform EVD on $\widehat{B}_r$ to obtain $\widehat{B}_r = \widehat{Q}_r \Lambda_r \widehat{Q}_r^\top$.
 3: **end for**
 4: Recover the diagonal signs $\widehat{S}_{ii} = \mathrm{sign}\big([\widehat{Q}_2^\top \mathbf{1}]_i / [\widehat{Q}_1^\top \mathbf{1}]_i\big)$ and set $\widehat{S} = \mathrm{diag}(\widehat{S}_{ii})$.
 5: Recover the permutation matrix as $\widetilde{P} = \mathrm{LAP}(\widehat{Q}_1 \widehat{S} \widehat{Q}_2^\top)$.
 6: **return** $\widehat{S}$, $\widetilde{P}$.

---

This result provides a direct algebraic path to finding $P^*$. To ensure this process is robust against the noise present in estimates $\widehat{B}_r$, we require a slightly stronger condition that the eigenvalues are not just distinct, but well-separated. This, along with a technical condition ensuring that no eigenvector is orthogonal to the all-ones vector, guarantees that the matching permutation $P^*$ is **unique** and that our spectral approach is stable. We formalize these combined requirements as the "$(\eta, \theta)$-friendly" property in Appendix A.1.

Our proposed spectral matching algorithm, based on Lemma 2.1, is outlined in Algorithm 1. It relies on solving the *linear assignment problem* (LAP), defined for a $K \times K$ matrix $Q$ as:

$$\mathrm{LAP}(Q) := \underset{P \in \Pi_K}{\mathrm{argmax}} \ \mathrm{tr}(PQ). \tag{8}$$

The LAP can be solved efficiently, for example, by the Hungarian algorithm [23].

The intuition behind Algorithm 1 follows directly from Lemma 2.1. From (7), we can left-multiply by $\mathbf{1}^\top$ to get $\mathbf{1}^\top Q_2 = \mathbf{1}^\top P^* Q_1 S^*$. Since $\mathbf{1}^\top P^* = \mathbf{1}^\top$, this simplifies to $Q_2^\top \mathbf{1} = S^* Q_1^\top \mathbf{1}$. This gives us a way to recover the unknown signs: $S_{ii}^* = [Q_2^\top \mathbf{1}]_i / [Q_1^\top \mathbf{1}]_i$. The aforementioned condition that eigenvectors are not orthogonal to the all-ones vector ensures the denominator $[Q_1^\top \mathbf{1}]_i$ is non-zero, making this step well-defined. For robustness in the noisy case, we only take the sign of this ratio, which is exactly Step 3 of the algorithm.

Once the sign matrix $\widehat{S} \approx S^*$ is known, we can rearrange (7) to isolate the permutation matrix: $P^* = Q_2 S^* Q_1^\top$. In the noisy setting with estimates $\widehat{Q}_r$, the product $\widehat{Q}_2 \widehat{S} \widehat{Q}_1^\top$ will be a noisy version of $P^*$. We find the *closest* valid permutation matrix by projecting it onto the set $\Pi_K$, which can be done by solving the LAP:

$$\underset{P \in \Pi_K}{\mathrm{argmin}} \ \|\widehat{Q}_2 - P\widehat{Q}_1 \widehat{S}\|_F^2 = \underset{P \in \Pi_K}{\mathrm{argmax}} \ \mathrm{tr}(\widehat{Q}_2^\top P \widehat{Q}_1 \widehat{S}) = \mathrm{LAP}(\widehat{Q}_1 \widehat{S} \widehat{Q}_2^\top).$$

This is precisely Step 4 of the algorithm. In the sequel, we write $\mathcal{M}(\widehat{B}_1 \to \widehat{B}_2)$ to denote the permutation matrix $\widetilde{P}$ returned.

**Remark 2.1** (Computational Complexity). A key advantage of our spectral matching approach is its computational efficiency. The complexity of Algorithm 1 is dominated by the EVD and the LAP solution. The LAP on a $K \times K$ matrix can be solved in $O(K^3)$ time using the Hungarian algorithm. This cost depends only on the number of communities $K$, not the number of nodes $n$. In practice, the matching step is extremely fast, making our overall test scalable to very large networks.

## 3 Test construction

We are now ready to describe our main algorithm for constructing an SBM two-sample test. The first step is to estimate the connectivity matrix $B$ for each observed network. Given an adjacency matrix $A$ and a vector of estimated community labels $\widehat{z} \in [K]^n$ (obtained from an algorithm like spectral clustering [4] or Bayesian community detection [35]), we define the block-sum operator $\mathcal{S}$ and block-count operator $\mathcal{C}$:

$$[\mathcal{S}(A, z)]_{k\ell} = \sum_{i,j} A_{ij} \mathbf{1}\{z_i = k, z_j = \ell\}, \tag{9}$$

$$[\mathcal{C}(z)]_{k\ell} = n_k(z)\big(n_\ell(z) - \mathbf{1}\{k = \ell\}\big), \tag{10}$$

---

**Algorithm 2** SBM Two-Sample Test (SBM-TS)

---

**Input:** Adjacency matrices $A_{rt}$ and initial label estimates $\widehat{z}_{rt}^{(0)}$, for $t \in [N_r]$ and $r = 1, 2$.
**Output:** Two-sample test statistic $\widehat{T}_n$.

  *Match to the first network within each sample $r$:*
1: **for** $t = 1, 2, \ldots, N_r$ and $r = 1, 2$ **do**
2:     Set $\widehat{z}_{rt} = \sigma_{rt} \circ \widehat{z}_{rt}^{(0)}$ for independent random permutation $\sigma_{rt}$.
3:     Set $\widehat{S}_{rt} = \mathcal{S}(A_{rt}, \widehat{z}_{rt})$ and $\widehat{m}_{rt} = \mathcal{C}(\widehat{z}_{rt})$.
4:     Set $\widehat{B}_{rt} = \widehat{S}_{rt} \oslash \widehat{m}_{rt}$.
5:     Set $\widehat{P}_{rt} = \mathcal{M}(\widehat{B}_{rt} \to \widehat{B}_{r1})$.
6: **end for**
  *Find the global matching permutation:*
7: Set $\widehat{B}_r = \frac{1}{N_r} \sum_{t=1}^{N_r} \widehat{P}_{rt} \widehat{B}_{rt} \widehat{P}_{rt}^\top$ for $r = 1, 2$.
8: Set $\widehat{P} = \mathcal{M}(\widehat{B}_2 \to \widehat{B}_1)$.
  *Align the sums and counts, across all samples, and form aggregate estimates:*
9: Set $\widehat{S}_r = \sum_{t=1}^{N_r} \widehat{P}_{rt} \widehat{S}_{rt} \widehat{P}_{rt}^\top$ and $\widehat{m}_r = \sum_{t=1}^{N_r} \widehat{P}_{rt} \widehat{m}_{rt} \widehat{P}_{rt}^\top$ for $r = 1, 2$.
10: Set $\widehat{S}_2' = \widehat{P}\widehat{S}_2\widehat{P}^\top$ and $\widehat{m}_2' = \widehat{P}\widehat{m}_2\widehat{P}^\top$.
11: Let $\widehat{B} = (\widehat{B}_{k\ell})$ where $\widehat{B}_{k\ell} = \dfrac{\widehat{S}_{1k\ell} + \widehat{S}_{2k\ell}'}{\widehat{m}_{1k\ell} + \widehat{m}_{2k\ell}'}$ and set $\widehat{\sigma}_{k\ell}^2 = \widehat{B}_{k\ell}(1 - \widehat{B}_{k\ell})$.
12: Let $\widehat{B}^{(1)} = \widehat{S}_1 \oslash \widehat{m}_1$ and $\widehat{B}^{(2)} := \widehat{S}_2' \oslash \widehat{m}_2'$.          (*Stronger estimates*)
13: Let $\widehat{h}_{k\ell}$ be the harmonic mean of $\widehat{m}_{1k\ell}$ and $\widehat{m}_{2k\ell}'$ and form the test statistic:

$$\widehat{T}_n := \sum_{k \leq \ell} \frac{\widehat{h}_{k\ell}}{2\widehat{\sigma}_{k\ell}^2} \big(\widehat{B}_{k\ell}^{(1)} - \widehat{B}_{k\ell}^{(2)}\big)^2. \tag{11}$$

---

where $n_k(z) = \sum_{i=1}^n 1\{z_i = k\}$. Both operators produce a $K \times K$ matrix. The estimate of the connectivity matrix is then $\mathcal{B}(A, z) = \mathcal{S}(A, z) \oslash \mathcal{C}(z)$, where $\oslash$ is elementwise division. For diagonal blocks ($k = \ell$) a double-counting occurs in both the numerator and denominator which can be avoided, for computational efficiency, by restricting the sums to $i < j$.

## 3.1 Main algorithm

Our testing procedure, detailed in Algorithm 2, unfolds in three main stages. First, to enable aggregation, we align all networks *within* each sample to a common reference (e.g., the first network). Second, with the intra-sample alignments complete, we compute an average connectivity matrix for each sample and perform a single *global* match between the two samples. Finally, with all networks aligned to a common orientation, we aggregate the block-level edge counts across all networks to form powerful, low-variance estimates of the connectivity matrices and construct the final chi-squared test statistic.

To understand the form of the final statistic, consider an idealized scenario where all matchings are perfect and the community labels are known. Under the null hypothesis ($B_1 = B_2 = B$), the aggregate edge counts $\widehat{S}_{rk\ell}$ would follow Binomial distributions, and by the CLT, the estimates $\widehat{B}_{k\ell}^{(1)}$ and $\widehat{B}_{k\ell}^{(2)}$ are approximately independent normal variables: $\widehat{B}_{k\ell}^{(r)} \overset{d}{\approx} N(B_{k\ell}, \sigma_{k\ell}^2/m_{rk\ell})$ for $r = 1, 2$, where $\sigma_{k\ell}^2 = B_{k\ell}(1 - B_{k\ell})$ is the common Bernoulli variance and $m_{rk\ell}$ are the total block pair counts for each sample. The standardized difference of these two estimates is

$$\sqrt{h_{k\ell}/(2\sigma_{k\ell}^2)}(\widehat{B}_{k\ell}^{(1)} - \widehat{B}_{k\ell}^{(2)}) \overset{d}{\approx} N(0, 1)$$

where $h_{k\ell}$ is the harmonic mean of $m_{1k\ell}$ and $m_{2k\ell}$. Squaring this quantity, we see that each term in our test statistic (11) is an estimate of this squared standard normal variable. This provides the core intuition why $\widehat{T}_n$ will converge to $\chi_{K(K+1)/2}^2$ distribution, a result we formalize in Section 4.

**Remark 3.1** (Algorithmic Refinements). Two details in Algorithm 2 warrant mention. First, the final estimates $\widehat{B}^{(r)}$ (Step 12) are constructed by pooling edge counts *before* dividing. This produces a lower-variance estimate than averaging the individual $\widehat{B}_{rt}$'s. Second, the randomization of labels in Step 2 is a technical device to ensure the estimated matchings are statistically independent of the data, which simplifies the theoretical analysis (see Lemma E.3 in the appendix).

# 4 Theory

In this section, we present the main theoretical guarantees for our proposed test. For clarity and accessibility, we state our key results—the asymptotic null distribution and the consistency of the test—under a simplified set of assumptions. These assumptions presume the use of a community detection algorithm with a near-optimal misclassification rate, which is achievable in the sparse regimes we consider (e.g., [42]). This allows us to express the conditions directly in terms of fundamental network parameters like the average expected degree $\nu_n$. We provide the more general, finite-sample versions of these theorems, which hold for any community detection algorithm via its misclassification rate, in Appendix A.

## 4.1 Matching Consistency

Our test relies on the ability of the spectral matching algorithm to correctly align the estimated SBM parameters. The following result provides an informal guarantee, with the formal statement deferred to Theorem A.1 in the appendix.

**Theorem 4.1** (Matching Consistency, Informal). If the underlying connectivity matrices $B_r$ have distinct eigenvalues and their eigenvectors are not orthogonal to the all-ones vector (i.e., they are "friendly" per Definition A.1 in the appendix), our spectral matching algorithm (Algorithm 1) consistently recovers the correct permutation between them from noisy estimates $\widehat{B}_r$, provided the estimation error is sufficiently small.

## 4.2 Asymptotic Guarantees for the Test

We now present the main results for our two-sample test statistic, $\widehat{T}_n$. We begin with its limiting distribution under the null hypothesis.

**Theorem 4.2** (Null Distribution). Assume that Algorithm 2 is applied to two SBM samples with the same connectivity matrix $B = (\nu_n/n)B^0$ for a fixed $K \times K$ matrix $B^0$. Assume that:

(i) $B^0$ has distinct eigenvalues, and no eigenvector of it sums to zero.

(ii) The number of networks in each sample grows at most polynomially in the number of nodes $n$, i.e., $N := \max\{N_1, N_2\} = O(n^a)$ for some $a \geq 0$.

(iii) The average expected degree grows at least logarithmically with the network size, i.e., $\nu_n = \Omega(\log n)$.

Then, the test statistic $\widehat{T}_n$ converges in distribution to a chi-squared distribution:

$$\widehat{T}_n \Rightarrow \chi^2_{K(K+1)/2}.$$

Next, we show that the test is consistent under the alternative hypothesis, meaning its power approaches one as the network size grows.

**Theorem 4.3** (Consistency). Assume that Algorithm 2 is applied to two SBM samples with different connectivity matrices $B_r = (\nu_n/n)B_r^0$ for $r = 1, 2$, such that $B_1^0$ is not a permutation of $B_2^0$. Under the same conditions on the eigenvalues/eigenvectors of $B_r^0$ and the growth of $N_r$ as in Theorem 4.2, and assuming the average expected degree $\nu_n \to \infty$, the test is consistent. Specifically, with probability approaching 1,

$$\widehat{T}_n = \Omega(N\nu_n^2),$$

where $N = \max\{N_1, N_2\}$. This ensures that for any fixed significance level, the power of the test approaches 1 as $n \to \infty$.

**Remark 4.4** (On the Sparsity Condition). One might intuitively expect that aggregating $N$ networks could weaken the sparsity requirement of $\nu_n = \Omega(\log n)$. However, this fundamental dependence on $\log n$ persists even for large $N$. To see why, consider a sample of $N$ networks of size $n$. This can be viewed as a single, large, block-diagonal adjacency matrix of size $nN \times nN$, where the off-diagonal blocks corresponding to inter-network connections are unobserved. For a single, fully observed network of size $nN$, established results for tasks like community detection require the average degree to scale as $\Omega(\log(nN)) = \Omega(\log n + \log N)$. Since our problem has less information (due to the missing off-diagonal blocks), the requirement on the $\log n$ term cannot be removed by increasing $N$.

**Remark 4.5** (Practicality of the Sparsity Condition). It is important to note that the condition $\nu_n = \Omega(\log n)$ still describes a very sparse graph regime. For instance, if the average degree grows as $\log n$, adding one million nodes to a network would only increase the average degree by approximately $\log(10^6) \approx 14$. As we demonstrate empirically in Appendix C.2, the average degree in many real-world networks grows even faster than logarithmically with network size, suggesting this assumption is mild in practice.

# 5 Experimental results

In this section, we provide experimental results on real and simulated networks and compare our proposed test to two existing approaches. The code for reproducing all experiments in this section is publicly available at `https://github.com/aaamini/sbm-ts`.

## 5.1 On Model Misspecification and the Choice of K

A key motivation for our SBM-based test is its applicability beyond the strict SBM setting. Because SBMs can approximate general smooth graphons with arbitrary precision by increasing the number of communities $K$, our test provides a robust method for non-parametric network comparison. Our experiments on data from graphon and RDPG models (Subsections 5.5 and 5.4) provide strong empirical evidence for this robustness.

**Theoretical Guidance on the Range of $K$ for Graphons.** While a full theoretical analysis under a general graphon model is beyond our scope, we can construct a semi-rigorous argument by combining our test statistic's structure with recent oracle inequalities for graphon estimation [21]. This analysis, detailed in Appendix C.3, shows that the power of the test depends on balancing three competing factors: (i) the model approximation error (bias), which decreases as $K$ grows; (ii) two sources of statistical estimation error (variance), which increase with $K$. For the test to be consistent, all three error terms must vanish relative to the signal strength. This leads to a sufficient set of conditions on the growth of $K$:

$$K \to \infty, \quad \text{while} \quad K^2 = o(n\nu_n) \quad \text{and} \quad \log K = o(\nu_n).$$

These conditions provide valuable theoretical guidance: $K$ should grow with the network size to ensure the SBM is a good approximation, but this growth is limited by the network's size and sparsity. This leaves a wide and practical range for choosing $K$.

**Practical Selection of $K$.** In practice, where the optimal $K$ is unknown, we propose a data-driven procedure akin to a parametric bootstrap to select a value that maximizes test power for a given problem scale:

(i) For a candidate $K_0$ (e.g., $K_0 = 10$), fit a $K_0$-SBM to one sample, yielding estimates $(\widehat{B}, \widehat{\pi})$.
(ii) Create a synthetic alternative by perturbing the estimate: $\widehat{B}_{alt} = \widehat{B} + \text{Noise}$.
(iii) For a range of values $K \in [2, K_{\max}]$, generate many synthetic datasets under $H_0 : (\widehat{B}, \widehat{B})$ and $H_1 : (\widehat{B}, \widehat{B}_{alt})$.
(iv) For each $K$, calculate the Area Under the ROC Curve (AUC) of our test in distinguishing these synthetic samples.
(v) Select the $K$ that yields the highest AUC.

In our experiments, this procedure proved effective. For instance, in the graphon experiment of Section 5.5, it correctly suggested a small value of $K = 2$.

Table 1: Mean AUC-ROC for various methods for the general SBM experiment.

| $K$ | 2 | 3 | 15 | 20 |
|---|---|---|---|---|
| SBM-TS | 0.95 | 0.91 | 0.83 | 0.81 |
| ASE | 0.62 | 0.54 | 0.49 | 0.48 |
| NCLM | 0.70 | 0.64 | 0.50 | 0.50 |

## 5.2 Competing methods

We benchmark our method by comparing it against the two test from the literature: (1) Network Clustering based on Log Moments (NCLM) [28] and (2) the Maximum Mean Discrepancy of the Adjacency Spectral Embeddings (ASE-MMD) [37]. Whilst these two methods were not originally designed for two-sample testing (NCLM is a clustering algorithm for networks and ASE-MMD is designed for sample size 1), there is a natural way to extend them into this setting. See section D.1 of supplementary material for a more detailed discussion.

## 5.3 General SBM with random $B$

We generate 50 instances of the testing problem $B_1 = B_2 = \rho B^{(i)}$ versus $(B_1, B_2) = (\rho B^{(i)}, \rho B_\varepsilon^{(i)})$, where $B^{(i)}$ is a symmetric matrix whose entries are drawn from $\mathsf{U}(0.2, 0.7)$ and $B_\varepsilon^{(i)} = B^{(i)} + M_\varepsilon^{(i)}$ such that $M_\varepsilon^{(i)}$ has entries $N(0, \varepsilon^2)$ subject to symmetry. We set the sample sizes to $N_r = 100$, the number of vertices to $n_{rt} = 10000$, the noise level to $\varepsilon = 0.05$, and sparsity factor to $\rho = 0.1$.

From Table 1, we can see that SBM-TS exhibits superior performance over the competitors, but its performance somewhat deteriorates as $K$ increases. The primary reason is that significantly increasing $K$ makes it more challenging to satisfy the approximate matching condition (14). Nonetheless, the median area under the ROC curve (AUC) for $K \in \{15, 20\}$ is 0.89 and 0.87 respectively, which shows the strong performance of our proposed test in the cases where (14) holds.

## 5.4 Random dot product graphs

We consider two experiments around the two sample testing problem

$$H_0 : A, A' \sim \mathsf{RDPG}_n(\mathbb{F}, \rho), \quad \text{against} \quad H_1 : A \sim \mathsf{RDPG}_n(\mathbb{F}, \rho), \ A' \sim \mathsf{RDPG}_n(\mathbb{F}', \rho)$$

We set the number of vertices to $n = 10000$ and the sparsity parameter $\rho = 0.15$. For Experiment 1, we take $\mathbb{F} = \mathbb{F}_1 := N(0, \Sigma)$ and $\mathbb{F}' = N(0, I_2)$ where $\Sigma = \begin{bmatrix} 3 & 2 \\ 2 & 3 \end{bmatrix}$. For Experiment 2, $\mathbb{F} = \mathbb{F}_2 := \frac{1}{2} N(0, \Sigma) + \frac{1}{2} N(0, O\Sigma O^\top)$ while $\mathbb{F}' = N(0, I_2)$ as in Experiment 1. Here, $\mathbb{F}_2$ is a mixture of two Gaussian distributions and $O = \begin{pmatrix} 0 & 1 \\ -1 & 0 \end{pmatrix}$. Due to the invariance of the RDPG to an orthogonal transformation, the two experiments are equivalent.

Figure 1c summarizes the results, comparing the performance of SBM-TS to ASE-MMD. While SBM-TS essentially exhibits the ROC of the perfect test, ASE-MMD shows poor performance. The reason is that, as proposed in [37], ASE-MMD does not have a matching mechanism to account for the potential orthogonal transformation mismatch between the estimated latent positions for $A$ and $A'$ under null. This defect is present in the original paper [37] as can be seen from the presence of the orthogonal matrix $\boldsymbol{W}_{n,m}$ in the asymptotic null limit of the statistic in Theorem 5 of [37].

## 5.5 Graphon

Consider the two-sample testing problem

$$H_0 : A, A' \sim \mathsf{Graphon}(\rho W), \quad \text{against} \quad H_1 : A \sim \mathsf{Graphon}(\rho W), \ A' \sim \mathsf{Graphon}(\rho W_{\varepsilon, \delta})$$

where $\rho$ is a sparsity factor. We take $W(v_1, v_2) = \frac{1}{4}(v_1^2 + v_2^2 + \sqrt{v_1} + \sqrt{v_2})$, the graphon from [2], and let $W_{\varepsilon, \delta}$ be the following perturbation of $W$,

$$W_{\varepsilon, \delta}(v_i, v_j) = W(v_i, v_j) + \varepsilon \cdot 1\{v_i, v_j \in [0.5 - \delta; 0.5 + \delta]\}.$$

In this experiment, we let $\varepsilon = 0.05$, $\delta = 0.2$. For a general graphon, spectral clustering does not provide a good SBM fit. To fit a proper $K$-SBM, we use a Bayesian community detection algorithm [35]. The algorithm of Section 5.1 suggests $K = 2$ as the optimal choice and we set $d = K$ for ASE-MMD. Figure 1f illustrates the resulting ROCs for two cases of $(n, \rho)$, namely, $(10^3, 0.5)$ and $(10^4, 0.05)$, showing a clear advantage for SBM-TS against the ASE-MMD.

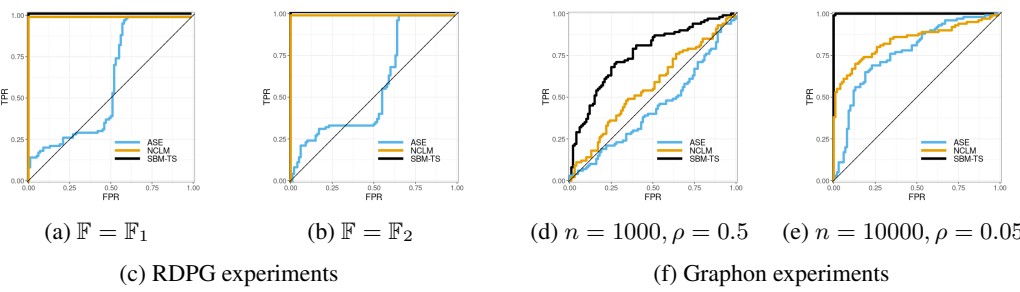

(a) $\mathbb{F} = \mathbb{F}_1$       (b) $\mathbb{F} = \mathbb{F}_2$       (d) $n = 1000, \rho = 0.5$       (e) $n = 10000, \rho = 0.05$

(c) RDPG experiments            (f) Graphon experiments

Figure 1: ROC curves for the RDPG and Graphon experiments

## 5.6 COLLAB dataset

The COLLAB dataset is a scientific collaboration dataset first introduced in [40]. This dataset contains 5000 networks derived from public collaboration data in three scientific fields: $\mathcal{C}_1$) High energy Physics, $\mathcal{C}_2$) Condensed Matter Physics, and $\mathcal{C}_3$) Astrophysics. Each graph corresponds to an ego-network of a researcher and is labeled by their primary field of research. We consider two-sample testing problems of the form (1) with $N_1 = N_2 = m$; the two samples under null will be drawn at random (without replacement) both from class $\mathcal{C}_i$, and under the alternative from classes $\mathcal{C}_i$ and $\mathcal{C}_j$ for $i \neq j$. Two choices of $m \in \{5, 10\}$ and several choices of $(i, j)$ are considered, as detained in Figure 2 where the resulting ROCs are plotted. One observes that SBM-TS has superior or comparable performance to the competing methods in distinguishing the two classes in each case.

## 6 Discussion

We have introduced a practical and theoretically-grounded framework for the two-sample testing of unlabeled networks, addressing the core challenge of permutation invariance with a scalable spectral matching algorithm.

Our approach is inherently modular, separating the problem into a matching stage and a testing stage. This structure opens several avenues for future work. For instance, while our current method solves the matching problem in two sequential steps, one could explore alternative optimization strategies, such as alternating minimization over permutations and signs. Low-rank approaches that match only the most informative eigenvectors instead of full matrices are another option.

The choice of the stochastic block model was motivated by its ability to approximate more general network models. Our experiments successfully demonstrated that our test is a robust and practical option for non-SBMs, such as those from graphon distributions. A key theoretical next step is to extend our analysis to provide formal guarantees for the test under this broader class of models, solidifying the connection between SBM-based testing and non-parametric network analysis.

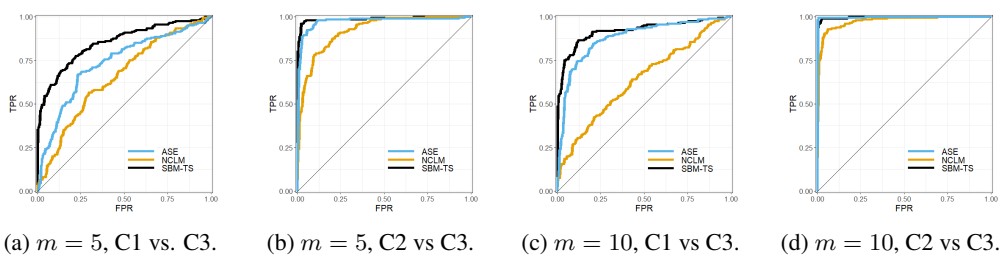

(a) $m = 5$, C1 vs. C3.    (b) $m = 5$, C2 vs C3.    (c) $m = 10$, C1 vs C3.    (d) $m = 10$, C2 vs C3.

Figure 2: ROC curves for the COLLAB dataset.

## Acknowledgments

This material is based upon work supported by the National Science Foundation under Award No. 1945667.

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

# A  Full Theoretical Statements

This section provides the formal definitions and the complete versions of the theoretical results presented in Section 4. We follow a top-down approach, beginning with the general theorems expressed via the misclassification rate (Tier 2), then presenting the full finite-sample bounds from which they are derived (Tier 3), and finally providing the complete proofs.

## A.1  Formal Definition and Matching Guarantee

We begin with the formal definition of an $(\eta, \theta)$-friendly matrix, which is a key condition for our spectral matching algorithm.

**Definition A.1** $((\eta, \theta)$-friendly$)$. Let $B \in [0, 1]^{K \times K}$ be a symmetric matrix with eigenvalue decomposition $B = \sum_{k=1}^{K} \lambda_k q_k q_k^\top$. We call $B$ $(\eta, \theta)$-friendly if:

$$\min_{1 \leq k \neq \ell \leq K} |\lambda_k - \lambda_\ell| > \eta, \qquad \text{(Eigenvalue Separation)} \qquad (12)$$

$$\min_{k \in [K]} |q_k^\top \mathbf{1}| > \theta. \qquad \text{(Eigenvector Condition)} \qquad (13)$$

We consider connectivity matrices $B_1$ and $B_2$ that are $(\theta, \eta)$-friendly. Recall our shorthand notation for an exact matching, introduced in (47). Similarly, it is helpful to introduce the following graphical mnemonic for approximate matching:

$$B_1 \xrightarrow{P} B_2 \quad \Longleftrightarrow \quad \|B_2 - PB_1P^\top\|_F \leq \frac{\theta\eta}{2\sqrt{2K}}. \qquad (14)$$

Then, we have following guarantee for the matching algorithm:

**Theorem A.1** (Matching Consistency). Suppose that $B_1$ and $B_2$ are $K \times K$ connectivity matrices that are $(\theta, \eta)$-friendly with $\theta < \frac{1}{4\sqrt{K}}$, and with EVDs given by $B_r = Q_r \Lambda Q_r^\top$, $r = 1, 2$. Moreover, assume $B_1 \xrightarrow{P^*} B_2$ for some permutation matrix $P^*$. For $r = 1, 2$, let $\widehat{B}_r$ be an estimate of $B_r$ satisfying

$$\widehat{B}_r \xrightarrow{P_r} B_r \qquad (15)$$

for some permutation matrices $P_r$. Let $\widehat{Q}_1, \widehat{Q}_2, \widehat{S}$ and $\widetilde{P}$ be as defined in Algorithm 1, and let

$$S_r = \operatorname*{argmin}_{S \in \Psi_K} \|Q_r - P_r \widehat{Q}_r S\|_F, \quad r = 1, 2, \qquad (16)$$

where $\Psi_K$ is the set of $K \times K$ diagonal sign matrices. Then, the following holds:

   (a) *Sign recovery:* $\widehat{S} = \widetilde{S} := S_1 S^* S_2$ with $S^*$ as in Lemma 2.1.
   (b) *Permutation recovery:* $\mathrm{LAP}(\widehat{Q}_1 \widetilde{S} \widehat{Q}_2^\top) = \{\widetilde{P}\}$ where $\widetilde{P} := P_2^\top P^* P_1$, and

$$\|\widehat{B}_2 - \widetilde{P}\widehat{B}_1 \widetilde{P}^\top\|_F \leq \sum_{r=1}^{2} \|B_r - P_r \widehat{B}_r P_r^\top\|_F. \qquad (17)$$

**Remark A.2.** One can also ask whether our matching algorithm is *self-consistent*? That is, if we reverse the order of the $\widehat{B}_1$ and $\widehat{B}_2$, do we get permutations that are transposes of each other? The

Figure 3: Schematic diagram of permutation recovery in Theorem A.1. The solid and dashed straight arrows correspond to exact and approximate match. The bent arrow represents an application of the matching algorithm $\mathcal{M}$.

answer is yes. Notice that if we reverse the order of $\widehat{B}_1$ and $\widehat{B}_2$, the sign matrix $\hat{S}$ does not change. Moreover,

$$B_1 \xrightarrow{P^*} B_2 \quad \text{implies} \quad B_2 \xrightarrow{(P^*)^\top} B_1.$$

Hence, $\mathrm{LAP}(\widehat{Q}_2 \widetilde{S} \widehat{Q}_1^\top) = \{P'\}$ where $P' = P_1^\top (P^*)^\top P_2 = (P_2^\top P^* P_1)^\top = \widetilde{P}^\top$.

**Intuituon behind Theorem A.1**  To gain a deeper understanding of why our target permutation matrix is $P_2^\top P^* P_1$, let us analyze Figure 3. Within this illustration, each edge symbolizes a matching between matrices based on both the direction of the edge and the permutation matrix associated with it. To clarify the nature of these matchings, we use different types of arrows:

1. Dashed straight arrows represent approximate matching. For instance, the top edge corresponds to $\widehat{B}_1 \overset{P_1}{\rightsquigarrow} B_1$, following the notation from Equation (14).

2. In contrast, the solid straight solid arrow signifies exact matching corresponding to $B_1 \xrightarrow{P^*} B_2$, with the notation from (5).

Considering that the inverse of a permutation matrix is its transpose, we can trace the path from $\widehat{B}_1$ to $\widehat{B}_2$ in Figure 3. We start from $\widehat{B}_1$, progress through $B_1$, then $B_2$, and finally arrive at $\widehat{B}_2$. The permutation matrices associated with this path are $P_1$, $P^*$ and $P_2^\top$ (the inverse of $P_2$). By multiplying these matrices together, we obtain the desired permutation matrix, which is $P_2^\top P^* P_1$.

## A.2  General Results via Misclassification Rate (Tier 2)

The simplified theorems in the main paper (Theorems 4.2 and 4.3) are direct consequences of the more general results presented here. These theorems are applicable to *any* community detection algorithm, with their conditions expressed in terms of the algorithm's misclassification rate, $\widetilde{\varepsilon} := \max_{r,t} \mathrm{Mis}(z_{rt}, \widehat{z}_{rt})$. In what follows, $N = \max\{N_1, N_2\}$.

**Theorem A.3** (Null Distribution, Tier 2)**.**  Assume Algorithm 2 is applied to two SBM samples generated from $B = (\nu_n/n)B^0$ for a fixed $B^0$. The null distribution $\widehat{T}_n \Rightarrow \chi^2_{K(K+1)/2}$ holds if:

(i) $B^0$ has distinct eigenvalues and no eigenvector of it sums to zero.

(ii) The number of networks grows at most polynomially: $N = o(n^\alpha)$ for some $\alpha > 0$.

(iii) The misclassification rate $\widetilde{\varepsilon}$ satisfies both:

$$\sqrt{\nu_n n^{1+\alpha}} \cdot \widetilde{\varepsilon} = o_p(1),$$
$$\widetilde{\varepsilon} \le CK^{-3/2} \quad \text{for some constant } C = C(B^0).$$

**Theorem A.4** (Consistency, Tier 2)**.**  Assume Algorithm 2 is applied to two SBM samples from $B_1 = (\nu_n/n)B_1^0$ and $B_2 = (\nu_n/n)B_2^0$. The test is consistent, with $\widehat{T}_n = \Omega(N\nu_n^2)$, if:

(i) $B_1^0$ and $PB_2^0 P^\top$ are different for any permutation matrix $P$.

(ii) The misclassification rate $\widetilde{\varepsilon}$ satisfies both:

$$\widetilde{\varepsilon} = o_p(1),$$
$$\widetilde{\varepsilon} \leq CK^{-3/2} \quad \text{for some constant } C = C(B_1^0, B_2^0).$$

**Remark A.5** (Connecting Tiers). The Tier 1 results in the main paper follow from the theorems above. Near-optimal community detection algorithms [42] can achieve $\widetilde{\varepsilon} = O(N \exp(-c\nu_n/K))$. Taking $\nu_n = C \log n$ for a sufficiently large $C$, this rate is small enough to satisfy all conditions on $\widetilde{\varepsilon}$ in Theorems A.3 and A.4 for sufficiently large $n$ (assuming $K$ is fixed). Note that in the main paper, we stated $N = O(n^a)$ while here $N = o(n^\alpha)$, noting that the former implies the latter by taking say $a \leq \alpha/2$ for any $\alpha > 0$. The next section details how these Tier 2 conditions are, in turn, derived from our full finite-sample analysis.

### A.3  Complete Finite-Sample Bounds (Tier 3)

The Tier 2 results presented above are convenient simplifications of our full, non-asymptotic technical bounds (Tier 3). We state these full bounds below and explain how the Tier 2 conditions arise from them.

We make the following "sparsity" and "size" assumptions

$$\frac{\nu_n}{n} \leq B_{k\ell} \leq \frac{C_1 \nu_n}{n}, \tag{18}$$

$$n \leq n_{rt} \leq C_2 n \tag{19}$$

for all $k, \ell \in [K]$ and $t \in [N_r], r = 1, 2$. We also write $\pi_k = \mathbb{P}(z_{rti} = k)$, recalling that we are under the null, and set $\pi_{\min} = \min_k \pi_k$.

Let us fix some $\kappa \in (0, 1)$ and $\alpha > 0$ and let $\beta = 1/(\bar{\kappa}\pi_{\min})$ where $\bar{\kappa} = 1 - \kappa$. For any $N$, let

$$\delta_n^{(N)} := \sqrt{\frac{3\beta^2 \alpha \log n}{Nn\nu_n}}. \tag{20}$$

For convenience, we assume that ($n$ is large enough so that)

$$\delta_n^{(1)} \leq 1, \quad \frac{\log n}{n} \leq \frac{\kappa^2 \pi_{\min}}{3\alpha}. \tag{21}$$

Moreover, define

$$\gamma_n = C_1 \Big(56\beta^3 \cdot \widetilde{\varepsilon} + \delta_n^{(1)}\Big), \quad \text{where} \quad \widetilde{\varepsilon} := \max_{\substack{t \in [N_r] \\ r = 1,2}} \text{Mis}(z_{rt}, \widehat{z}_{rt}).$$

**Theorem A.6** (Null distribution, Tier 3). Fix $K$ and assume that Algorithm 2 is applied to two SBM samples, with sizes satisfying (19), and the same connectivity matrix $B$ satisfying (18) and $\|B\|_{\max} \leq 0.99$. Moreover, suppose that $B$ is $(\eta, \theta)$-friendly and with probability $1 - o(1)$,

$$\frac{\nu_n}{n}\gamma_n \leq \min\Big(\theta, \frac{1}{5\sqrt{K}}\Big) \cdot \frac{\eta}{2\sqrt{2}K}. \tag{22}$$

Let $N = \max\{N_1, N_2\}$ and assume that for some $\kappa \in (0, 1)$ and $\alpha > 0$ and $c > 0$, we have

$$\gamma_n = o_p(1), \quad \sqrt{N\nu_n n}\,\widetilde{\varepsilon} = o_p(1), \quad N\nu_n n \to \infty, \quad N = o(n^\alpha),$$

and $\min\{N_1, N_2\} \geq cN$. Then, for $\widehat{T}_n$, the output of Algorithm 2, we have

$$\widehat{T}_n \Rightarrow \chi^2_{K(K+1)/2}.$$

**Derivation of Tier 2 from Tier 3 (Null):** The conditions in Theorem A.3 ensure that the premises of Theorem A.6 hold. Specifically, since we always have $\delta_n^{(1)} = o(1)$, the condition $\gamma_n = o_p(1)$ is satisfied if $\widetilde{\varepsilon} = o_p(1)$. Assuming $B = (\nu_n/n)B^0$, and $B_0$ has distinct eigenvalues, with no eigenvector that is orthogonal to the all-ones vector, $B$ will be $((\nu_n/n)\eta_0, \theta_0)$-friendly for some constants $\theta_0 > 0$ and $\eta_0 > 0$. As a result condition (22) holds if $\gamma_n \leq \min(\theta_0, \frac{1}{5\sqrt{K}}) \cdot \frac{\eta_0}{2\sqrt{2}K}$,

and since $\gamma_n \asymp \widetilde{\varepsilon}$, the condition holds if $\widetilde{\varepsilon} \lesssim K^{-3/2}$. The condition $\sqrt{\nu_n n^{1+\alpha}} \cdot \widetilde{\varepsilon} = o_p(1)$ is a slightly stronger restatement of the $\sqrt{N\nu_n n} \cdot \widetilde{\varepsilon} = o_p(1)$ requirement, using $N = o(n^\alpha)$. Condition $N\nu_n n \to \infty$ trivially holds given $\nu_n = \Omega(\log n)$.

Next, we show that the test is consistent, in the sense that $\widehat{T}_n \to \infty$ under the alternative hypothesis that the two SBMs are different. For two $K \times K$ matrices $B_1, B_2$, consider the pseudometric

$$d_F^{\mathrm{ma}}(B_1, B_2) := \min_{P \in \Pi_K} \|B_2 - P B_1 P^\top\|_F, \tag{23}$$

where $\Pi_K$ is the set of $K \times K$ permutation matrices.

**Theorem A.7** (Consistency, Tier 3)**.** Assume that Algorithm 2 is applied to two SBM samples, with sizes satisfying (19), and connectivity matrices $B_1$ and $B_2$ satisfying (18). For $r = 1, 2$, let

$$\xi_r := C_1\big(40\beta^3\widetilde{\varepsilon} + \delta_n^{(N_r)}\big)$$

and assume that

$$\sqrt{12}K \max_{r=1,2} \xi_r \leq \frac{d_F^{\mathrm{ma}}(B_1, B_2)}{\nu_n/n}. \tag{24}$$

Moreover, let let $\mathcal{G}$ be the event that (22) holds. and assume that $48C_2\beta^4\widetilde{\varepsilon} \leq 1$. Here, $C_1$ and $C_2$ are the same constants as in (18). Then, with probability at least $1 - 19N_+K^2 n^{-\alpha} - P(\mathcal{G}^c)$,

$$\widehat{T}_n \geq \frac{Nn^2}{12\beta^2} \cdot d_F^{\mathrm{ma}}(B_1, B_2)^2.$$

**Derivation of Tier 2 from Tier 3 (Consistency):** The premise of Theorem A.7 is dominated by the misclassification rate $\widetilde{\varepsilon}$ through the term $\xi_r$. If we assume $\widetilde{\varepsilon} = o_p(1)$ as in Theorem A.4, then $\xi_r \to 0$. Since $d_F^{\mathrm{ma}}(B_1, B_2)/(\nu_n/n) = d_F^{\mathrm{ma}}(B_1^0, B_2^0) > 0$ is a constant under the alternative, the condition holds for large $n$, guaranteeing consistency. It follows that $\widehat{T}_n \geq \frac{N\nu_n^2}{12\beta^2} d_F^{\mathrm{ma}}(B_1^0, B_2^0)^2 = \Omega(N\nu_n^2)$.

# B   Proofs of the main results

## B.1   Notation

We often consider a one-to-one correspondence between permutations matrices $P \in \mathbb{R}^{K \times K}$ and permutations $\sigma$ on the set $[K] := \{1, \ldots, K\}$. The correspondence $\sigma \mapsto P_\sigma$ is defined by the following identity: $(P_\sigma v)_i = v_{\sigma(i)}$ for all $v \in \mathbb{R}^K$. It follows that $(P_\sigma^\top v)_i = v_{\sigma^{-1}(i)}$. It is also helpful to note that $[P_\sigma]_{i*} = e_{\sigma(i)}^\top$ and $[P_\sigma^\top]_{*j} = [P_\sigma]_{j*}^\top = e_{\sigma(j)}$. This implies that if $B$ is a $K \times K$ matrix, $[P_\sigma B P_\sigma^\top]_{ij} = B_{\sigma(i),\sigma(j)}$. The linear assignment problem is often written as $\max_\sigma \sum_{i=1}^K B_{i,\sigma(i)}$. The cost function in this case is equivalent to $\mathrm{tr}(BP_\sigma^\top)$. This follows by noting $[BP_\sigma^\top]_{ij} = B_{i*} e_{\sigma(j)} = B_{i,\sigma(j)}$.

Let $P$ and $Q$ be matrices with associated permutations $\sigma$ and $\tau$ respectively. Notice that $\sigma \circ \tau$ is the permutation associated with $QP$, since $(QPv)_i = (Pv)_{\tau(i)} = v_{\sigma(\tau(i))}$.

For $p \in [1, \infty)$ and a $n \times m$ matrix $A$, the $\ell_p \to \ell_p$ operator norm of a matrix $A = (a_{ij})$ is $\|A\|_p := \sup_{x \neq 0} \|Ax\|_p/\|x\|_p$. In the special cases $p = 1, \infty$, we have $\|A\|_1 = \max_j \sum_i |a_{ij}|$ and $\|A\|_\infty = \max_i \sum_j |a_{ij}|$. We also write $\|A\|_{\max} = \max_{i,j} |A_{ij}|$.

For label vectors $z, \widehat{z} \in [K]^n$, we write $d_{\mathrm{H}}(z, \widehat{z}) = \sum_{i=1}^n \mathbf{1}\{z_i \neq \widehat{z}_i\}$ for their Hamming distance, $d_{\mathrm{NH}}(z, \widehat{z}) = d_{\mathrm{H}}(z, \widehat{z})/n$ for their the normalized Hamming distance, and $\mathrm{Mis}(z, \widehat{z}) = \min_{\sigma \in \Pi_K} d_{\mathrm{NH}}(z, \widehat{z} \circ \sigma)$ for the corresponding misclassification rate.

## B.2   Proof of Theorem A.1 (Matching Consistency)

Recall that $\|\Delta\|_\infty = \max_{i,j} |\Delta_{ij}|$. We need the following lemma on the perturbation of the LAP problem:

**Lemma B.1.** We have $\mathrm{LAP}(I + \Delta) = \{I\}$ as long as $\|\Delta\|_\infty < 1/2$.

Since $B_1$ and $B_2$ are similar matrices, they have the same eigenvalues. If the EVD of $B_1$ is $B_1 = Q_1 \Lambda Q_1^\top$, then the EVD of $B_2$ can written as $B_2 = Q_2 \Lambda Q_2^\top$. By Lemma 2.1,

$$Q_2 = P^* Q_1 S^* \tag{25}$$

for some sign matrix $S^*$.

First, we show assertion (a). Let $\widehat{\Delta}_r := P_r \widehat{Q}_r S_r - Q_r$, so that $\widehat{Q}_r = P_r^\top (Q_r + \widehat{\Delta}_r) S_r$ and

$$\widehat{Q}_r^\top \mathbf{1} = S_r (Q_r + \widehat{\Delta}_r)^\top \mathbf{1}$$

using $P_r \mathbf{1} = \mathbf{1}$. Then, $[\widehat{Q}_r^\top \mathbf{1}]_k = S_{r,kk}(Q_{r,*k}^\top \mathbf{1} + \widehat{\Delta}_{r,*k}^\top \mathbf{1})$ where $Q_{r,*k}$ and $\Delta_{r,*k}$ are the $k$-th columns of $Q_r$ and $\Delta_r$. From the definition of $\widehat{S}_{kk}$,

$$\widehat{S}_{kk} = \operatorname{sign}\left(\frac{[\widehat{Q}_1^\top \mathbf{1}]_k}{[\widehat{Q}_2^\top \mathbf{1}]_k}\right) = \operatorname{sign}\left(\frac{Q_{1,*k}^\top \mathbf{1} + \widehat{\Delta}_{1,*k}^\top \mathbf{1}}{Q_{2,*k}^\top \mathbf{1} + \widehat{\Delta}_{2,*k}^\top \mathbf{1}}\right) S_{1,kk} S_{2,kk}. \tag{26}$$

From (25), it follows that $S_{kk}^* = \operatorname{sign}\left([Q_2^\top \mathbf{1}]_k / [Q_1^\top \mathbf{1}]_k\right)$. Then, to show $\widehat{S}_{kk} = S_{1,kk} S_{kk}^* S_{2,kk}$, it suffices to show that

$$|Q_{r,*k}^\top \mathbf{1}| = \left|\sum_{j=1}^K Q_{r,jk}\right| > \left|\sum_{j=1}^K \widehat{\Delta}_{r,jk}\right| = |\widehat{\Delta}_{r,*k}^\top \mathbf{1}| \tag{27}$$

for $r = 1, 2$. Since $B_r$ are $(\theta, \eta)$-friendly, we have $|\sum_{j=1}^K Q_{r,jk}| \geq \theta$ and so it is enough to show that $|\sum_{j=1}^K \widehat{\Delta}_{r,jk}| \leq \|\widehat{\Delta}_{r,*k}\|_1 < \theta$.

The noise matrices $\widehat{\Delta}_r$ are controlled by the Davis–Kahan theorem,

$$\|\widehat{\Delta}_{r,*k}\|_2 = \|Q_{r,*k} - P_r \widehat{Q}_{r,*k} S_{r,kk}\|_2$$
$$\leq \frac{2\sqrt{2}}{\eta}\|B_r - P_r \widehat{B}_r P_r^\top\|_F.$$

Then, using assumption (15),

$$\|\widehat{\Delta}_{r,*k}\|_1 \leq \sqrt{K}\|\widehat{\Delta}_{r,*k}\|_2 \leq \frac{2\sqrt{2K}}{\eta}\frac{\theta\eta}{2\sqrt{2}K} \leq \theta, \tag{28}$$

proving assertion (a). Note also that we have shown

$$\|\widehat{\Delta}_r\|_1 = \max_k \|\widehat{\Delta}_{r,*k}\|_1 \leq \theta. \tag{29}$$

Next, we prove $\operatorname{LAP}(\widehat{Q}_1 \widetilde{S} \widehat{Q}_2^\top) = \{\widetilde{P}\}$. Using (25), and the definitions of $\widehat{\Delta}_r$, we have

$$\widehat{Q}_2 = P_2^\top (Q_2 + \widehat{\Delta}_2) S_2$$
$$= P_2^\top (P^* Q_1 S^* + \widehat{\Delta}_2) S_2$$
$$= P_2^\top (P^* (P_1 \widehat{Q}_1 S_1 - \widehat{\Delta}_1) S^* + \widehat{\Delta}_2) S_2$$
$$= P_2^\top P^* P_1 \widehat{Q}_1 S_1 S^* S_2 - P_2^\top P^* \widehat{\Delta}_1 S^* S_2 + P_2^\top \widehat{\Delta}_2 S_2$$
$$= \widetilde{P} \widehat{Q}_1 \widetilde{S} + \Delta$$

where we let $\Delta = -P_2^\top P^* \widehat{\Delta}_1 S^* S_2 + P_2^\top \widehat{\Delta}_2 S_2$. We can then write

$$\widehat{Q}_1 \widetilde{S} \widehat{Q}_2^\top = \widehat{Q}_1 \widetilde{S} (\widetilde{P} \widehat{Q}_1 \widetilde{S} + \Delta)^\top = \widetilde{P}^\top (I + \Delta_0). \tag{30}$$

where $\Delta_0 = \widetilde{P} \widehat{Q}_1 \widetilde{S} \Delta^\top$. It is then enough to study

$$\operatorname{LAP}\left(\widetilde{P}^\top (I + \Delta_0)\right) = \operatorname{LAP}(I + \Delta_0) \cdot \widetilde{P}$$

where the equality follows by a change-of-variable argument. The result follows from Lemma B.1 if we show $\|\Delta_0\|_\infty \leq 1/2$.

We note that for permutation and sign matrices, both $\|\cdot\|_\infty$ and $\|\cdot\|_1$ are equal to 1. Using the submultiplicative property of $\|\cdot\|_p$ for $p = 1, 2$, we have

$$\|\Delta^\top\|_\infty = \|\Delta\|_1 \le \|\widehat{\Delta}_1\|_1 + \|\widehat{\Delta}_2\|_1 < 2\theta$$

where we have used (29). Then,

$$\|\Delta_0\|_\infty \le \|\widehat{Q}_1\|_\infty \|\Delta^\top\|_\infty \le 2\theta\|\widehat{Q}_1\|_\infty.$$

Since $\widehat{Q}_1$ has unit-norm rows, that is, $\|\widehat{Q}_{1,k*}\|_2 = 1$ for all $k$, we obtain

$$\|\widehat{Q}_1\|_\infty = \max_k \|\widehat{Q}_{1,k*}\|_1 \le \sqrt{K}.$$

Putting the pieces together

$$\|\Delta_0\|_\infty \le \|\Delta_0\|_\infty \le 2\theta\sqrt{K} < 1/2$$

where the last inequality is by assumption. This proves $\mathrm{LAP}(\widehat{Q}_1\widetilde{S}\widehat{Q}_2^\top) = \{\widetilde{P}\}$.

To prove the inequality in part (b), let $D_r := B_r - P_r\widehat{B}_r P_r^\top$. Then, some algebra, using $B_2 = P^*B_1P^{*T}$, gives

$$\begin{aligned}
\widetilde{P}\widehat{B}_1\widetilde{P}^\top &= P_2^\top P^* P_1 \widehat{B}_1 P_1^\top P^{*T} P_2 \\
&= P_2^\top P^* (B_1 - D_1) P^{*T} P_2 \\
&= P_2 P^* B_1 P^{*T} P_2 - P_2^\top P^* D_1 P^{*T} P_2 \\
&= P_2^\top B_2 P_2 - P_2^\top P^* D_1 P^{*T} P_2 \\
&= P_2^\top D_2 P_2 + \widehat{B}_2 - P_2^\top P^* D_1 P^{*T} P_2,
\end{aligned}$$

and so

$$\begin{aligned}
\|\widetilde{P}\widehat{B}_1\widetilde{P}^\top - \widehat{B}_2\|_F &\le \|P_2^\top D_2 P_2 - P_2^\top P^* D_1 P^{*T} P_2\|_F \\
&\le \|D_1\|_F + \|D_2\|_F.
\end{aligned}$$

The proof is complete.

### B.3 Proof of Theorem A.6 (Null distribution, Tier 3)

If $B$ is $(\eta, \theta)$-friendly, then it is $(\eta, \theta')$-friendly with $\theta' = \min(\theta, \frac{1}{5\sqrt{K}})$. For simplicity, let us redefine $\theta$ to be $\theta'$, so that $B$ is $(\eta, \theta)$-friendly with $\theta < \frac{1}{4\sqrt{K}}$. Let $\mathcal{D}$ be the event that (22) holds. After redefinition, $\mathcal{D}$ is equivalent to

$$\frac{\nu_n}{n}\gamma_n \le \frac{\theta\eta}{2\sqrt{2}K}, \tag{31}$$

and by assumption, we have $P(\mathcal{D}) = 1 - o(1)$. Let $B_r^*$ be some (a priori) fixed version of the connectivity matrix for each of the two groups $r = 1, 2$. By Proposition F.3 and the union bound, there is an event $\mathcal{A}$ with

$$P(\mathcal{A}^c) \le 3N_+ K^2 n^{-\alpha} + KN_+ e^{-\kappa^2 n\pi_{\min}/3}$$

and permutation matrices $\widetilde{P}_{rt} \in \Pi_K$ such that on $\mathcal{A} \cap \mathcal{D}$:

$$\|\widetilde{P}_{rt}\widehat{B}_{rt}\widetilde{P}_{rt}^\top - B_r^*\|_{\max} \le C_1 \frac{\nu_n}{n}\left(56\beta^2 \cdot \widetilde{\varepsilon}_{rt} + \delta_n^{(1)}\right) \le \frac{\nu_n}{n}\gamma_n \le \frac{\theta\eta}{2\sqrt{2}K}$$

for all $t \in [N_r]$ and $r = 1, 2$, where $\widetilde{\varepsilon}_{rt} = \mathrm{Mis}(z_{rt}, \widehat{z}_{rt})$. This implies that

$$\mathcal{A} \cap \mathcal{D} \subset \mathcal{E}_2 := \left\{\widehat{B}_{rt} \overset{\widetilde{P}_{rt}}{\rightsquigarrow} B_r^* \quad \text{for all } t = 1, \ldots, N_r, \ r = 1, 2\right\}.$$

According to Lemma E.3, we can assume that, in the above, $\widetilde{P}_{rt}$ is independent of everything else.

(a) Step 5      (b) Simplification by introducing $B_r$.      (c) After conditioning on $B_r$

Figure 4: Matching to first network in group $r$. Here, $t > 1$ in the bottom level. Bent arrow is an application of the matching algorithm $\mathcal{M}$. The edges can be reversed in which case the permutation matrix should be replaced with its transpose. Left-side quantities are random, while right-side quantities are deterministic. Dashed and solid arrows correspond to approximate and exact matching, respectively. See the discussion at the end of Section 4.1 for more details on the nature of these diagrams.

Figure 4a illustrates the "matching-to-first" in step 5 of the algorithm. As this diagram shows, by Theorem A.1, on event $\mathcal{E}_2$, we have $\widehat{P}_{rt}^\top = \widetilde{P}_{rt}^\top I_K \widetilde{P}_{r1}$ that is,

$$\mathcal{E}_2 \subset \mathcal{E}_1 := \big\{ \widehat{P}_{rt} = \widetilde{P}_{r1}^\top \widetilde{P}_{rt} \big\}.$$

To simplify, let us define $B_r := \widetilde{P}_{r1}^\top B_r^* \widetilde{P}_{r1}$, that is,

$$B_r^* \xrightarrow{\widetilde{P}_{r1}^\top} B_r.$$

Note that $B_r$ is random version of the true $B_r^*$, due to the randomness of $\widetilde{P}_{r1}$, although, it is independent of everything else. Then, a little algebra shows that on $\mathcal{E}_1 \cap \mathcal{E}_2$,

$$\widehat{B}_{rt} \xrightsquigarrow{\widehat{P}_{rt}} B_r$$

for all $t \in [N_r]$ and $r = 1, 2$. Figure 4b illustrates the above inequality (the red path). Now, since $B_r, r = 1, 2$ are independent of everything else, we can condition on them and continue with the argument as if they were deterministic. The resulting diagram is shown in Figure 4c; the effect is as if we assumed $\widetilde{P}_{r1} = I_K$ and $B_r = B_r^*$. The above conditioning argument shows that we can do this without loss of generality.

From now on, we work on $(\mathcal{A} \cap \mathcal{D}) \cap \mathcal{E}_1 \cap \mathcal{E}_2 = \mathcal{A} \cap \mathcal{D}$ (by the above argument), on which we have $\widehat{P}_{rt} = \widetilde{P}_{rt}$ as discussed above. Then, from the definition of $\widehat{B}_r$ in step 7, we note

$$\widehat{B}_r - B_r = \frac{1}{N_r} \sum_{t=1}^{N_r} (\widetilde{P}_{rt} \widehat{B}_{rt} \widetilde{P}_{rt}^\top - B_r).$$

On $\mathcal{E}_2$ the $\|\cdot\|_F$ of each term on the RHS is bounded by $\theta\eta/(2\sqrt{2}K)$, and since a norm is a convex function, the same holds for the LHS. That is,

$$\widehat{B}_r \xrightsquigarrow{I_K} B_r \tag{32}$$

for $r = 1, 2$. Since we are under the null, there is a permutation $P^*$ such that $B_2 \xrightarrow{P^*} B_1$. Combining with (32), we can applyTheorem A.1—with $P_1 = P_2 = I_K$ and the roles of $B_1$ and $B_2$ switched—to conclude that $\widehat{P} = P^*$ in step 8.

**Remark B.1.** We could have assumed $B_1^* = B_2^*$ in the above argument, since we are under the null. However, when passing to $B_r, r = 1, 2$ we could lose the equality among $B_1$ and $B_2$ (due to $\widetilde{P}_{r1}, r = 1, 2$ potentially being different). Hence, we do not gain anything by making the assumption $B_1^* = B_2^*$.

**Remark B.2.** Everything up to and including (32) holds under the alternative $B_1 \neq B_2$ as well. This will be used in the proof of Theorem A.7.

The following arguments are all on $\mathcal{A}$. Let $S_{rt} = \mathcal{S}(A_{rt}, z_{rt})$, $m_{rt} = \mathcal{C}(A_{rt}, z_{rt})$ and

$$\widetilde{B}_r := \frac{S_r}{m_r}, \quad S_r := \sum_t S_{rt}, \quad m_r := \sum_t m_{rt}$$

Since $\widehat{P}_{rt} = \widetilde{P}_{rt}$, we have

$$\frac{\widehat{S}_r}{\widehat{m}_r} = \frac{\sum_t \widetilde{P}_{rt}\widehat{S}_{rt}\widetilde{P}_{rt}^\top}{\sum_t \widetilde{P}_{rt}\widehat{m}_{rt}\widetilde{P}_{rt}^\top}$$

where $\widehat{S}_r$ and $\widehat{m}_r$ are as defined in step 9. Let $N = \max\{N_1, N_2\}$. Then, from Proposition F.4 and union bound on $r = 1, 2$, there is an event $\mathcal{B}_1$ with

$$P(\mathcal{B}_1^c) \leq 2(C_2 N + 2K^2)n^{-\alpha} + 2NKe^{-\kappa^2 n\pi_{\min}/3} \tag{33}$$

such that on $\mathcal{B}_1$, we have

$$\|(\widehat{S}_r/\widehat{m}_r) - \widetilde{B}_r\|_{\max} \leq 40\, C_1 \beta^3 \frac{\nu_n}{n}\widetilde{\varepsilon} \tag{34}$$

where $\widetilde{\varepsilon} = \max_{r,t} \widetilde{\varepsilon}_{rt}$. Also, since $\widehat{P} = P^*$, we have $\widehat{S}_2' = P^*\widehat{S}_2 P^{*T}$ and $\widehat{m}_2' = P^*\widehat{m}_2 P^{*T}$. Moreover, from Proposition F.4, on $\mathcal{B}_1$,

$$\|\widetilde{B}_r - B_r\|_{\max} \leq C_1 \frac{\nu_n}{n}\delta_n^{(N_r)} \leq C_1 \frac{\nu_n}{n} \tag{35}$$

since $\delta_n^{(N_r)} \leq \delta_n^{(1)} \leq 1$. Using $\|B_r\|_{\max} \leq C_1\nu_n/n$ and triangle inequality, we have

$$\|\widetilde{B}_r\|_{\max} \lesssim \frac{\nu_n}{n}, \quad \|\widehat{S}_r/\widehat{m}_r\|_{\max} \lesssim \frac{\nu_n}{n} \tag{36}$$

where we have treated $C_1, C_2$ and $\beta^3$ as constants and absorbed them into $\lesssim$ symbol; this will do from time to time in the rest of the proof.

Next, we have $\|m_{rt} - \widetilde{P}_{rt}\widehat{m}_{rt}\widetilde{P}_{rt}^\top\|_{\max} \leq 6\beta\, n_{rt}^2\, \widetilde{\varepsilon}_{rt}$. It follows that

$$\|m_r - \widehat{m}_r\|_{\max} \leq 6\beta \sum_t n_{rt}^2\, \widetilde{\varepsilon}_{rt} \leq 6C_2^2\beta^2 N_r n^2 \widetilde{\varepsilon} \tag{37}$$

where $m_r = \sum_t m_{rt}$. Let $m_2' = P^*m_2 P^{*T}$. Then, the same bound as above holds for $\|m_2' - \widehat{m}_2'\|_{\max}$. Let $h$ be the elementwise harmonic mean of $m_1$ and $m_2'$. Since $\widehat{h}$ is the elementwise harmonic mean of $\widehat{m}_1$ and $\widehat{m}_2'$, we have

$$\|h - \widehat{h}\|_{\max} \leq 12C_2\beta^2 N n^2 \widetilde{\varepsilon} \tag{38}$$

where $N = \max\{N_1, N_2\}$. Since $h$ is elementwise the harmonic mean of $m_r, r = 1, 2$, we have

$$\min_{k,\ell} h_{k\ell} \geq \min_{k,\ell,r} [m_r]_{k\ell} \geq cNn^2/(2\beta^2). \tag{39}$$

The factor 2 is for handling the case $k = \ell$.

**Controlling $\widehat{\sigma}$.** Note that

$$\widehat{B} = \frac{\widehat{S}_1 + P^*\widehat{S}_2 P^*}{\widehat{m}_1 + P^*\widehat{m}_2 P^{*T}}.$$

Since $B_1 = P^*B_2 P^{*T}$, this is essentially an estimator like $\widehat{B}$ in Proposition F.4 based on an independent sample of size $N_+ := N_1 + N_2$ from $\mathsf{SBM}(B_1, \pi)$. It follows from Proposition F.4 that there is an event $\mathcal{B}_2$ with

$$P(\mathcal{B}_2^c) \leq (C_2 N_+ + 2K^2)n^{-\alpha} + N_+ Ke^{-\kappa^2 n\pi_{\min}/3} \tag{40}$$

such that on $\mathcal{B}_2$, we have

$$\|\widehat{B} - B_1\|_{\max} \leq C_1 \frac{\nu_n}{n}\left(40\beta^3\widetilde{\varepsilon} + \delta_n^{(N_+)}\right) \leq \frac{\nu_n}{n}\gamma_n$$

where the second inequality is by $\delta_n^{(N_+)} \leq \delta_n^{(1)}$.

Let $\sigma^2 = (\sigma_{k\ell}^2)$ where $\sigma_{k\ell}^2 = B_{1k\ell}(1-B_{1k\ell})$. The function $f(x) = x(1-x)$ has derivative satisfying $|f'(x)| \leq 1$ for $x \in [0,1]$, hence $f$ is 1-Lipschitz there implying

$$\|\widehat{\sigma}^2 - \sigma^2\|_{\max} \leq \|\widehat{B} - B_1\|_{\max}.$$

Since $\nu_n/n \leq B_{1k\ell} \leq 0.99$, we have $\sigma_{k\ell}^2 \geq 0.01\nu_n/n$. Ignoring constants, we have shown

$$\|\widehat{B} - B_1\|_{\max} \leq \frac{\nu_n}{n}\gamma_n, \quad \|\widehat{\sigma}^2 - \sigma^2\|_{\max} \leq \frac{\nu_n}{n}\gamma_n, \quad \min_{k,\ell}\widehat{\sigma}_{k\ell}^2 \gtrsim \frac{\nu_n}{n}.$$

**High probability event.** Let $\mathcal{B} = \mathcal{B}_1 \cap \mathcal{B}_2$. Then, the event $\mathcal{A} \cap \mathcal{B}$ has high probability. Indeed, we have

$$P(\mathcal{A}^c) \leq 3N_+K^2n^{-\alpha} + KN_+e^{-\kappa^2 n\pi_{\min}/3}$$
$$P(\mathcal{B}^c) \leq 3(C_2N_+ + 2K^2)n^{-\alpha} + 3N_+Ke^{-\kappa^2 n\pi_{\min}/3}.$$

Using union bound and further bounding $C_2N_+ + 2K^2 \leq 4C_2N_+K^2$ and $e^{-\kappa^2 n\pi_{\min}/3} \leq n^{-\alpha}$ (by assumption (21)), we obtain

$$P(\mathcal{A}^c \cup \mathcal{B}^c) \leq 19N_+K^2n^{-\alpha} \tag{41}$$

which goes to zero under the assumption $N_+K^2n^{-\alpha} = o(1)$.

**Controlling $\widehat{T}_n$.** Define

$$D := \frac{S_1}{m_1} - \frac{S_2'}{m_2'}, \quad \widehat{D} := \frac{\widehat{S}_1}{\widehat{m}_1} - \frac{\widehat{S}_2'}{\widehat{m}_2'},$$

$$\widehat{E} := \frac{\sqrt{\widehat{h}}}{\sqrt{2}\widehat{\sigma}}\widehat{D}, \quad E := \frac{\sqrt{h}}{\sqrt{2}\sigma}D$$

where $S_2' = P^*S_2P^{*T}$ and $m_2' = P^*m_2P^{*T}$. Note also that $S_r/m_r = \widetilde{B}_r$. Let $d = K(K+1)/2$. For the rest of the proof, we treat the above matrices as vectors in $\mathbb{R}^d$ by considering only the elements on and above the diagonal, in a particular order (say rowwise).

We have $\widehat{T}_n = \|\widehat{E}\|_F^2$ on $\mathcal{A} \cap \mathcal{B}$ and since $P(\mathcal{A} \cap \mathcal{B}) = 1 - o(1)$, it is enough to establish $\widehat{E} \Rightarrow (N(0,1))^{\otimes d}$; see [20, Theorem 9.15]. Clearly,

$$\widehat{E} = \frac{\sqrt{\widehat{h}}}{\sqrt{h}} \cdot \frac{\sigma}{\widehat{\sigma}} \cdot E + \frac{\sqrt{\widehat{h}}}{\sqrt{h}} \cdot \frac{\sigma}{\widehat{\sigma}} \cdot \frac{\sqrt{h}}{\sqrt{2}\sigma}(\widehat{D} - D). \tag{42}$$

From (34), with high probability, we have that

$$\|\frac{\sqrt{h}}{\sqrt{2}\sigma}(\widehat{D} - D)\|_{\max} \lesssim \frac{\sqrt{Nn^2}}{(\nu_n/n)^{1/2}}\frac{\nu_n\widetilde{\varepsilon}}{n} = \sqrt{Nn\nu_n}\widetilde{\varepsilon}. \tag{43}$$

Next we have

$$\left\|\frac{\sigma^2}{\widehat{\sigma}^2} - \mathbf{1}_d\right\|_{\max} \leq \frac{\|\sigma^2 - \widehat{\sigma}^2\|_{\max}}{\min_{k,\ell}\widehat{\sigma}_{k,\ell}^2} \lesssim \frac{(\nu_n/n)\gamma_n}{\nu_n/n} \leq \gamma_n$$

hence

$$\sigma/\widehat{\sigma} = \mathbf{1}_d + o_p(1). \tag{44}$$

Similarly,

$$\left\|\frac{\widehat{h}}{h} - \mathbf{1}_d\right\|_{\max} \leq \frac{\|\widehat{h} - h\|_{\max}}{\min_{k,\ell}h_{k\ell}} \lesssim \frac{Nn^2\widetilde{\varepsilon}}{Nn^2} \leq \widetilde{\varepsilon}$$

and hence

$$\sqrt{\widehat{h}}/\sqrt{h} = \mathbf{1}_d + o_p(1). \tag{45}$$

**Lemma B.2.** $E := (\sqrt{h}/(\sqrt{2}\sigma))D \Rightarrow (N(0,1))^{\otimes d}$, under the assumptions of Theorem A.6.

*Proof.* See Section E.1. $\qquad\qquad\qquad\qquad\qquad\qquad\qquad\qquad\qquad\qquad\qquad\qquad\qquad\quad$ $\square$

Therefore, from (42)–(45), Lemma B.2, and Slutsky's theorem, we obtain that $\widehat{E} \Rightarrow (N(0,1))^{\otimes d}$, provided that $\sqrt{N\nu_n n}\,\widetilde{\varepsilon} = o(1)$. The proof is complete.

## B.4 Proof of Theorem A.7 (Consistency, Tier 3)

We will follow the notation and argument in the proof of Theorem A.6. On event $\mathcal{A} \cap \mathcal{D} \cap \mathcal{B}$ defined there, we have correct matching $\widehat{P}_{rt} = \widetilde{P}_{rt}$—where $\widetilde{P}_{rt}$ is as defined in the proof of Theorem A.6— and (34) and (35) hold. Since $\mathcal{G} \subset \mathcal{D}$, all the above also holds on $\mathcal{A} \cap \mathcal{G} \cap \mathcal{B}$. This is the event we will work on.

The two inequalities (34) and (35) give

$$\|(\widehat{S}_r/\widehat{m}_r) - B_r\|_{\max} \le C_1 \frac{\nu_n}{n}\Big(40\beta^3\widetilde{\varepsilon} + \delta_n^{(N_r)}\Big) = \xi_r \frac{\nu_n}{n}, \quad r = 1, 2.$$

Since, $\widehat{S}_2'/\widehat{m}_2' = \widehat{P}(\widehat{S}_2/\widehat{m}_2)\widehat{P}^\top$, we also have

$$\|(\widehat{S}_2'/\widehat{m}_2') - \widehat{P}B_2\widehat{P}^\top\|_{\max} \le \xi_2 \frac{\nu_n}{n}.$$

Using $(a + b + c)^2 \le 3(a^2 + b^2 + c^2)$, we obtain

$$\begin{aligned}
\|(\widehat{S}_1/\widehat{m}_1) - (\widehat{S}_2'/\widehat{m}_2')\|_F^2 &\ge \frac{1}{3}\|B_1 - \widehat{P}B_2\widehat{P}^\top\|_F^2 \\
&\quad - \|(\widehat{S}_1/\widehat{m}_1) - B_1\|_F^2 - \|(\widehat{S}_2'/\widehat{m}_2') - \widehat{P}B_2\widehat{P}^\top\|_F^2 \\
&\ge \frac{1}{3}\min_P \|B_1 - PB_2P^\top\|_F^2 - 2K^2\Big(\frac{\nu_n}{n}\Big)^2 \max_{r=1,2} \xi_r^2 \\
&\ge \frac{1}{6}\min_P \|B_1 - PB_2P^\top\|_F^2
\end{aligned}$$

by assumption (24). Next, we note that (37) holds in this case. Let $m_2' = \widehat{P}m_2\widehat{P}^\top$ and let $h$ be the elementwise harmonic mean of $m_1$ and $m_2'$. Then, both (38) and (39) hold, irrespective of the specific $\widehat{P}$. On $\mathcal{G}$, we have $48C_2\beta^4\widetilde{\varepsilon} \le 1$ which, combined with the latter two inequalities, yields

$$\min_{k\ell} \widehat{h}_{k\ell} \ge Nn^2/(4\beta^2).$$

Also $\widehat{\sigma}_{k\ell}^2 = \widehat{B}_{k\ell}(1 - \widehat{B}_{k\ell}) \le 1/4$. Then,

$$\widehat{T}_n \ge \frac{Nn^2/(4\beta^2)}{2 \cdot \frac{1}{4}}\|(\widehat{S}_1/\widehat{m}_1) - (\widehat{S}_2'/\widehat{m}_2')\|_F^2. \tag{46}$$

Putting the pieces together, we have the desired inequality on $(\mathcal{A} \cap \mathcal{B}) \cap \mathcal{G}$. Combined with the probability bound (41) for $(\mathcal{A} \cap \mathcal{B})^c$, the proof is complete.

## B.5 Proof of Lemma 2.1

We restate the lemma here with the extra uniqueness clause added.

**Lemma B.3.** Consider two $K \times K$ matrices $B_1$ and $B_2$ with EVDs given by $B_r = Q_r\Lambda Q_r^\top$, $r = 1, 2$, for some diagonal matrix $\Lambda$ with distinct diagonal entries. Then, a permutation matrix $P^*$ satisfies

$$B_1 \xrightarrow{P^*} B_2 \tag{47}$$

if and only if there exists a diagonal sign matrix $S^*$, such that

$$Q_2 = P^*Q_1S^*. \tag{48}$$

Moreover if $B_1$ is $(\theta, \eta)$ friendly, then there is at most one $P^*$ satisfying (47).

*Proof.* Since $P^*Q_1$ is an orthogonal matrix, by absorbing $P^*$ into $Q_1$ and redefining $Q_1$, we can assume $P^* = I$, without loss of generality. The problem reduces to showing that (*) $Q_2 \Lambda Q_2^T = Q_1 \Lambda Q_1^T$ iff there is a sign matrix $S^*$ such that $Q_2 = Q_1 S^*$. Let $Q = Q_2^T Q_1$ and note that $Q$ is an orthogonal matrix. Multiplying (*) on the left and right by $Q_2^T$ and $Q_2$, the problem reduces to showing that (**) $\Lambda = Q \Lambda Q^T$ for an orthogonal matrix $Q$ and diagonal matrix $\Lambda$ iff $Q$ is a diagonal sign matrix (i.e., $Q = S^*$).

Assume (**) holds, the other direction being trivial. Note that changing $\Lambda$ to $\Lambda + \alpha I$ does not change (**), hence we can shift the diagonal entries of $\Lambda$ arbitrarily. Let $\Lambda = \mathrm{diag}(\lambda_k)$. Since $(\lambda_k)$ are distinct, we can shift them so that $\lambda_1 < 0$ and $\lambda_k > 0$ for $k \geq 2$. From (**), looking at the first entries of the two sides, $\lambda_1 = \sum_{k \geq 1} Q_{ik}^2 \lambda_k$, hence

$$0 = -(1 - Q_{11}^2)\lambda_1 + \sum_{k \geq 2} Q_{1k}^2 \lambda_k.$$

Every term on the RHS is non-negative. It follows that every term has to be zero, implying $Q_{11}^2 = 1$ and $Q_{1k} = 0$ for $k \geq 2$. This proves the assertion for the first row of $Q$. Repeating the argument for the other rows, the result follows.

Next, we prove the uniqueness. Suppose that there exist permutation matrices $P$ and $\tilde{P}$ such that $PB_1P^T = B_2 = \tilde{P}B_1\tilde{P}^T$. By the argument above, there exist sign matrices $S$ and $\tilde{S}$ such that $PQ_1 S = Q_2, = \tilde{P}Q_1\tilde{S}$. Hence,

$$Q_1 = P^T \tilde{P} Q_1 \tilde{S} S.$$

The problem then reduces to showing that if $Q = PQS$ where $Q$ is an orthogonal matrix, $P$ a permutation matrix and $S$ a sign matrix, then $P = I$. If $S = I$ (the trivial sign matrix), then $P = QQ^T = I$. So assume $S \neq I$ and $P \neq I$. Then, there is $j$ such that $PQ_{.,j} = -Q_{.,j}$, hence,

$$\mathbf{1}^T Q_{.,j} = \mathbf{1}^T PQ_{.,j} = -\mathbf{1}^T Q_{.,j}$$

where $\mathbf{1}$ is the all-ones vector. This gives $\mathbf{1}^T Q_{.,j} = 0$, contradicting friendliness of $B_1$. The proof is complete.

$\square$

# C  Additional Discussion

## C.1  Class proportions and community refinements

Consider the following example to illustrate how class-proportion differences can be absorbed into refined community structures. Suppose we start with two SBMs with equal connectivity matrices but differing class proportions:

$$\pi_1 = (0.5, 0.5), \quad \pi_2 = (0.4, 0.6), \quad B_1 = B_2 = \begin{pmatrix} 0.4 & 0.1 \\ 0.1 & 0.3 \end{pmatrix}.$$

To eliminate the discrepancy in proportions, we refine each community into two sub-communities, effectively doubling the number of communities. Specifically, for $\pi_1$, we split its first community into sub-communities of proportions $(0.4, 0.1)$ and its second community into $(0.1, 0.4)$, so that the total proportions of the four new sub-communities become $(0.4, 0.1, 0.1, 0.4)$. For $\pi_2$, we retain its original first community (proportion $0.4$), and split its second community (proportion $0.6$) into three sub-communities of sizes $0.1$, $0.1$, $0.4$. These four new sub-communities also have overall proportions $(0.4, 0.1, 0.1, 0.4)$. Hence the refined SBMs now have the same class proportions:

$$\tilde{\pi}_1 = \tilde{\pi}_2 = (0.4, 0.1, 0.1, 0.4).$$

However, the connectivity matrices enlarge to reflect the finer partition:

$$\tilde{B}_1 = \begin{pmatrix} 0.4 & 0.4 & 0.1 & 0.1 \\ 0.4 & 0.4 & 0.1 & 0.1 \\ 0.1 & 0.1 & 0.3 & 0.3 \\ 0.1 & 0.1 & 0.3 & 0.3 \end{pmatrix}, \quad \tilde{B}_2 = \begin{pmatrix} 0.4 & 0.1 & 0.1 & 0.1 \\ 0.1 & 0.3 & 0.3 & 0.3 \\ 0.1 & 0.3 & 0.3 & 0.3 \\ 0.1 & 0.3 & 0.3 & 0.3 \end{pmatrix}.$$

Note that class-proportion differences vanish (since $\tilde{\pi}_1 = \tilde{\pi}_2$), but the matrices $\tilde{B}_1$ and $\tilde{B}_2$ now differ explicitly. This clarifies that proportion differences can equivalently be represented as connectivity differences through community refinement, motivating our focus on testing differences in $B_r$ modulo permutations while treating class proportions as nuisance parameters.

## C.2 Empirical Analysis of Average Degree Growth

To support the claim in Remark 4.5 that the $\nu_n = \Omega(\log n)$ condition is mild in practice, we analyzed the average degree growth in several real-world network datasets from the PyTorch Geometric library. We sampled subgraphs of increasing size from each dataset and computed the ratio of the average degree to $\log n$. As shown in Table 2, this ratio tends to grow with $n$, indicating that the average degree in these networks grows faster than $\log n$. Qualitatively similar behavior is observed across other datasets (`amazon-photo`, `wikics`, `corafull`, `coauthor-cs`, and `planetoid-cora`).

Table 2: Average degree vs. size for subsets of the Amazon Computers dataset. The final column shows that the average degree grows faster than $\log n$.

| Dataset | $n$ | $n/n_{\text{total}}$ | avg_degree | avg_degree / $\log n$ |
|---|---|---|---|---|
| amazon-computers | 138 | 0.01 | 0.435 | 0.088 |
| amazon-computers | 276 | 0.02 | 0.623 | 0.111 |
| amazon-computers | 688 | 0.05 | 1.674 | 0.256 |
| amazon-computers | 1376 | 0.1 | 3.078 | 0.426 |
| amazon-computers | 2751 | 0.2 | 6.057 | 0.765 |
| amazon-computers | 6876 | 0.5 | 17.121 | 1.938 |
| amazon-computers | 13752 | 1.0 | 35.756 | 3.752 |

**Implementation.** The experiment was implemented in a lightweight PyTorch Geometric script (`avg_degree_tables_min.py`, provided in the supplementary material for reproducibility.), which randomly subsamples nodes, constructs the induced subgraph, and computes the mean degree via `torch_geometric.utils.degree`. We fix a random seed for reproducibility and report average values over multiple fractions $n/n_{\text{total}} \in \{0.01, 0.02, 0.05, 0.1, 0.2, 0.5, 1.0\}$.

## C.3 Derivation of the Permissible Range of $K$ for Graphons

This section provides the semi-rigorous derivation for the permissible range of $K$ when applying our test to general graphon models, as summarized in Section 5.1.

We aim to find conditions on $K$ that ensure our test has power when the underlying data comes from two different $\beta$-smooth graphons, $f_1$ and $f_2$. The argument combines oracle inequalities from [21] with the structure of our test statistic. We apply their results with $\rho_n = \nu_n/n$ and $n_0 = n/K$.

Let $\Theta = (\Theta_{ij}) \in [0,1]^{n \times n}$ be the matrix of connection probabilities, or "discrete graphon," where $\Theta_{ij} = \mathbb{E}[A_{ij}]$. Let $\widehat{\Theta}$ be its $K$-block estimate based on the observed adjacency matrix $A$, and let $\check{\Theta}$ be the best $K$-block approximation of $\Theta$. By Proposition 2.1 of [21], the expected squared Frobenius error of the estimate is bounded:

$$\mathbb{E}\|\widehat{\Theta} - \Theta\|_F^2 \lesssim \|\Theta - \check{\Theta}\|_F^2 + (n \log K + K^2)\Big(\frac{\nu_n}{n} + \frac{K \log K}{n}\Big).$$

Furthermore, for a $\beta$-smooth graphon, Proposition 2.5 of [21] bounds the approximation error:

$$\|\Theta - \check{\Theta}\|_F^2 \lesssim n^2\Big(\frac{\nu_n}{n}\Big)^2 K^{-2\beta} = \nu_n^2 K^{-2\beta}.$$

Assuming $\nu_n \gtrsim K \log K$, the estimation error bound simplifies to:

$$\mathbb{E}\|\widehat{\Theta} - \Theta\|_F^2 \lesssim \nu_n^2 K^{-2\beta} + (n \log K + K^2)\frac{\nu_n}{n}$$

$$= \nu_n^2 K^{-2\beta} + \nu_n \log K + \frac{\nu_n K^2}{n} =: \Delta_{n,K}.$$

Now, consider the alternative hypothesis where we have two discrete graphons, $\Theta_1$ and $\Theta_2$. Ignoring the permutation matching step for simplicity (i.e., assuming identity permutation), our test statistic $\widehat{T}_n$ is proportional to the squared difference of the estimated block models. This is, in turn, related to the squared difference of the estimated discrete graphons:

$$\frac{\widehat{T}_n}{N} \gtrsim \Big(\frac{n}{K}\Big)^2 \|\widehat{B}_1 - \widehat{B}_2\|_F^2 = \|\widehat{\Theta}_1 - \widehat{\Theta}_2\|_F^2.$$

Using a triangle inequality, we can bound this difference:

$$\|\widehat{\Theta}_1 - \widehat{\Theta}_2\|_F^2 \geq \frac{1}{3}\|\Theta_1 - \Theta_2\|_F^2 - \|\widehat{\Theta}_1 - \Theta_1\|_F^2 - \|\widehat{\Theta}_2 - \Theta_2\|_F^2.$$

The true difference is $\|\Theta_1 - \Theta_2\|_F^2 \asymp n^2(\nu_n/n)^2\|f_1 - f_2\|_{L^2}^2 = \nu_n^2\|f_1 - f_2\|_{L^2}^2$. The two estimation error terms are of order $\Delta_{n,K}$. For the test to have power, the signal term must dominate the error terms. This leads to the condition:

$$\nu_n^2\|f_1 - f_2\|_{L^2}^2 \gg \Delta_{n,K}.$$

Dividing by $\nu_n^2$, we require $\frac{\Delta_{n,K}}{\nu_n^2} = o(1)$. This yields the final condition:

$$K^{-2\beta} + \frac{\log K}{\nu_n} + \frac{K^2}{n\nu_n} = o(1),$$

A sufficient set of conditions for this to hold is $K \to \infty$ (to handle the first term), while $\log K = o(\nu_n)$ (for the second term) and $K^2 = o(n\nu_n)$ (for the third term). These also ensure the earlier assumption $\nu_n \gtrsim K \log K$. This provides the justification for the claims made in the main text.

## D    Additional experiments

Here, we provide more details of the experimental setup and report on more experiments. All experiments were conducted on an internal computing cluster with Intel(R) Xeon(R) Platinum 8160 CPUs (48 cores, 2.10GHz).

### D.1    Competing methods

In this section, we describe the competing methods used in the experiments in the main text and below. Some of them were not originally designed for two-sampling testing, but have a natural extension to this setting which we outline below. Throughout this section, we use the terms "network" and "adjacency matrix" interchangeably, since the resulting distance measures between adjacency matrices will be invariant to node permutations.

To measure a distance between two adjacency matrices, NCLM constructs a feature vector for each graph based on its so-called log-moments. To be more precise, [28] considers the $k$-th *graph moment* of a matrix $A$, $m_k(A) = \mathrm{tr}[(A/n)^k]$, which is the normalized count of closed walks of length $k$. The feature vector for an adjacency matrix $A$ is then defined as

$$g_J(A) := \big(\log m_j(A), \, j \in [J]\big)$$

where $J$ is some positive integer. The test statistic for comparing two adjacency matrices $A_1$ and $A_2$ is naturally given as the $\ell_2$-distance between the feature vectors of the two graphs:

$$d_{\mathrm{NCLM}}(A_1, A_2) := \|g_J(A_1) - g_J(A_2)\|_2.$$

Our experiments, reported in Appendix D.6, suggest that a larger value of $J$ improves performance. However, increasing $J$ quickly increases the computation cost. In the results to follow, we have chosen $J = 20$ which provides a reasonable balance between performance and cost.

To form a distance between two adjacency matrices, ASE-MMD first computes the so-called adjacency spectral embedding (ASE) for each matrix. For an adjacency matrix $A$, consider $|A| = (A^\top A)^{1/2}$ and let $|A| = \sum_{i=1}^n \lambda_i u_i u_i^\top$ be its eigen-decomposition where $\lambda_1 \geq \cdots \geq \lambda_n \geq 0$ are the eigenvalues and $\{u_i\}$ the corresponding orthonormal basis of eigenvectors. Then, the adjacency spectral embedding of $A \in \{0,1\}^{n \times n}$ into $\mathbb{R}^d$ is

$$\hat{X}(A) = U_A\sqrt{S_A} \in \mathbb{R}^{n \times d} \tag{49}$$

where $S_A = \mathrm{diag}(\lambda_1, \lambda_2, \ldots, \lambda_d)$ and $U_A$ is $n \times d$ matrix whose columns are $u_1, u_2, \ldots, u_d$. The rows of $\hat{X}(A)$ define an empirical distribution $P_{\hat{X}(A)} := \frac{1}{n}\sum_{i=1}^n \delta_{\hat{X}_{i*}(A)}$ where $\delta_x$ is the point mass as $x$. ASE-MDD then measures the distance between two adjacency matrices $A_1$ and $A_2$ as the maximum mean discrepancy of the corresponding empirical distributions:

$$d_{\mathrm{ASE\text{-}MMD}}(A_1, A_2) = \mathrm{MMD}(P_{\hat{X}(A_1)}, P_{\hat{X}(A_2)}).$$

The MMD relies on a positive definite kernel function $\kappa : \mathbb{R}^d \times \mathbb{R}^d \to \mathbb{R}$. Letting $\hat{X} = \hat{X}(A_1)$ and $\hat{Y} = \hat{X}(A_2)$, one has

$$\text{MMD}(P_{\hat{X}}, P_{\hat{Y}}) =$$
$$\frac{1}{n(n-1)} \sum_{i \neq j} \kappa(\hat{X}_{i*}, \hat{X}_{j*}) - \frac{2}{mn} \sum_{i \neq j} \kappa(\hat{X}_{i*}, \hat{Y}_{j*}) + \frac{1}{n(n-1)} \sum_{i \neq j} \kappa(\hat{Y}_{i*}, \hat{Y}_{j*}).$$

In experiments reported here, we consider a Gaussian kernel and use random Fourier features to approximate the MMD. The bandwidth is set to be $\sigma^2 = 1$. Additional experiments, reported in Appendix D.5, show that the results are not sensitive to the choice of bandwidth.

**Adapting to multiple samples.** Next, we describe how we adapt the test statistics to the cases where the sample size is greater than 1. Any measure of dissimilarity, $d(\cdot, \cdot)$, between two networks, can be generalized to the two-sample case, by averaging the dissimilarity of pairs of networks from different samples. More specifically, given the two samples $A_{rt}$, $t \in [N_r]$ for $r = 1, 2$, we have the two-sample test statistic $\frac{1}{N_1 N_2} \sum_{t=1}^{N_1} \sum_{s=1}^{N_2} d(A_{1t}, A_{2s})$.

**A note on network generation.** Throughout the experiments, we replace Bernoulli variables with a clipped version. In particular, write $X \sim \overline{\text{Ber}}(p)$ for $p \in \mathbb{R}$ to denote $X = 1\{U < p\}$ for some $U \sim [0, 1]$. When $p \in [0, 1]$, $\overline{\text{Ber}}(p) = \text{Ber}(p)$, but otherwise $\overline{\text{Ber}}(p)$ is naturally clipped to either 0 or 1. When we write $\text{SBM}(B, \pi)$ in experiments, it is based on $\overline{\text{Ber}}(p)$ generation.

### D.2 Simple 2-SBM

Our first simulation reproduces the experiment from [37] on a 2-SBM. In particular, we consider testing $H_0 : A, A' \sim \text{SBM}(B_0, \pi)$ versus $H_1 : A \sim \text{SBM}(B_0, \pi)$, $A' \sim \text{SBM}(B_\varepsilon, \pi)$, where

$$B_\varepsilon = \begin{bmatrix} 0.5 + \varepsilon & 0.2 \\ 0.2 & 0.5 + \varepsilon \end{bmatrix}, \tag{50}$$

and $\pi = (0.4, 0.6)$. Note that each sample contains a single adjacency matrix. We consider vertex sizes $n \in \{100, 200, 500, 1000\}$ and noise levels $\varepsilon \in \{0.01, 0.02, 0.05, 0.1\}$ and evaluate the power of the our SBMTS using 1000 Monte-Carlo replications. Table 3 summarizes the results alongside the power estimates for ASE-MMD reported in [37]. The results clearly show the superior performance of SBMTS along both dimensions $(n, \varepsilon)$.

Table 3: Power estimates for ASE and SBM-TS for a simple 2-block SBM experiment.

| $n$ | $\varepsilon = 0.01$ | | $\varepsilon = 0.02$ | | $\varepsilon = 0.05$ | | $\varepsilon = 0.1$ | |
| --- | --- | --- | --- | --- | --- | --- | --- | --- |
| | SBM-TS | ASE | SBM-TS | ASE | SBM-TS | ASE | SBM-TS | ASE |
| 100 | 0.047 | - | 0.161 | 0.06 | 0.776 | 0.09 | 1 | 0.27 |
| 200 | 0.15 | - | 0.599 | 0.09 | 1 | 0.17 | 1 | 0.83 |
| 500 | 0.785 | - | 1 | 0.01 | 1 | 0.43 | 1 | 1 |
| 1000 | 1 | 0.14 | 1 | 1 | 1 | 1 | 1 | 1 |

### D.3 SW–GOT dataset

The Star Wars (SW)–Game of Thrones (GOT) dataset [34] is derived from popular films and television series. We consider 13 networks: six from the original and sequel SW trilogies, and seven from each of the GOT series. In each graph, vertices correspond to characters and edges indicate whether two characters share a scene. Let us denote the SW and GOT networks as class $\mathcal{C}_1$ and $\mathcal{C}_2$, respectively. The set of characters for both classes overlap across multiple networks, but no vertex correspondence is utilized because network vertices are unlabeled and each network has different number of vertices. Sample graphs from this dataset are shown in Figure 10 in Appendix D.7.

Similar to the COLLAB dataset, we consider a two sample testing problem for distinguishing a null of $(\mathcal{C}_1, \mathcal{C}_1)$ vs. an alternative of $(\mathcal{C}_1, \mathcal{C}_2)$. The resulting ROCs are shown in Figure 5 showing a significant advantage for SBM-TS compared to the competitors.

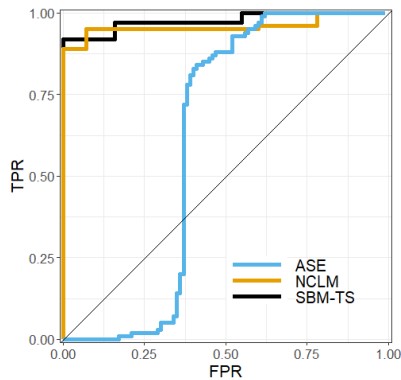

Figure 5: ROC curves for the SW-GOT dataset.

## D.4 ROC plots for general SBM with random $B$

Figure 6 shows the mean ROC curves for the experiment in Section 5.3. The different plots correspond to different values of $K$. We refer to Section 5.3 for the detailed description of the experiments, where a summary of these curves via their "area under the curve (AUC)" was provided in Table 1.

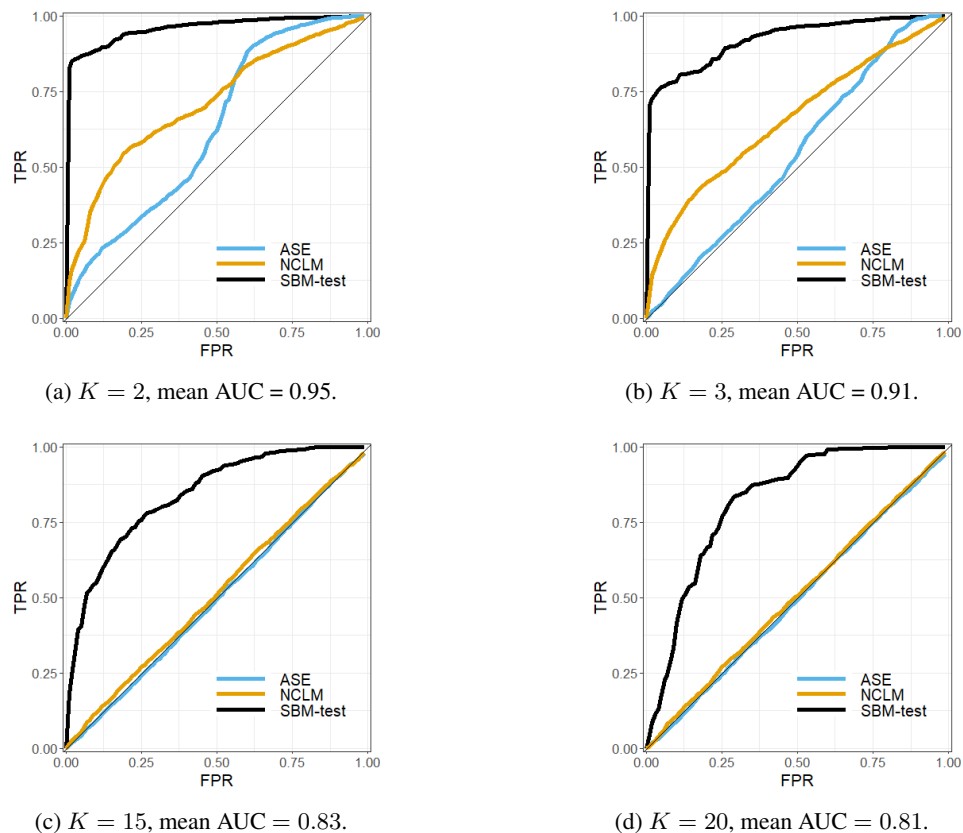

(a) $K = 2$, mean AUC $= 0.95$.

(b) $K = 3$, mean AUC $= 0.91$.

(c) $K = 15$, mean AUC $= 0.83$.

(d) $K = 20$, mean AUC $= 0.81$.

Figure 6: ROC curves for the three methods (SBM-TS, MMD of ASE, and test statistic based on NCLM) averaged over 50 different experiments.

## D.5 Bandwidth of ASE-MMD

We have conducted additional experiments in the same setting of Section 5.3, that is, general SBM with random $B$ to determine the effect of bandwidth on the performance of ASE-MMD. The results

are summarized in Figure 7. One observes that the bandwidth does not have a significant bearing on the power of the ASE-MMD test, with ROCs remaining almost the same across the range of $\sigma^2 \in \{0.01, 0.1, 1, 10, 100\}$.

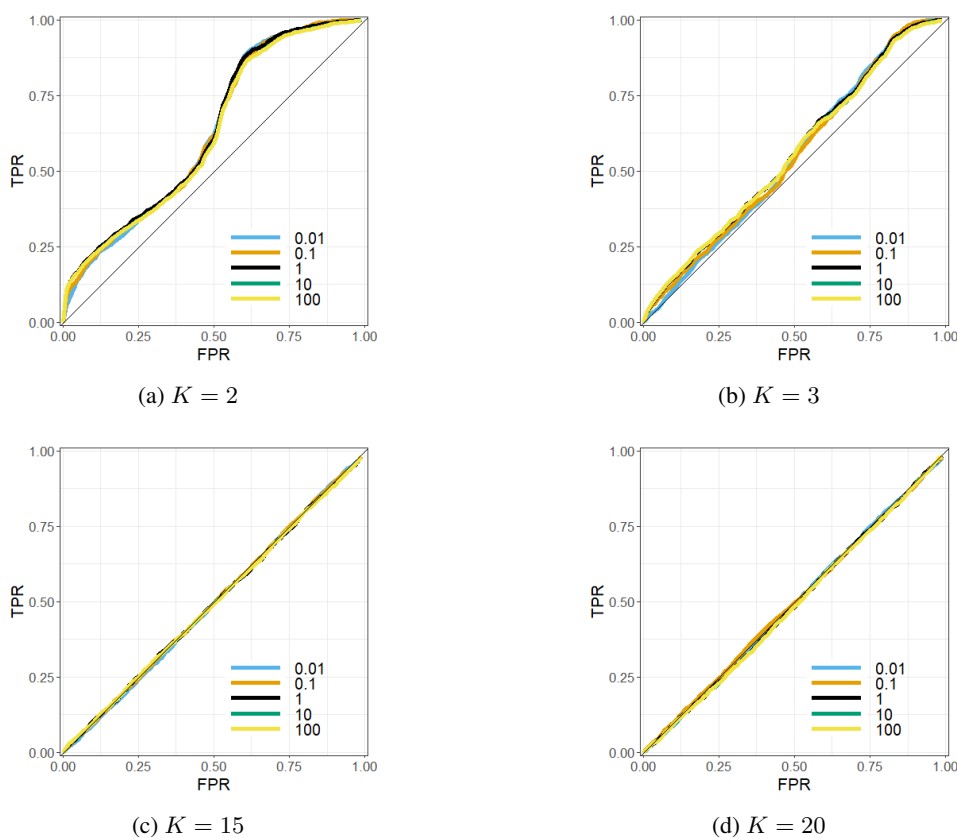

(a) $K = 2$

(b) $K = 3$

(c) $K = 15$

(d) $K = 20$

Figure 7: ROC curves for different choices of the bandwidth $\sigma^2$ in the experiment with a general $B$.

### D.6 Number of Log Moments for NCLM

We have performed additional experiments in the same setting of Section 5.3, that is, general SBM with random $B$ to determine the effect of the number $J$ of log-moments on the performance of NCLM. The results are summarized in Figure 8. The general trend is that higher $J$ improves performance with $J = 20$ (the maximum we considered) producing the best results. We also note that this is mainly for small values of $K$, while for larger $K \in \{15, 20\}$ the test is powerless regradless of the value of $J$.

### D.7 Sample graphs for real-world data

Example of graphs from the COLLAB dataset are shown in Figure 9.

### D.8 Empirical runtime of Algorithm 1 With Respect To $K$

We investigate the mean runtime of Algorithm 1 as a function of the block dimension $K$. For varying values of $K$, we generate a random $K \times K$ symmetric matrix $B_1$ and its random permutation $B_2 = PBP^\top$ and apply Algorithm 1. The results are summarized in Table 4.

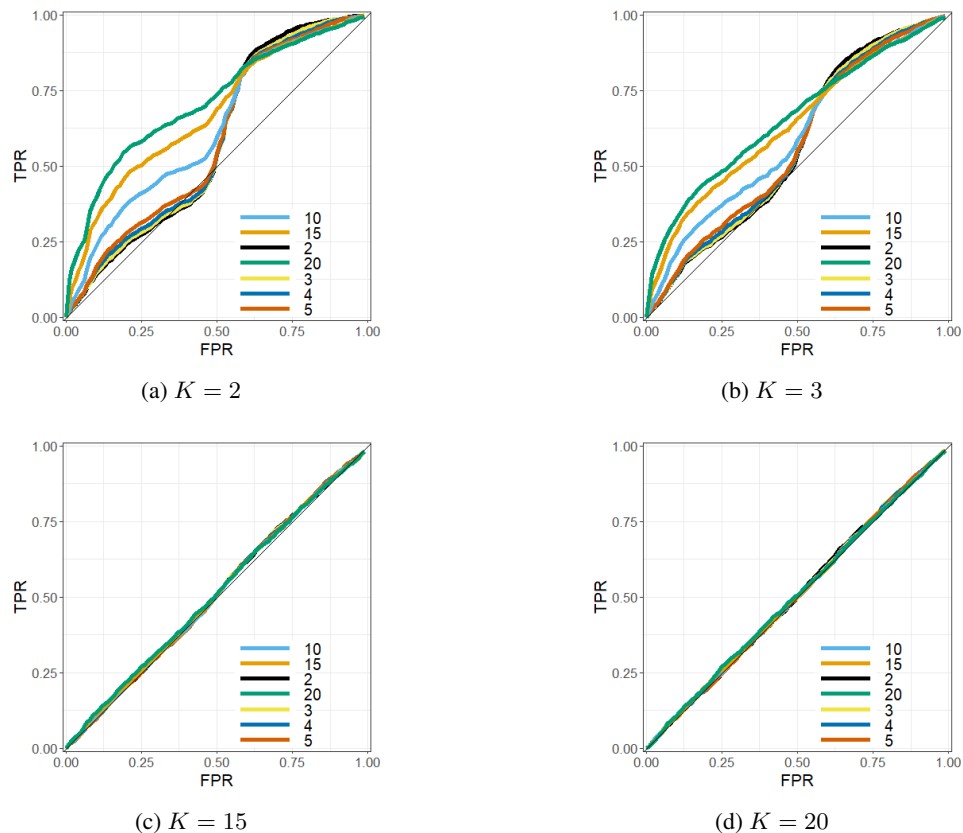

(a) $K = 2$

(b) $K = 3$

(c) $K = 15$

(d) $K = 20$

Figure 8: ROC curves for different choices of the number of log-moments in the experiment with a general $B$.

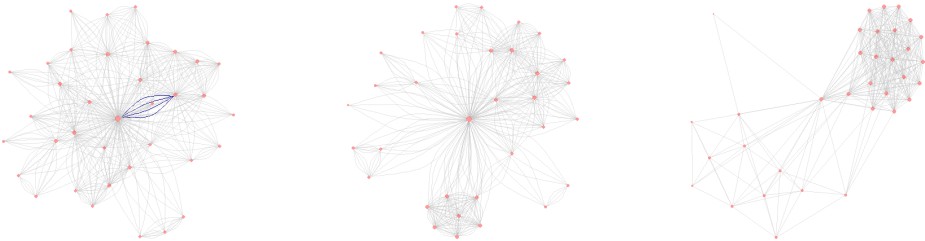

Figure 9: Sample graphs from the COLLAB dataset: High Energy Physics (left), Condensed Matter Physics (middle) and Astrophysics (right).

# E    Proofs auxiliary results

## E.1    Proof of Lemma B.2

Recall that the $L^1$ Wasserstein distance between the distributions of two random variables $Y$ and $Z$ can be expressed as

$$d_{W_1}(Y, Z) = \sup_{h:\ \|h\|_{\mathrm{Lip}} \leq 1} |E[h(Y)] - E[h(Z)]|$$

Table 4: Mean runtimes (ms) of `matching(B₁, B₂)` for selected values of $K$.

|  | $K = 2$ | $K = 10$ | $K = 20$ | $K = 50$ | $K = 100$ |
|---|---|---|---|---|---|
| Mean runtime (ms) | 0.6836 | 1.0366 | 1.3055 | 3.0415 | 8.3679 |

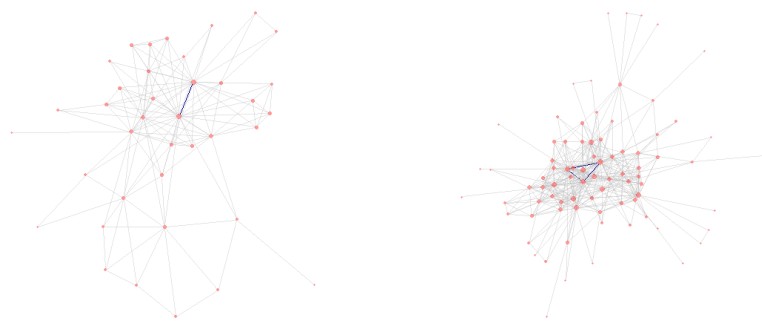

Figure 10: Sample graphs from the movie/television dataset: Class 1 (left) and Class 2 (right)

where $h$ ranges over all 1-Lipschitz functions $h : \mathbb{R} \to \mathbb{R}$, that is, $h$ that satisfy $|h(x)-h(y)| \leq |x-y|$ for all $x, y \in \mathbb{R}$. See [10, Chapter 4] or [33].

Lemma B.2 follows form the following results:

**Proposition E.1.** Let $S \sim \mathsf{Bin}(n, p)$ with $p \leq 1/2$ and let $W = \frac{\sqrt{n}}{\sigma}(\frac{S}{n} - p)$ where $\sigma = \sqrt{pq}$ and $q = 1 - p$. Let $C \leq 0.4785$ be the constant in the Berry-Esseen bound. Then,

$$\sup_{x \in \mathbb{R}} |P(W \leq x) - \Phi(x)| \leq \frac{C}{\sqrt{np/2}} \tag{51}$$

$$\sup_{h : \|h\|_{\mathrm{Lip}} \leq 1} \big| E[h(W)] - E[h(Z)] \big| \leq \frac{1}{\sqrt{np/2}} \tag{52}$$

where $Z \sim N(0, 1)$.

*Proof.* We can write $S = \sum_{i=1}^{n} B_i$ where $B_i$ are i.i.d. Bernoulli($p$) variables. Let $X_i = (B_i - p)/\sigma$ be the standardized versions and note that $W = \frac{1}{\sqrt{n}} \sum_{i=1}^{n} X_i$. Berry-Esseen bound gives

$$\sup_{x \in \mathbb{R}} |P(W \leq x) - \Phi(x)| \leq C \frac{E[|X_1|^3]}{\sqrt{n}}$$

and Corollary 4.1 of [10, page 67] gives $d_{W_1}(W, Z) \leq \frac{E[|X_1|^3]}{\sqrt{n}}$. We have

$$E[|X_1|^3] = \frac{1}{\sigma^3} \cdot E[|B_1 - p|^3] = \frac{p^3 q + q^3 p}{(pq)^{3/2}} = \frac{p^2 + q^2}{\sqrt{pq}} \leq \frac{(p+q)^2}{\sqrt{p/2}}$$

which gives the desired inequalities.

$\square$

**Lemma E.1.** Suppose that for $r \in \{1, 2\}$ and $t \in [N_r]$ the symmetric matrices $A_{rt} \in \{0, 1\}^{n_{rt} \times n_{rt}}$ are independent with independent entries on and above diagonal that satisfy $(A_{rt})_{ij} \sim \mathsf{Ber}(B_{(z_{rt})_i, (z_{rt})_j})$ where $z_{rt} \in [K]^{n_{rt}}$ is deterministic. Let

$$S_r = \sum_{t=1}^{N_r} \mathcal{S}(A_{rt}, z_{rt}), \quad m_r = \sum_{t=1}^{N_r} \mathcal{C}(A_{rt}, z_{rt}),$$

and for all $k, \ell \in [K]$, let $\sigma_{k\ell} = \sqrt{B_{k\ell}(1 - B_{k\ell})}$ and set

$$\xi_{k\ell} = \frac{\sqrt{\bar{m}_{k\ell}}}{\sqrt{2} \sigma_{k\ell}} \Big( \frac{(S_1)_{k\ell}}{m_{1k\ell}} - \frac{(S_2)_{k\ell}}{m_{2k\ell}} \Big) \tag{53}$$

where $\bar{m}_{k\ell}$ is the harmonic mean of $(m_1)_{k\ell}$ and $(m_2)_{k\ell}$. Assume that $m_{2k\ell} \leq c_1 m_{1k\ell}$ for some $c_1 > 0$. Then

$$\sup_{x \in \mathbb{R}} \big| P(\xi_{k\ell} \leq x) - P(Z \leq x) \big| \leq C \sqrt{\frac{n}{\nu_n}} \Big( \frac{1}{\sqrt{m_{1k\ell}}} + \frac{1}{\sqrt{m_{2k\ell}}} \Big).$$

where $C > 0$ is a constant dependent on $c_1$ and $Z \sim N(0, 1)$.

*Proof.* Let $\sigma_{kl} = \sqrt{B_{kl}(1 - B_{kl})}$. First, notice that by Proposition E.1, it holds that

$$\sup_{x \in \mathbb{R}} \left| P\left( \frac{\sqrt{m_{rkl}}}{\sigma_{kl}} \left( \frac{S_{rkl}}{m_{rkl}} - B_{kl} \right) \leq x \right) - \Phi(x) \right| \leq \frac{C}{\sqrt{m_{rkl}\nu_n/n}}$$

for some constant $C > 0$, and for $r = 1, 2$. Hence,

$$\sup_{x \in \mathbb{R}} \left| P\left( \frac{\sqrt{\bar{m}_{k\ell}}}{\sqrt{2}\,\sigma_{kl}} \cdot \frac{S_{rkl}}{m_{rkl}} \leq x \right) - \Phi\left( \sqrt{\frac{2m_{rkl}}{\bar{m}_{k\ell}}} \Big( x - \frac{\sqrt{\bar{m}_{k\ell}} B_{kl}}{\sqrt{2}\sigma_{kl}} \Big) \right) \right| \leq \frac{C}{\sqrt{m_{rkl}\nu_n/n}}. \tag{54}$$

Define the random variables

$$X_r = \frac{\sqrt{\bar{m}_{k\ell}}}{\sqrt{2}\,\sigma_{kl}} \cdot \frac{S_{rkl}}{m_{rkl}}, \quad r = 1, 2,$$

and consider two independent random variable $Y_1$ and $Y_2$ with

$$Y_r \sim N\left( \frac{\sqrt{\bar{m}_{k\ell}} B_{k\ell}}{\sqrt{2}\sigma_{k\ell}}, \frac{\bar{m}_{k\ell}}{2m_{rk\ell}} \right).$$

Notice that $Y_1 - Y_2 \sim N(0, 1)$. Let $F_Z$ denote the CDF of a random variable $Z$. Then, we can rewrite (54) as

$$\sup_{x \in \mathbb{R}} |F_{X_1}(x) - F_{Y_1}(x)| \leq \frac{C}{\sqrt{m_{rkl}\nu_n/n}}. \tag{55}$$

For any $x, x_2 \in \mathbb{R}$, we have

$$P(X_1 - X_2 \leq x \,|\, X_2 = x_2) = P(X_1 \leq x + x_2) = F_{X_1}(x + x_2),$$

by independence of $X_1$ and $X_2$. Fix $x \in \mathbb{R}$. Since $P(X_1 - X_2 \leq x) = E[P(X_1 - X_2 \leq x \,|\, X_2)]$, it follows that

$$
\begin{aligned}
|P(X_1 &- X_2 \leq x) - P(Y_1 - Y_2 \leq x)| \\
&= \left| E[F_{X_1}(x + X_2)] - E[F_{Y_1}(x + Y_2)] \right| \\
&= \left| E[F_{X_1}(x + X_2)] - E[F_{Y_1}(x + X_2)] + E[F_{Y_1}(x + X_2)] - E[F_{Y_1}(x + Y_2)] \right| \\
&\leq E\big[ \big| F_{X_1}(x + X_2) - F_{Y_1}(x + X_2) \big| \big] + \left| E[F_{Y_1}(x + X_2)] - E[F_{Y_1}(x + Y_2)] \right| \\
&\leq \frac{C}{\sqrt{m_{1kl}\nu_n/n}} + |E[h(X_2)] - E[h(Y_2)]|
\end{aligned}
$$

where we have defined $h(z) = F_{Y_1}(x + z)$. Since

$$
\begin{aligned}
\|h\|_{\mathrm{Lip}} &\leq \|h'\|_\infty = \sqrt{\frac{2m_{2kl}}{\bar{m}_{k\ell}}} \cdot \|\Phi'\|_\infty \\
&= \sqrt{1 + \frac{m_{2kl}}{m_{1kl}}} \cdot \frac{1}{\sqrt{2\pi}} \leq \sqrt{1 + c_1} \cdot \frac{1}{\sqrt{2\pi}} =: c_2
\end{aligned}
$$

by assumption. Then, from Proposition E.1, $|E[h(X_2)] - E[h(Y_2)]| \leq c_2/\sqrt{m_{2kl}\nu_n/n}$

$$|P(X_1 - X_2 \leq x) - P(Y_1 - Y_2 \leq x)| \leq \frac{C}{\sqrt{m_{1kl}\nu_n/n}} + \frac{c_2}{\sqrt{m_{2kl}\nu_n/n}}$$

which gives the desired result. $\qquad\square$

Let $\mathcal{I}(\mathbb{R})$ be the set of indicator functions of half-intervals, that is,

$$\mathcal{I}(\mathbb{R}) = \big\{ t \mapsto 1\{t \leq x\} : x \in \mathbb{R} \big\}.$$

**Lemma E.2.** Consider random variables $X_{ni}, i \in [K]$ and $Y_n$ and assume that $\{X_{ni}, i \in [K]\}$ are independent conditional on $Y_n$. In addition, we have

$$\sup_{h \in \mathcal{I}(\mathbb{R})} |E[h(X_{ni}) \,|\, Y_n] - E[h(z)]| \cdot 1\{Y_n \in \mathcal{A}_n\} \leq \varepsilon_n, \quad i \in [K]$$

for some sequence of events $\mathcal{A}_n$ and a deterministic sequence of $\varepsilon_n > 0$, and some random variable $Z \sim \mu$. Assume that $\varepsilon_n \to 0$ and $P(Y_n \in \mathcal{A}_n^c) \to 0$ as $n \to \infty$. Then

$$(X_{n1}, \ldots, X_{nK}) \Rightarrow \mu^{\otimes K}.$$

*Proof.* Let $Z_i, i \in [K]$ be i.i.d. draws from $\mu$. It is enough to show that

$$E\Big[\prod_{i=1}^{K} f_i(X_{ni})\Big] \to E\Big[\prod_{i=1}^{K} f_i(Z_i)\Big] = \prod_i E[f_i(Z_i)] \tag{56}$$

for any collection of $f_1, \ldots, f_K \in \mathcal{I}(\mathbb{R})$.

Let us fix one such collection and, for simplicity, write $W_n = \prod_i f_i(X_{ni})$ and $C = \prod_i E[f_i(Z_i)]$. We want to show $E[W_n] \to C$. From the assumption, it follows that

$$\big| E[f_i(X_{ni}) \,|\, Y_n] - E[f_i(Z_i)] \big| \le \varepsilon_n + 2 \cdot 1\{Y_n \in \mathcal{A}_n^c\}.$$

By conditional independence, $E[W_n \,|\, Y_n] = \prod_i E[f_i(X_{ni}) \,|\, Y_n]$. Then, using $|\prod_{i=1}^{K} a_i - \prod_{i=1}^{K} b_i| \le K \max_i |a_i - b_i|$, which holds if $a_i, b_i \in [-1, 1]$ for all $i$, we have

$$\big| E[W_n \,|\, Y_n] - C \big| \le K\varepsilon_n + 2K \cdot 1\{Y_n \in \mathcal{A}_n^c\}.$$

Then, we have

$$\begin{aligned}
|E[W_n] - C| &= |E[E[W_n \,|\, Y_n]] - C| \\
&\le E\big\{\big| E[W_n \,|\, Y_n]\big\} - C\big| \le K\varepsilon_n + 2K P(Y_n \in \mathcal{A}^c).
\end{aligned}$$

Letting $n \to \infty$, the desired result follows from the assumptions. $\qquad\square$

*Proof of Lemma B.2.* Let $z = (z_{rt}, t \in [N_r], r = 1, 2)$. Let $\mathcal{M}(c_1) = \{z : m_{2k\ell} \le c_1 m_{1k\ell}\}$. By Lemma E.1, we have

$$\sup_{h \in \mathcal{I}(\mathbb{R})} \big| E[h(\xi_{k\ell}) \,|\, z] - E[h(z)]\big| \cdot 1\{z \in \mathcal{M}(c_1)\} \le C(c_1)\sqrt{\frac{n}{\nu_n}}\Big(\frac{1}{\sqrt{m_{1k\ell}}} + \frac{1}{\sqrt{m_{2k\ell}}}\Big) \tag{57}$$

where $\xi_{k\ell}$ as defined in (53), $Z \sim N(0, 1)$ and $C(c_1)$ is some constant dependent on $c_1$. Let $n_{rtk} := \sum_{i=1}^{n_t} 1\{(z_{rt})_i = k\}$, and consider the event

$$\mathcal{A}_n := \Big\{\min_{r,k} n_{rtk} \ge n_t/\beta, \ \forall t \in [N]\Big\}.$$

Then on $\mathcal{A}_n$, we have $m_{rk\ell} \ge cNn^2/\beta^2$; see also (39). We also have $P(\mathcal{A}_n^c) = o(1)$ under the assumptions of Theorem A.6, by the same argument that controls $\mathcal{E}_1$ in Proposition F.4. Moreover, assuming $N_2 \le N_1$—without loss of generality—on $\mathcal{A}_n$, we have

$$\frac{m_{2k\ell}}{m_{1k\ell}} \le \frac{N_2(C_2 n)^2}{N_1 n^2/\beta^2} \le \beta^2 C_2^2$$

where we have used the size assumption (19). That is, $\mathcal{A}_n \subset \{z \in \mathcal{M}(\beta^2 C_2)\}$. Taking $c_1 = \beta^2 C_2^2$ in (57) and multiplying both sides of the inequality by $1_{\mathcal{A}_n}$, we obtain

$$\sup_{h \in \mathcal{I}(\mathbb{R})} \big| E[h(\xi_{k\ell}) \,|\, z] - E[h(z)]\big| \cdot 1_{\mathcal{A}_n} \le \frac{2\beta\, C(\beta^2 C_2)}{\sqrt{c}} \frac{1}{\sqrt{Nn\nu_n}}.$$

Since $\{\xi_{k\ell}\}$ are independent given $z$, the result now follows from Lemma E.2 given $Nn\nu_n \to \infty$. $\quad\square$

## E.2 Proof of Lemma B.1

For $P \in \Pi_K$, let

$$\begin{aligned}
J_1(P) &= \{i : P_{ii} \neq 0\}, \\
J_2(P) &= \{(i,j) : P_{ij} \neq 0, \ i \neq j\},
\end{aligned}$$

and note that $|J_1(P)| + |J_2(P)| = K$ for any $P \in \Pi_K$. By assumption $\|\Delta\|_\infty < 1/2$, and hence

$$\begin{aligned}
1 + \Delta_{ii} &> 1/2 \quad \text{for} \quad i \in J_1(P) \\
\Delta_{ij} &< 1/2 \quad \text{for} \quad (i,j) \in J_2(P).
\end{aligned}$$

We also have

$$\text{tr}\big(P^T(I+\Delta)\big) = \sum_{i,j}(1_{\{i=j\}}+\Delta_{ij})P_{ij}$$

$$= \sum_{i\in J_1(P)}(1+\Delta_{ii}) + \sum_{(i,j)\in J_2(P)}\Delta_{ij}.$$

Let $P\neq I$, so that both $J_2(P)$ and $[K]\setminus J_1(P)$ are nonempty. Then,

$$\text{tr}(I+\Delta) - \text{tr}\big(P^T(I+\Delta)\big) = \sum_{i\in[K]\setminus J_1(P)}(1+\Delta_{ii}) - \sum_{(i,j)\in J_2(P)}\Delta_{ij}$$

$$> \frac{1}{2}(K-|J_1(P)|) - \frac{1}{2}|J_2(P)| = 0,$$

showing that identity is the unique optimal solution.

### E.3 Randomization

Let $\mathcal{U}(\Pi_K)$ denote the uniform distribution on the set of permutation matrices $\Pi_K$.

**Lemma E.3** (Randomization)**.** Assume that there is a permutation matrix $C_{rt}=C_{rt}(A_{rt})$ potentially dependent on the adjacency matrix $A_{rt}$ such that

$$\widetilde{B}_{rt} \xrightarrow{C_{rt}} B_{rt}$$

where $\widetilde{B}_{rt} = \mathcal{S}(A_{rt},\widehat{z}_{rt}^{(0)}) \oslash \mathcal{C}(\widehat{z}_{rt}^{(0)})$. Let $\widehat{B}_{rt}$ be constructed as in step step 4. Then,

$$\widehat{B}_{rt} \xrightarrow{P_{rt}} B_{rt} \tag{58}$$

where $P_{rt}\sim\mathcal{U}(\Pi_K)$ independent of $A_{rt}$ (and hence $\widehat{B}_{rt}$).

*Proof.* Notice that by construction $\widehat{B}_{rt} = U_{rt}\widetilde{B}_{rt}U_{rt}^\top$ with $U_{rt}\sim\mathcal{U}(\Pi_K)$. Let $P_{rt}=C_{rt}U_{rt}^\top\in\Pi_K$. Then, $C_{rt}\widetilde{B}_{rt}C_{rt}^\top = P_{rt}\widehat{B}_{rt}P_{rt}^\top$ and (58) follows. It remains to show independence of $P_{rt}$ and its uniform distribution. Indeed, let $P_0\in\Pi_K$ and $A_0\in\{0,1\}^{n\times n}$. We have (showing the dependence of $C_{rt}$ on $A_{rt}$ explicitly)

$$P(P_{rt}=P_0, A_{rt}=A_0) = P(U_{rt}=P_0^\top C_{rt}(A_{rt}), A_{rt}=A_0)$$

$$= P(U_{rt}=P_0^\top C_{rt}(A_0), A_{rt}=A_0)$$

$$= P(U_{rt}=P_0^\top C_{rt}(A_0)) \cdot P(A_{rt}=A_0)$$

$$= \frac{1}{|\Pi_K|} \cdot P(A_{rt}=A_0)$$

where the third line is by the independence of $U_{rt}$ and $A_{rt}$ and the fourth line since $U_{rt}\sim\mathcal{U}(\Pi_K)$. The above shows that the joint distribution factorizes as uniform distribution for $P_{rt}$ and original (marginal) distribution for $A_{rt}$, which proves the claim. $\qquad\square$

## F Consistency of connectivity matrix estimation

Given an $n\times n$ adjacency matrix $A$, we apply a community detection algorithm to obtain label vector $\widehat{z}=(\widehat{z}_1,\ldots,\widehat{z}_n)\in[K]^n$. Let $d_{\text{NH}}$ be the normalized Hamming distance, that is, $d_{\text{NH}}(z,\widehat{z}) = d_{\text{H}}(z,\widehat{z})/n$ where $d_{\text{H}}(z,\widehat{z}) = \sum_{i=1}^n 1\{z_i\neq\widehat{z}_i\}$ is the Hamming distance.

**Remark F.1.** In the proof of consistency below, our results are given in terms of $\|E\|_{\max}$ where $E$ here is a $K\times K$ matrix placeholder for different matrices as given in Propositions F.3 and F.4. However, to simplify the proofs below, when computing upper bounds for $\|E\|_{\max} = \max_{k,\ell\in\{1,\ldots,K\}}|E_{k\ell}|$, we focus on the case $k\neq\ell$ and omit the case $k=\ell$ as it is similar. The only difference in dealing with $k=\ell$ is that the constants would need to be inflated.

We repeated use the following result, which follows for example from Bernstein's inequality:

**Proposition F.2.** Suppose that $S = \sum_{i=1}^{n} X_i$ where $X_i \sim \mathsf{Ber}(p_i)$ independently for $i \in [n]$. Assume that $\max_i p_i \leq p$. Then, for all $\delta \in [0, 1]$,

$$P\left(\widehat{p} - p \geq \delta p\right) \leq e^{-\delta^2 np/3},$$

and the same inequality holds for $P(p - \widehat{p} \geq \delta p)$.

*Proof.* By Bernstein inequality, using $|X_i - p_i| \leq 1$, for any $u > 0$, we have

$$P(S - E[S] \geq nu) \leq \exp\left(-\frac{nu^2}{2\bar{\sigma}^2 + 2u/3}\right)$$

where $\bar{\sigma}^2 = \frac{1}{n}\sum_{i=1}^{n} \mathrm{var}(X_i)$. We have $\bar{\sigma}^2 \leq p$ and $E[S] \leq np$. Setting $u = \delta p$, we have

$$P(S - np \geq \delta np) \leq \exp\left(-\frac{n\delta^2 p^2}{2p + 2\delta p/3}\right)$$

$$= \exp\left(-\frac{\delta^2}{2 + 2\delta/3} np\right) \leq \exp(-3\delta^2 np/8)$$

where the last inequality uses $\delta \leq 1$. Replacing $3/8$ with $1/3$ gives a further upper bound. $\square$

Let us write $(A, z) \sim \mathsf{SBM}_n(B, \pi)$ for an $n$-node draw from a Bayesian SBM, with connectivity $B$ and class prior $\pi$.

**Proposition F.3.** Assume that $(A, z) \sim \mathsf{SBM}_n(B, \pi)$ with $B$ satisfying (18). Let $S = \mathcal{S}(A, z)$, $\widehat{S} = \mathcal{S}(A, \widehat{z})$ and $m = \mathcal{C}(z)$, $\widehat{m} = \mathcal{C}(\widehat{z})$. Set

$$\widehat{B} = \mathcal{B}(A, \widehat{z}) = \widehat{S}/\widehat{m}, \quad \widetilde{B} = \mathcal{B}(A, z) = S/m.$$

Fix $\kappa \in (0, 1)$ and $\alpha > 0$, and let $\beta = 1/(\bar{\kappa}\pi_{\min})$ with $\bar{\kappa} = 1 - \kappa$, and define

$$\delta_n := \sqrt{\frac{3\beta^2 \alpha \log n}{n\nu_n}}.$$

Assume that $\delta_n \leq 1$ and $\nu_n \geq 3(1 + \alpha)\log n$. For $\sigma \in \Pi_K$, let $\varepsilon(\sigma) = d_{\mathrm{NH}}(z, \sigma \circ \widehat{z})$. Then, with probability at least $1 - 3K^2 n^{-\alpha} - Ke^{-\kappa^2 n\pi_{\min}/3}$, for all $\sigma \in \Pi_K$ such that $12\beta^2\varepsilon(\sigma) \leq 1$,

$$\|P_\sigma \widehat{B} P_\sigma^T - \widetilde{B}\|_{\max} \leq 56C_1\beta^2 \cdot \frac{\nu_n}{n}\varepsilon(\sigma), \tag{59}$$

$$\|\widetilde{B} - B\|_{\max} \leq C_1 \frac{\nu_n}{n}\delta_n, \tag{60}$$

$$\|S - P_\sigma \widehat{S} P_\sigma^T\|_{\max} \leq 8C_1\beta \cdot n\nu_n\varepsilon(\sigma), \tag{61}$$

$$\|m - P_\sigma \widehat{m} P_\sigma^T\|_{\max} \leq 6\beta \cdot n^2\varepsilon(\sigma), \tag{62}$$

$$\min_{k,\ell} m_{k\ell} \geq n^2/\beta^2. \tag{63}$$

*of Proposition F.3.* By redefining $\widehat{z}$ to be $\sigma \circ \widehat{z}$, one gets $P_\sigma \widehat{B} P_\sigma^T$ in place of $\widehat{B}$. Thus, without loss of generality, we can assume $\sigma$ to be the identity permutation. Let

$$d_k(z, \widehat{z}) = \sum_i \big|1\{z_i = k\} - 1\{\widehat{z}_i = k\}\big|, \quad d_H(z, \widehat{z}) = \sum_i 1\{z_i \neq \widehat{z}_i\}$$

so that $\sum_k d_k(z, \widehat{z}) = 2d_H(z, \widehat{z})$. This can verified by writing

$$\big|1\{z_i = k\} - 1\{\widehat{z}_i = k\}\big| = 1\{z_i = k, \widehat{z}_i \neq k\} + 1\{z_i \neq k, \widehat{z}_i = k\}.$$

Let $n_k = \sum_{i=1}^{n} 1\{z_i = k\}$ and $\widehat{n}_k = \sum_{i=1}^{n} 1\{\widehat{z}_i = k\}$ and note that $|n_k - \widehat{n}_k| \leq d_k(z, \widehat{z})$. Let us also write

$$\rho := \max_k \frac{d_k(z, \widehat{z})}{n_k} \tag{64}$$

and note that $|(\widehat{n}_k/n_k) - 1| \leq \rho$.

Let us consider some events. First, consider event

$$\mathcal{E}_0 := \Big\{ \max_j \sum_i A_{ij} \leq 2C_1\nu_n \Big\}. \tag{65}$$

Applying Proposition F.2 conditioned on $z$, with $p = C_1\nu_n/n$, and then taking expectation of both sides to remove the conditioning, we have

$$P\Big( \sum_i A_{ij} \geq C_1\nu_n(1+u) \Big) \leq e^{-u^2 C_1\nu_n/3}.$$

Take $u = 1/\sqrt{C_1}$. Then, $P(\mathcal{E}_0^c) \leq n^{-\alpha}$ by union bound and assumption $\nu_n/3 \geq (1+\alpha)\log n$.

Next, note that $P(n_k \leq (1-\kappa)n\pi_k) \leq \exp(-\kappa^2 n\pi_k/3)$ for $\kappa \in (0,1)$, by Proposition F.2. Consider the event

$$\mathcal{E}_1 := \Big\{ \min_{k=1,\dots K} n_k \geq n/\beta \Big\},$$

where $1/\beta = (1-\kappa)\pi_{\min}$. Then, by union bound $P(\mathcal{E}_1^c) \leq K\exp(-\kappa^2 n\pi_{\min}/3)$. On $\mathcal{E}_1$,

$$\rho := \max_k \frac{d_k(z,\widehat{z})}{n_k} \leq \frac{2d_H(z,\widehat{z})}{\bar{\kappa}\pi_{\min}n} = 2\beta d_{\mathrm{NH}}(z,\widehat{z}). \tag{66}$$

Next, we apply Proposition F.2 conditioned on $z$, to get

$$P\big( |B_{k\ell} - \widetilde{B}_{k\ell}| \geq \delta_n B_{k\ell} \mid z \big) \cdot 1_{\mathcal{E}_1} \;\leq\; 2e^{-\delta_n^2 n_k n_\ell B_{k\ell}/3} \cdot 1_{\mathcal{E}_1}$$
$$\leq\; 2e^{-\delta_n^2(n^2/\beta^2)(\nu_n/n)/3}$$

where we have used the lower bound in (18). Take $\delta_n^2 = 3\beta^2\alpha \log n/(n\nu_n)$ and let

$$\mathcal{E}_2 = \Big\{ |B_{k\ell} - \widetilde{B}_{k\ell}| \leq \delta_n B_{k\ell} \text{ for all } k,\ell \in [K] \Big\}. \tag{67}$$

Then, by union bound, $P(\mathcal{E}_1 \cap \mathcal{E}_2^c) \leq P(\mathcal{E}_2^c \mid \mathcal{E}_1) \leq 2K^2 n^{-\alpha}$. We will work on $\mathcal{E}_0 \cap \mathcal{E}_1 \cap \mathcal{E}_2$ and note that

$$P(\mathcal{E}_0 \cap \mathcal{E}_1 \cap \mathcal{E}_2) \geq 1 - P(\mathcal{E}_0^c) - P(\mathcal{E}_1^c) - P(\mathcal{E}_1 \cap \mathcal{E}_2^c).$$

Note that on $\mathcal{E}_2$,

$$\max_{k,\ell} |B_{k\ell} - \widetilde{B}_{k\ell}| \leq \delta_n \cdot C_1\nu_n/n, \tag{68}$$

$$\max_{k,\ell} \widetilde{B}_{k\ell} \leq 2C_1\nu_n/n \tag{69}$$

using the upper bound in (18) and assumption $\delta_n \leq 1$. This proves (60).

Now, let us establish (61). We have

$$|S_{k\ell} - \widehat{S}_{k\ell}| \leq \sum_{i,j} A_{ij}|1\{z_i = k, z_j = \ell\} - 1\{\widehat{z}_i = k, \widehat{z}_j = \ell\}|.$$

Adding and subtracting $1\{\widehat{z}_i = k, z_j = \ell\}$ and expanding by triangle inequality, we get $|S_{k\ell} - \widehat{S}_{k\ell}| \leq T_{21} + T_{22}$ where

$$T_{21} := \sum_{i,j} A_{ij}|1\{z_i = k, z_j = \ell\} - 1\{\widehat{z}_i = k, z_j = \ell\}|,$$

$$T_{22} := \sum_{i,j} A_{ij}|1\{\widehat{z}_i = k, \widehat{z}_j = \ell\} - 1\{\widehat{z}_i = k, z_j = \ell\}|.$$

Consider $T_{22}$ first. We have

$$T_{22} = \sum_j \Big( \sum_i A_{ij} 1\{\widehat{z}_i = k\} \cdot |1\{\widehat{z}_j = \ell\} - 1\{z_j = \ell\}| \Big)$$
$$\leq \sum_j \Big( \sum_i A_{ij} \cdot |1\{\widehat{z}_j = \ell\} - 1\{z_j = \ell\}| \Big)$$
$$\leq 2C_1\nu_n \cdot d_\ell(z,\widehat{z})$$

on $\mathcal{E}_0$. Similarly, $T_{21} \le 2C_1\nu_n \cdot d_k(z, \widehat{z})$ on $\mathcal{E}_0$. Then,

$$|S_{k\ell} - \widehat{S}_{k\ell}| \le 2C_1\nu_n \cdot (n_\ell + n_k)\rho \le 4C_1 n\nu_n\rho \tag{70}$$

using $n_k \le n$ for all $k$. Combined with (66), this proves (61).

Let us define $\widetilde{B}' = \widehat{S}/m$. For $k \ne \ell$, we have $m_{k\ell} = n_k n_\ell$. Then, on $\mathcal{E}_1$,

$$|\widetilde{B}_{k\ell} - \widetilde{B}'_{k\ell}| = \frac{|S_{k\ell} - \widehat{S}_{k\ell}|}{n_k n_\ell} \le 2C_1\nu_n \cdot (n_k^{-1} + n_\ell^{-1})\rho \le 4C_1\beta\frac{\nu_n}{n}\rho$$

using $n_k \ge n/\beta$ for all $k$. By (69), on $\mathcal{E}_2$, we also have

$$\widetilde{B}'_{k\ell} \le 2C_1\frac{\nu_n}{n}(1 + 2\beta\rho) \le 4C_1\frac{\nu_n}{n}.$$

assuming $2\beta\rho \le 1$. Next we note that

$$\left|\frac{\widehat{m}_{k\ell}}{m_{k\ell}} - 1\right| = \left|\frac{\widehat{n}_k\widehat{n}_\ell}{n_k n_\ell} - 1\right| \le 3\rho.$$

Assuming that $3\rho \le 1/2$, letting $x = \widehat{m}_{k\ell}/m_{k\ell}$, we have $|1 - x| \le 3\rho \le 1/2$, hence $|1 - x^{-1}| \le 6\rho$. It follows that on $\mathcal{E}_2$

$$|\widetilde{B}'_{k\ell} - \widehat{B}_{k\ell}| = \frac{\widehat{S}_{k\ell}}{m_{k\ell}}\left|1 - \frac{m_{k\ell}}{\widehat{m}_{k\ell}}\right| \le \widetilde{B}'_{k\ell} \cdot 6\rho \le 24C_1\frac{\nu_n}{n}\rho.$$

By triangle inequality

$$\begin{aligned} \|\widehat{B} - \widetilde{B}\|_{\max} &\le \|\widehat{B} - \widetilde{B}'\|_{\max} + \|\widetilde{B}' - \widetilde{B}\|_{\max} \\ &\le 24C_1\frac{\nu_n}{n}\rho + 4C_1\beta\frac{\nu_n}{n}\rho \\ &\le 28C_1\frac{\nu_n}{n}\beta\rho \end{aligned}$$

using $\beta \ge 1$ (we are weakening the constants in favor of a simpler expression).

For (62), we have $|[m - \widehat{m}]_{k\ell}| \le n_k n_\ell|\frac{\widehat{n}_k\widehat{n}_\ell}{n_k n_\ell} - 1| \le n^2 3\rho$. Putting the pieces together combined with (66) proves the claim. $\qquad\square$

### F.1 Multi-network extension

**Proposition F.4.** Assume that $(A_t, z_t) \sim \mathsf{SBM}_{n_t}(B, \pi_t)$ for $t \in [N]$, where $B$ satisfies (18) and $n_t$ satisfy

$$n \le n_t \le C_2 n. \tag{71}$$

Let $S_t = \mathcal{S}(A_t, z_t)$, $\widehat{S}_t = \mathcal{S}(A_t, \widehat{z}_t)$ and $m_t = \mathcal{C}(z_t)$, $\widehat{m}_t = \mathcal{C}(\widehat{z}_t)$. For $\sigma = (\sigma_t) \in \Pi_K^{\otimes N}$, set

$$\widehat{B}(\sigma) := \frac{\sum_t P_{\sigma_t}\widehat{S}_t P_{\sigma_t}^T}{\sum_t P_{\sigma_t}\widehat{m}_t P_{\sigma_t}^T}, \quad \widetilde{B} := \frac{\sum_t S_t}{\sum_t m_t}, \quad \varepsilon_t(\sigma) := d_{\mathrm{NH}}(z_t, \sigma_t \circ \widehat{z}_t)$$

where we interpret the division of matrices as elementwise.

Fix $\kappa \in (0, 1)$ and $\alpha > 0$, let $\beta = 1/(\bar{\kappa}\pi_{\min})$ with $\bar{\kappa} = 1 - \kappa$, and define

$$\delta_n := \sqrt{\frac{3\beta^2\alpha\log n}{Nn\nu_n}}.$$

Assume that $\delta_n \le 1$ and $\nu_n \ge 3(1+\alpha)\log n$. Then, with probability at least $1 - (C_2 N + 2K^2)n^{-\alpha} - NKe^{-\kappa^2 n\pi_{\min}/3}$, we have, for all $\sigma = (\sigma_t) \in \Pi_K^{\otimes N}$ for which $\max_t \varepsilon_t(\sigma) \le 1/(2\beta^3)$,

$$\|\widehat{B}(\sigma) - \widetilde{B}\|_{\max} \le 40\, C_1\beta^3\frac{\nu_n}{n}\varepsilon, \tag{72}$$

$$\|\widetilde{B} - B\|_{\max} \le C_1\frac{\nu_n}{n}\delta_n, \tag{73}$$

$$\|S_t - P_{\sigma_t}\widehat{S}_t P_{\sigma_t}^T\|_{\max} \le 8C_1\beta \cdot \nu_n n_t\,\varepsilon_t(\sigma), \tag{74}$$

$$\|m_t - P_{\sigma_t}\widehat{m}_t P_{\sigma_t}^T\|_{\max} \le 6\beta \cdot n_t^2\,\varepsilon_t(\sigma), \tag{75}$$

$$\min_{k,\ell}[m_t]_{k\ell} \ge n_t^2/\beta^2, \quad \text{for all } t \in [N] \tag{76}$$

where $\beta = 1/(\bar{\kappa}\pi_{\min})$ with $\bar{\kappa} = 1 - \kappa$.

Note that $n_t$ here is deterministic (size of the $t$-th network) and different from $n_k$ in the proof of Proposition F.3.

*Proof.* By redefining $\hat{z}_t$ to be $\sigma_t(\hat{z}_t)$, we can assume, without loss of generality, that $\sigma_t$ is the identity permutation (and hence $P_{\sigma_t} = I_K$) for all $t$. Note that none of the events we consider below depend on $\sigma$, hence the result indeed holds for all $\sigma$ simultaneously.

First, consider event

$$\mathcal{E}_0 := \Big\{ \max_{t \in [N], \, j \in [n_t]} \sum_{i=1}^{n_t} [A_t]_{ij} \leq 2C_1 C_2 \nu_n \Big\}. \tag{77}$$

Applying Proposition F.2 conditioned on $z_t$, with $p = C_1 \nu_n/n$ and $n_t p \leq C_1 C_2 \nu_n$, and then taking expectation of both sides to remove the conditioning, we have

$$P\Big(\sum_{i=1}^{n_t} [A_t]_{ij} \geq C_1 C_2 \nu_n (1+u)\Big) \leq e^{-u^2 C_1 C_2 \nu_n/3}.$$

Take $u = 1/\sqrt{C_1 C_2} \leq 1$. Then, by union bound and assumption $\nu_n/3 \geq (1+\alpha)\log n$, we have $P(\mathcal{E}_0^c) \leq C_2 N n^{-\alpha}$.

Next, let $n_{tk} := \sum_{i=1}^{n_t} 1\{(z_t)_i = k\}$, the (true) number of nodes in community $k$ in network $t$. By Proposition F.2, we have $P(n_{tk} \leq \bar{\kappa} n_t \pi_{tk}) \leq \exp(-\kappa^2 n_t \pi_{tk}/3)$ for $\kappa \in (0,1)$. Consider the event defined as

$$\mathcal{E}_1 := \Big\{ \min_{k=1,\ldots K} n_{tk} \geq n_t/\beta, \; \forall t \in [N] \Big\},$$

where $\beta = 1/(\bar{\kappa}\pi_{\min})$ with $\bar{\kappa} = 1 - \kappa$. Then, by union bound and using $n_t \geq n$,

$$P(\mathcal{E}_1^c) \leq NK \exp(-\kappa^2 n \pi_{\min}/3).$$

By the same argument as in the proof of Proposition F.3, and letting $\varepsilon_t = d_{\mathrm{NH}}(z_t, \hat{z}_t)$, we have on $\mathcal{E}_0 \cap \mathcal{E}_1$,

$$\|S_t - \hat{S}_t\|_{\max} \leq 8C_1 \beta \nu_n n_t \varepsilon_t, \quad \|m_t - \hat{m}_t\|_{\max} \leq 6\beta n_t^2 \varepsilon_t, \quad [m_t]_{k\ell} \geq n_t^2/\beta^2. \tag{78}$$

Let us now control $\widetilde{B} = (\sum_t S_t)/(\sum_t m_t)$. Conditioned on $z_t$, we have

$$[\textstyle\sum_t S_t]_{k\ell} \sim \mathsf{Bin}\big([x \textstyle\sum_t m_t]_{k\ell}, B_{k\ell}\big).$$

Also note that, for $k \neq \ell$, we have $[\sum_t m_t]_{k\ell} = \sum_t n_{tk} n_{t\ell} \geq \sum_t n_t^2/\beta^2 \geq Nn^2/\beta^2$ on $\mathcal{E}_1$. Then, applying Proposition F.2 conditioned on $z_t$, we get

$$
\begin{aligned}
P\big(|B_{k\ell} - \widetilde{B}_{k\ell}| \geq \delta_n B_{k\ell} \mid z_t\big) \cdot 1_{\mathcal{E}_1} &\leq 2e^{-\delta_n^2 [\sum_t m_t]_{k\ell} B_{k\ell}/3} \cdot 1_{\mathcal{E}_1} \\
&\leq 2e^{-\delta_n^2 (Nn^2/\beta^2)(\nu_n/n)/3}
\end{aligned}
$$

where we have used the lower bound in (18). Take $\delta_n^2 = 3\beta^2 \alpha \log n/(Nn\nu_n)$ and let

$$\mathcal{E}_2 = \Big\{ |B_{k\ell} - \widetilde{B}_{k\ell}| \leq \delta_n B_{k\ell} \; \text{ for all } k, \ell \in [K] \Big\}. \tag{79}$$

Then, by union bound, $P(\mathcal{E}_1 \cap \mathcal{E}_2^c) \leq P(\mathcal{E}_2^c \mid \mathcal{E}_1) \leq 2K^2 n^{-\alpha}$. We will work on $\mathcal{E}_0 \cap \mathcal{E}_1 \cap \mathcal{E}_2$ and note that

$$P(\mathcal{E}_0 \cap \mathcal{E}_1 \cap \mathcal{E}_2) \geq 1 - P(\mathcal{E}_0^c) - P(\mathcal{E}_1^c) - P(\mathcal{E}_1 \cap \mathcal{E}_2^c).$$

Note that on $\mathcal{E}_2$,

$$\max_{k,\ell} |B_{k\ell} - \widetilde{B}_{k\ell}| \leq \delta_n \cdot C_1 \nu_n/n, \tag{80}$$

$$\max_{k,\ell} \widetilde{B}_{k\ell} \leq 2C_1 \nu_n/n \tag{81}$$

using the upper bound in (18) and assumption $\delta_n \leq 1$. This proves (60).

Next we control the deviation $\widehat{B} - \widetilde{B}$. Treating division (and taking absolute values) of matrices as element-wise,

$$\frac{\sum_t \widehat{S}_t}{\sum_t \widehat{m}_t} = \frac{\sum_t S_t + \sum_t (\widehat{S}_t - S_t)}{\sum_t m_t + \sum_t (\widehat{m}_t - m_t)} =: \frac{a + b}{c + d}.$$

We then have

$$\left| \frac{a+b}{c+d} - \frac{a}{c} \right| = \left| \frac{(a/c) + (b/c)}{1 + (d/c)} - (a/c) \right| = \left| \frac{(b/c) - (a/c)(d/c)}{1 + (d/c)} \right| \leq \frac{|(b/c) - (a/c)(d/c)|}{1 - |d/c|}.$$

Let $\varepsilon = \max_t \varepsilon_t$. Then, we have

$$|[b/c]_{k\ell}| \leq \frac{\sum_t |[\widehat{S}_t - S_t]_{k\ell}|}{\sum_t [m_t]_{k\ell}} \overset{(a)}{\leq} 8C_1 \beta^3 \nu_n \frac{\sum_t n_t \varepsilon_t}{\sum_t n_t^2}$$

$$\overset{(b)}{\leq} 8C_1 \beta^3 \frac{\nu_n}{n} \frac{\sum_t n_t \varepsilon_t}{\sum_t n_t} \leq 8C_1 \beta^3 \frac{\nu_n}{n} \varepsilon$$

where (a) is by (78) and (b) uses assumption $n_t \geq n$. Similarly, we have $|[d/c]_{k\ell}| \leq 6\beta^3 \varepsilon$. Note that $a/c = \widetilde{B}$ hence elementwise in $[0, 2C_1 \nu_n/n]$ on $\mathcal{E}_2$. Assuming that $\beta^3 \varepsilon \leq 1/2$, we have,

$$\frac{|(b/c) - (a/c)(d/c)|}{1 - |d/c|} \leq \frac{|b/c| + (a/c)|d/c|}{1 - \frac{1}{2}} \leq 40\, C_1 \beta^3 \frac{\nu_n}{n} \varepsilon.$$

(where the final inequality means every element of the LHS is $\leq$ RHS) . That is, we have shown

$$\left| \frac{\sum_t \widehat{S}_t}{\sum_t \widehat{m}_t} - \frac{\sum_t S_t}{\sum_t m_t} \right| \leq 40\, C_1 \beta^3 \frac{\nu_n}{n} \varepsilon.$$

This proves (72) and finishes the proof. □

