# OpenReview forum: "Network two-sample test for block models"
_NeurIPS.cc/2025/Conference — NeurIPS 2025 poster_

### Official Review · Reviewer_oPrG · 2025-06-16

**Clarity:** 2
**Significance:** 2
**Originality:** 3
**Rating:** 4
**Confidence:** 4

**Summary:**

The paper proposes a hypothesis testing approach to determine whether two sequences of graphs are generated from an SBM with the same set of parameters. The approach consists of two separate modules: a spectral-based component to address the lack of labels, and a component of a statistical test. In general, the main ideas are clearly presented, which makes the approach convincing.

**Questions:**

See "Strengths And Weaknesses".

**Ethical Concerns:**

["NO or VERY MINOR ethics concerns only"]

**Final Justification:**

The authors have resolved most of my concerns in their rebuttal, and hence, I raise my score to 4. The paper offers a solid theoretical foundation, and I look forward to seeing its practical value demonstrated in future work.

**Limitations:**

No. There are some discussions of limitations in "Conclusion", however, the discussion is not thorough.

**Quality:**

3

**Strengths And Weaknesses:**

The paper is well-organized, and key ideas are well-explained. Sufficient theoretical justifications are provided, and thus the proposed approach is convincing. However, I still have a few concerns.
1. Motivation: I think the authors should give a more convincing argument regarding why the problem itself (including the assumptions, such as using SBM to generate the graph sequences) is important in practice. Though the paper has studied the real dataset COLLAB, it serves to verify the effectiveness of the algorithm. It is unclear to me the practical use of testing whether two sequences of graphs from COLLAB are from the same distribution. The authors are not required to perform any additional numerical study, but they should describe scenarios where the problem is important.
2. SBM: The paper focuses on graphs generated from SBM. It is indeed true that SBM can approximate graphs sampled from graphons, but provided the number of blocks becomes arbitrarily large. Therefore, I think considering SBM is somehow restricted. Please discuss.
3. Spectral-based matching: Spectral-based matching is not a new idea (e.g., [1-3]), though it may not have been directly applied to the problem studied in the paper. The authors should thoroughly discuss existing works on spectral-based graph matching.
4. Theoretical results: I think the theoretical results are not particularly instructive in their current form. For example, quantities such as $\gamma_n, \xi_r$ seem to come from nowhere and are hard to understand. It is unclear how to verify the validity of the complicated conditions of the theorems, and assess whether those numerical bounds are useful or not. The $(\theta, \eta)$-friendly condition in Definition 2.1 should be a key condition. However, its motivation is unclear, and how should one verify the condition given a sequence of graphs?
5. Proofreading: It seems that the authors haven't carefully proofread the paper. For example, eq. (60) in line 169 and eq. (59) in line 205 are missing. In line 230 (Remark 4.1), "more details" are not in C.3 (seems they are in D).

Based on the above, I give a Rating of 3. However, I may change my assessment and the rating based on the rebuttal.

[1] https://arxiv.org/pdf/1602.04181

[2] http://www.stat.yale.edu/~yw562/teaching/684/GM-grampa.pdf  (presentation slides)

[3] https://arxiv.org/pdf/2306.15747

---

> ### Author Rebuttal · Authors · 2025-07-31
>
> > 1. Motivation: ...
>
> The use of the COLLAB data was for illustration and mainly due to its having a clear "ground truth." We have another example in the supplement (a movie dataset which perhaps bears a similar criticism). The actual two-sample testing problem is widely applicable across the sciences, as we outline in the global response. In choosing the datasets, we were constrained by the limited availability of data with ground truth.
>
> Regarding the SBM, as you point out, SBMs can approximate general graphons, and that is perhaps a good enough justification. We have provided a thorough defense of SBM, and the idea of bin-and-test, in response to Reviewer EKux, which we avoid repeating due to the space constraint.
>
> As for concrete scenarios, below we provide a long list where two-sample network testing appears.
>
> ---
>
> ## Concrete applications of vertex-agnostic two-sample network tests
> *(Most reduce to $N_1=N_2=1$: one large graph vs another, no common labels.)*
>
> - **Clinical connectomics.** Compare functional or structural brain graphs of patients (e.g., Alzheimer’s, ASD) with healthy controls. Individual-specific parcellations make node correspondence impossible; detecting a distributional shift quantifies disease-related rewiring. *Bullmore & Sporns 2009; Zalesky, Fornito & Bullmore 2012*
>
> - **Protein / gene interaction.** Condition-specific interactomes—healthy vs tumour tissue, or two species—share only a subset of molecules. A label-free test spots pathway-level dysregulation or evolutionary divergence. *Menche et al. 2015; Pavlopoulos et al. 2011*
>
> - **Single-cell omics trajectories.** Cell–cell similarity graphs built separately for treatment and control arms trace developmental manifolds. Each run samples different cells, so matching is futile; the test asks whether the underlying trajectories differ. *Saelens, Cannoodt & Saeys 2019; Townes & Irizarry 2020*
>
> - **Ecological food-webs.** Predator–prey networks from protected reefs vs disturbed sites share few species due to turnover. Testing highlights ecosystem change beyond simple diversity counts. *Tylianakis & Morris 2017; Mougi & Kondoh 2012*
>
> - **Inter-bank lending.** Daily transaction graphs before vs after a policy shock (or calm vs crisis) have banks entering, exiting, or anonymised. Detecting a distributional shift is crucial for systemic-risk surveillance. *Bargigli et al. 2015; Squartini, van Lelyveld & Garlaschelli 2013*
>
> - **Online information diffusion.** Retweet graphs of a suspected misinformation campaign are compared with baseline news spread. Disposable or hashed accounts preclude alignment; a two-sample test flags coordinated manipulation. *Vosoughi, Roy & Aral 2018; Yang, Ferrara & Menczer 2021*
>
> - **Cyber-security flow graphs.** Current network-flow graph vs historical baseline under dynamic IPs and NAT. Vertex-free testing raises real-time intrusion alarms. *Heard et al. 2010; Irofti, Pătrașcu & Hîji 2022*
>
> - **Development-econ field trials.** Social networks from treatment vs control villages are disjoint. A two-sample test quantifies how the intervention reshaped interaction patterns. *Paluck, Shepherd & Aronow 2016; Cai, de Janvry & Sadoulet 2021*
>
> These diverse cases show that deciding whether two network samples share the same generating law is a core statistical task across science, finance, security, and policy.
>
> ----
>
> > 2. SBM: ...
>
> SBMs indeed can approximate graphs sampled from nearly arbitrary $L^2$ graphons by increasing the number of blocks. The key is that we **can allow** for an arbitrarily large number of blocks. There is no hard constraint on $K$ in our results (as we outline below). The only implicit condition on $K$ is what is required for the consistency of community detection algorithms, which imposes a very mild condition.
>
> We will clarify this in our response to your comment on simplifying the conditions.
>
> We also note that we have conducted experiments on graphon-based and random dot product graph (RDPG) models. These results demonstrate that our test remains robust and performs well even when the data are generated from more general distributions, suggesting that our method has strong potential for broader applicability beyond the SBM setting.
>
> > 3. Spectral-based matching: ...
>
> -   There is indeed a significant literature on spectral matching, and we did discuss several closely related works in this line in Section 1.2 [1, 11, 12, 20, 22, 30, 32, 33]. Thanks for pointing to additional references; we will add them to the discussion.
> -   It is worth noting that despite the vast literature, very few papers prove statistical consistency of these algorithms under a stochastic graph model, a contribution which we make to this literature.
>
> > Theoretical results ...
>
> Thank you for your suggestion. We agree and plan to simplify the conditions, in particular to emphasize that the conditions are not restrictive and that we can allow an arbitrarily large $K$. What we require is very mild. The presence of $\gamma_n$ and $\delta_n^{(N_r)}$ was due to the desire to preserve the finite-sample nature of our results and to aid in reading the proof, which we now realize is perhaps distracting.
>
> For example, the second term in $\gamma_n$ always goes to zero, so instead of $\gamma_n = o_p(1)$, we just need to state that $\widetilde{\varepsilon} = o_p(1)$, i.e., the misclassification rate goes to zero. As for $\delta_n^{(N_r)}$, we apologize for its being missing from the main text; it is
> $$
> \delta_n^{(N_r)} = \sqrt{\frac{3 \beta^2 \alpha \log n}{N_r n \nu_n}},
> $$
> a quantity essentially the same as the second term in $\gamma_n$ which always goes to zero (no condition needed). We always have $N n \nu_n$ going to infinity at least as fast as $n$, since $N \ge 1$ and $\nu_n = \Omega(1)$.
>
> Below we outline in detail how we can greatly simplify the conditions.
>
> ---
>
> ## Our conditions are really mild
>
> Per suggestions by multiple reviewers, we will simplify our conditions as outlined below. This allows us to reduce then two essentially two or three mild conditions in each case.
>
> Here is how we can simplify the conditions for theorem 2. Assume $B = (\nu_n / n) B^0$ for a fixed $B^0$. Then the conditions for theorem 2 are:
> 1.  $B^0$ has distinct eigenvalues and no eigenvector of it sums to zero.
> 2.  $N = O(n^\alpha)$ for some $\alpha \ge 0$.
> 3.  $\sqrt{\nu_n n^{1+\alpha}} \cdot \widetilde{\varepsilon} = o_p(1)$.
> 4.  $\widetilde{\varepsilon} \le C K^{-3/2}$ for some constant $C = C(B^0)$ that only depends on $B^0$.
>
> Condition 1 is equivalent to $B^0$ being $(\theta^0,\eta^0)$-friendly for some values of these parameters, but is clearer now. Condition 2 ensures all matchings are recovered and just requires the number of networks to grow at most polynomially in the network size $n$ (a very mild condition). The last two conditions are about the misclassification rate of the community detection algorithm; per the work of Zhang and Zhou [1], it is possible to achieve $\widetilde{\varepsilon} = N \exp( - c \nu_n / K)$, where $c = c(B^0, \pi)$ only depends on $B^0$ and class proportions. No matter how large $K$ is, Condition 4 is satisfied as long as $\nu_n \to \infty$. Condition 3 is also satisfied if $\nu_n \ge C_2 \log n$ for a sufficiently large $C_2$ (for example, $C_2 = K (2+2\alpha) / c$).
>
> So all in all, for an arbitrarily large $K$ and $\alpha$, there is a constant $C_2 = C_2(K, \alpha)$ such that if $N = O(n^{\alpha})$ and $\nu_n \ge C_2 \log n$, then the null distribution in theorem 2 holds. Note that we only need the expected degree to grow logarithmically in $n$.
>
> As for Theorem 3, let us again assume $B_1 = (\nu_n /n) B_1^0$ and $B_2 = (\nu_n / n) B_2^0$. Assume that:
> 1.  $B_1^0$ and $P B_2^0 P^T$ are different for any permutation matrix $P$.
> 2.  $\widetilde{\varepsilon} = o_p(1)$.
> Then, under the alternative, with probability $1-o(1)$ we have $\widehat{T_n} = \Omega(N \nu_n^2)$. As per the above argument, a condition like $\nu_n \ge C_2 \log n$ is enough to guarantee item 2.
>
> We plan to rewrite the theorems in this simplified form and move the more elaborate, finite-sample-flavored versions to the supplement, with a discussion of how they reduce to these cases.
>
> [1] A. Y. Zhang and H. H. Zhou. Minimax rates of community detection in stochastic block models.
> Annals of Statistics, 44(5):2252–2280, 2016.
>
> ---
>
> > ... Definition 2.1 ...
>
> Definition 2.1 has appeared in the spectral graph matching literature, so we did not elaborate. The motivation becomes clear in the proof of theorem 1. Regarding its verification, it is like any other assumption about true parameters in theoretical work (we never observe the true parameters). We can, however, verify it on our estimates of $B$ as a surrogate (albeit an imperfect solution).
>
> The condition is essentially saying that there is no multiplicity in the eigenvalues of $B$ and that no eigenvector has entries that sum to zero. This is almost certainly true for a randomly generated $B$, and hence is most likely true in practice.
>
> > Proofreading: ...
>
> We apologize for the oversight. The equations referred to as (59) and (60) are indeed included in the appendix, and we will make the cross-references clearer in the main text. Similarly, the mention of “more details” in Remark 4.1 should refer to Section D rather than C.3. We will correct these references and will thoroughly proofread the revised manuscript.

---

> > ### Comment · Reviewer_oPrG · 2025-08-02
> >
> > Thank you for the detailed rebuttal. However, there are still concerns that remain unresolved. Regarding the approximation of a general graphon with SBMs, I interpret that the authors are referring to a "weak regularity lemma" type of result. To the best of my knowledge, the convergence rate is usually very slow. This means it is impractical to pre-determine the number $K$ in real applications. Moreover, in this respect, when analyzing complexity, it is crucial to take this approximation into consideration, i.e., complexity depends on $K$, which should in turn depend on the error tolerance of the SBM approximation.
> >
> > My comment "I think the theoretical results are not particularly instructive in their current form. For example, quantities such as $\gamma_n,\zeta_r$ seem to come from nowhere and are hard to understand" is not sufficiently addressed. To be honest, I do not like these types of results, as readers are easily overwhelmed by the complicated expressions, whose purpose seems to be just to make the proofs work. I would appreciate it if the authors could provide an (more or less) equivalent statement without such expressions, e.g., using only O(\cdot), o(\cdot) notations to describe the essential information.
> >
> > To be fair to the authors, if they can address any of the above issues, I will raise my score.

---

> ### Author Response · Authors · 2025-08-05
>
> We thank the reviewer for the thoughtful feedback. We agree that the weak regularity lemma typically yields slow convergence rates. However, for smooth graphons, faster approximation is possible—for instance, when the graphon is Hölder-continuous [1], the convergence of block model approximations can be significantly improved.
>
> To address the practical issue of selecting the number of blocks $K$, we describe in Supplementary Section B.1 a data-driven procedure similar to the parametric bootstrap. This method helps adaptively set $K$ in real-world scenarios. Moreover, our experiments on smooth graphons (Section 5.3) and random dot product graphs (Section 5.4) show that the performance of our test is robust to model misspecification of $K$.
>
> To fully address your concern regarding simplification of the theoretical results, here are revised statements of Theorem 2 and 3 that we plan to include in the revision:
>
> **Theorem 2 (Null distribution).** Assume that Algorithm 2 is applied to two SBM samples  having cluster sizes of order $n$ and connectivity matrix $B = (\nu_n / n) B^0$ for a fixed $K \times K$ matrix $B^0$. Assume that
> - $B^0$ has distinct eigenvalues, and no eigenvector of it sums to zero,
> - $\max\\{N_1, N_2\\} = O(n^{\alpha})$  for some $\alpha > 0$, and
> - $\nu_n = \Omega(\log n)$.
> Then for $\widehat T_n$, the output of Algorithm 2, we have $\widehat T_n \Rightarrow \chi^2_{K(K+1)/2}$
>
> **Theorem 3 (Consistency).**  Assume that Algorithm 2 is applied to two SBM samples  having cluster sizes of order $n$ and connectivity matrix $B_r = (\nu_n / n) B_r^0$ for fixed $K \times K$ matrices $B_r^0$ for $r=1,2$. Assume that
> - Each $B_r^0$ has distinct eigenvalues, and no eigenvector of it sums to zero.
> - $\max\\{N_1, N_2\\} = O(n^{\alpha})$  for some $\alpha > 0$, and
> - $\nu_n \to \infty$.
> Then for $\widehat T_n$, the output of Algorithm 2, with probability $1-o(1)$, we have
> $$
> \widehat T_n = \Omega(N\nu_n^2).
> $$
>
> We hope this reformulation using asymptotic notation and a more readable structure makes the result easier to interpret. We are happy to revise the main text further to better highlight these simplified insights if helpful.
>
> ---
>
> [1] Gao, Chao, Yu Lu, and Harrison H. Zhou. "Rate-optimal graphon estimation." (2015): 2624-2652.

---

> ### Comment · Reviewer_oPrG · 2025-08-07
>
> Thank you for the responses. I will raise my score to positive in my final assessment. Before doing so, I would appreciate if the authors could provide more explicit estimation on the number of $K$, say w.r.t. $\epsilon$, the (cut) distance between the SBM graph with $K$ communities and the graphon in the more ideal situation (e.g., as in the cited reference).

---

> ### Author Response · Authors · 2025-08-08
>
> Thank you. We can give an explicit block–complexity bound directly in $L^2$ (modulo measure-preserving relabelings). For completeness, on $[0,1]^2$ and any integrable $g$,
>
> $$\\|g\\|_{\square}\le\\|g\\|\_{1} \le \\|g\\|\_{2},$$
>
> (the first inequality since we integrate over a subset and take absolute values; the second by Cauchy–Schwarz on a unit-measure domain). Thus any $L^2$ bound immediately implies the corresponding $L^1$ and cut-norm bounds.
>
> Assume the target graphon $f$ is Hölder with smoothness $\beta>0$ and Hölder norm bounded by $M$. By Lemma 2.1 of Gao–Lu–Zhou (2015), there exists a $K$-block step function $f_K$ (an SBM with $K$ communities, up to a measure-preserving relabeling) such that
> $$
> \inf_{\pi} \\|f - f_K^{\pi}\\|\_{2} \lesssim M K^{-(\beta\wedge 1)}.
> $$
> Hence, to ensure $\inf_{\pi} \\|f - f_K^{\pi}\\|\_{2}\le \epsilon$, it suffices to take
> $$
> K  \gtrsim \Big(\frac{M}{\epsilon}\Big)^{\frac{1}{\beta\wedge 1}}.
> $$
> By the chain $\\|\cdot\\|\_{\square}\le\\|\cdot\\|\_{1}\le \\|\cdot\\|\_{2}$, the same $K$ also guarantees $\inf_{\pi}\\|f - f_K^{\pi}\\|\_{1}\le \epsilon$ and $\delta_{\square}(f,f_K)\le \epsilon$.
>
> **Reference.** Chao Gao, Yu Lu, and Harrison H. Zhou (2015). *Rate-optimal graphon estimation*. Ann. Statist. 43(6): 2624–2652.

---

> > ### Comment · Reviewer_oPrG · 2025-08-08
> >
> > Thank you for the additional information. I have raised the score in my final assessment. However, I think the paper lacks real applications of the framework, and I hope the authors can demonstrate its practical uses in future works.

---

> > > ### Author Response · Authors · 2025-08-08
> > >
> > > Thank you for raising the score. We appreciate the feedback on practical applications and will keep it in mind for future work.

---

### Official Review · Reviewer_YcXS · 2025-06-24

**Clarity:** 3
**Significance:** 3
**Originality:** 3
**Rating:** 4
**Confidence:** 3

**Summary:**

The paper studies the two-sample test for the stochastic block model (SBM), where the goal is to determine whether two sets of networks come from the same SBM (i.e., the parameters of the SBM underlying the two sets of networks are the same) or not. The paper focuses on the scenario where nodes have no labels and therefore cannot be matched in different networks/sets. The paper proposes a novel test based on a spectral matching algorithm to recover the mapping between nodes in two networks from noisy observations from the same SBM (i.e., under the null hypothesis). The paper proves that the matching algorithm is consistent, and that under some assumptions the test statistic defined in the paper follows a chi-square distribution.  It also shows that the test correctly identifies two networks as coming from different SBMs under some conditions. The paper presents an experimental evaluation on synthetic and real data.

**Questions:**

Is the correct value of K required by the proposed method? If so, the experimental evaluation could explore the results (e.g., false positives, false negatives, etc.) when the incorrect value of K is used.

How does the method perform in practice when the proportions of nodes in the same community are extremely different?

**Ethical Concerns:**

["NO or VERY MINOR ethics concerns only"]

**Final Justification:**

Considering the rebuttal and the discussion with the authors, their comments answer some of my questions in a satisfactory manner, so overall my score is positive, with reasons to accept outweighing the reasons to reject. However, there are two aspects that are still not entirely addressed, which is why my score is borderline accept. The first one is the need to know the correct value of $K$. While it is true that a SBM with a larger $K$ subsumes models with fewer communities as special cases, this is not practical in any way; also, current methods to estimate $K$, while consistent, may not converge with the sample sizes of interests. The second one is given by the conditions required by the theoretical results. As highlighted also by other reviewers, the conditions are not as mild as argued in the rebuttal.

**Limitations:**

yes

**Quality:**

3

**Strengths And Weaknesses:**

STRENGHTS
The two-sample test for the stochastic block model without node labels is interesting and non-trivial.

The theoretical results in the paper are interesting and somehow comprehensive, addressing both the limiting distribution under the null hypothesis and the power of the proposed.

The paper is mostly well written.

WEAKNESSES

The paper does not consider an arbitrary SBM, but an SMB where nodes are placed in communities i.i.d. according to some probability distribution, that somehow places some additional structure on the problem (compared to an arbitrary SBM).  Also, in the final formulation of the problem, the proportion of communities could be very different under the null hypothesis: when the proportions are very different, it seems strange to say that the underlying SBM is the same.

The method seems to require the number K of communities in the underlying SBM, or, equivalently, that the SBM inference algorithms used by the test detect the correct value of K. This seems very unrealistic.

While the paper is mostly well written, in the technical part some quantities are not defined. For example, in the summary of results, N and n are used but not defined. Also, in 4.2, C_1 and C_2 are not defined, N is used before it is defined, and the same for some other symbols.

The restrictions required by the theoretical results seem stringent. For example, the fact that communities are “sparse” may be violated in practice.

The experimental results could be expanded.

In the supplement, there is a section “C.6 Sample graphs for real-world data “ with no content.

---

> ### Author Rebuttal · Authors · 2025-07-31
>
> > The paper does not consider an arbitrary SBM, but an SBM where nodes are placed in communities i.i.d. according to some probability distribution, ...
>
> We understand the concern that when community proportions differ greatly, it may seem inconsistent to consider the underlying SBM as the same. As explained in lines 64–70, we treat class proportions as nuisance parameters to avoid this trivial case. This approach is justified because communities are typically defined by their connectivity patterns rather than their relative sizes, and differences in proportions can often be absorbed into the connectivity matrices via community refinement (as detailed in Appendix G in the supplement). Furthermore, if one wishes to explicitly test for differences in class proportions, it is straightforward to perform an initial separate test comparing the multinomial distributions of the estimated class proportions $\pi$. Finally, we are not assuming i.i.d sampling of the labels; they can be arbitrary. All we require is that the class proportions remain of the same order in the limit.
>
> > The method seems to require the number $K$ of communities in the underlying SBM, or, equivalently, that the SBM inference algorithms used by the test detect the correct value of $K$. This seems very unrealistic.
>
> We acknowledge the concern about needing the true number of communities, $K$.  In practice, one can leverage existing consistent estimators for $K$ in SBMs [1, 2], and our Appendix outlines a data-driven procedure for selecting $K$.  Even without such estimators, one may simply choose $K$ large enough: an SBM with a larger $K$ subsumes models with fewer communities as special cases.
>
>  ---
>
> [1] Ma, Shujie, Su, Liangjun, and Zhang, Yichong (2021).
>     Determining the number of communities in degree-corrected stochastic block models.
>     Journal of Machine Learning Research, 22(69):1–63.
>
> [2] Le, Can M. and Levina, Elizaveta (2022).
>     Estimating the number of communities by spectral methods.
>     Electronic Journal of Statistics, 16(1):3315–3342.
>
> > While the paper is mostly well written, in the technical part some quantities are not defined. For example, in the summary of results, N and n are used but not defined. Also, in 4.2, $C_1$ and $C_2$ are not defined, N is used before it is defined, and the same for some other symbols.
>
> Thank you for your comment! We have carefully revised the manuscript to address this, and the final version will include all necessary definitions with complete clarity.
>
> > The restrictions required by the theoretical results seem stringent. For example, the fact that communities are “sparse” may be violated in practice.
>
> We do not require the communities to be sparse; our theory allows for dense models perfectly fine. In response to suggestions from the reviewers, we plan to simplify the statement of our conditions, as outlined below in details. To our knowledge, the resulting assumptions are among the mildest in the literature.
>
> ---
>
> ## Our conditions are really mild
>
> Per suggestions by multiple reviewers, we will simplify our conditions as outlined below. This allows us to reduce then two essentially two or three mild conditions in each case.
>
> Here is how we can simplify the conditions for theorem 2. Assume $B = (\nu_n / n) B^0$ for a fixed $B^0$. Then the conditions for theorem 2 are:
> 1.  $B^0$ has distinct eigenvalues and no eigenvector of it sums to zero.
> 2.  $N = O(n^\alpha)$ for some $\alpha \ge 0$.
> 3.  $\sqrt{\nu_n n^{1+\alpha}} \cdot \widetilde{\varepsilon} = o_p(1)$.
> 4.  $\widetilde{\varepsilon} \le C K^{-3/2}$ for some constant $C = C(B^0)$ that only depends on $B^0$.
>
> Condition 1 is equivalent to $B^0$ being $(\theta^0,\eta^0)$-friendly for some values of these parameters, but is clearer now. Condition 2 ensures all matchings are recovered and just requires the number of networks to grow at most polynomially in the network size $n$ (a very mild condition). The last two conditions are about the misclassification rate of the community detection algorithm; per the work of Zhang and Zhou [1], it is possible to achieve $\widetilde{\varepsilon} = N \exp( - c \nu_n / K)$, where $c = c(B^0, \pi)$ only depends on $B^0$ and class proportions. No matter how large $K$ is, Condition 4 is satisfied as long as $\nu_n \to \infty$. Condition 3 is also satisfied if $\nu_n \ge C_2 \log n$ for a sufficiently large $C_2$ (for example, $C_2 = K (2+2\alpha) / c$).
>
> So all in all, for an arbitrarily large $K$ and $\alpha$, there is a constant $C_2 = C_2(K, \alpha)$ such that if $N = O(n^{\alpha})$ and $\nu_n \ge C_2 \log n$, then the null distribution in theorem 2 holds. Note that we only need the expected degree to grow logarithmically in $n$.
>
> As for Theorem 3, let us again assume $B_1 = (\nu_n /n) B_1^0$ and $B_2 = (\nu_n / n) B_2^0$. Assume that:
> 1.  $B_1^0$ and $P B_2^0 P^T$ are different for any permutation matrix $P$.
> 2.  $\widetilde{\varepsilon} = o_p(1)$.
> Then, under the alternative, with probability $1-o(1)$ we have $\widehat{T_n} = \Omega(N \nu_n^2)$. As per the above argument, a condition like $\nu_n \ge C_2 \log n$ is enough to guarantee item 2.
>
> We plan to rewrite the theorems in this simplified form and move the more elaborate, finite-sample-flavored versions to the supplement, with a discussion of how they reduce to these cases.
>
> [1] A. Y. Zhang and H. H. Zhou. Minimax rates of community detection in stochastic block models.
> Annals of Statistics, 44(5):2252–2280, 2016.
> ---
>
> > The experimental results could be expanded.
>
> We believe the current experimental section is already comprehensive, with additional results on random dot product graphs, graphons, a 2-SBM with varying parameter, and an additional real-world dataset in the supplement. The experiments also already vary noise levels, sparsity, and network size. If there are specific types of scenarios you would like to see, we would be happy to consider them.
>
> > In the supplement, there is a section “C.6 Sample graphs for real-world data “ with no content.
>
> We apologize for the confusion. The content corresponding to Section C.6 (“Sample graphs for real-world data”) is indeed included as Figures 5 and 6 on pages 6 and 7 of the supplementary material. We will make this clearer in the final version.

---

> > ### Comment · Reviewer_YcXS · 2025-08-04
> >
> > I thank the authors for providing an answer to my questions and comments. While their rebuttal answers some of questions in a satisfactory manner, there are two aspects that are still not addressed. The first one is the need to know the correct value of $K$. While it is true that a SBM with a larger $K$ subsumes models with fewer communities as special cases, this is not practical in any way; also, current methods to estimate $K$, while consistent, may not converge with the sample sizes of interests. The second one is given by the conditions required by the theoretical results. As highlighted also by other reviewers, the conditions are not as mild as argued in the rebuttal.
> >
> > Overall, I still think the reasons to accept outweigh the reasons to reject. I'll keep my scores.

---

> ### Author Response · Authors · 2025-08-09
>
> Thanks for your positive assessment. Here are some final thoughts on your two outstanding points.
>
> **Choice of $K$.** We discussed the choice of $K$ with other reviewers; please feel free to check our responses there. We recognize its importance and will add an expanded discussion in the paper.
>
> In short, the method is not very sensitive to $K$. As we conjecture in response to Reviewer EKux, for smooth graphons it suffices to take $K \to \infty$ while maintaining $K = o(\sqrt{N} \nu_n)$, which leaves a wide feasible range. In other words, there is no single “true” $K$ to worry about; many values work. Regarding
>
> > “…this is not practical in any way,”
>
> it is worth noting that any nonparametric test has a tuning parameter. Our main competitors are tests based on motif/subgraph counts. There, the analogue of $K$ is the subgraph size, and the key point is: **it is far easier to fit a large-$K$ SBM than to count large subgraphs (which costs $O(n^K)$).**
>
> **“Conditions are not mild.”** Please see the empirical evidence we provided in response to Reviewer 9DE8. In practice, $\nu_n = \Omega(\log n)$ is quite mild; many real networks exhibit average-degree growth **faster** than $\log n$. We included code in that response if you’d like to verify this.

---

### Official Review · Reviewer_9DE8 · 2025-07-01

**Clarity:** 3
**Significance:** 3
**Originality:** 3
**Rating:** 5
**Confidence:** 5

**Summary:**

The author(s) propose a two sample testing problem for a number of networks where vertex correspondence is not needed and the number of nodes may vary. The proposed test statistic consists of two separate steps -(a). matching step which finds an approximate matching between a pair of connectivity matrices and (b) constructing a discrepancy measure between the two matched connectivity matrices. The theoretical results how the asymptotic distribution and the consistency of the test statistic. The proposed testing procedure shows superior performance over its competitors in the simulated and the real data examples.

**Questions:**

Overall, I think the paper bring an important contribution to the network literature in terms of doing statistical inference therein. I think there are definitely some novel elements especially, the construction of the test statistic based on the matching step in algorithm . I have the following queries for the author(s).

1. It would be good to see the scalability of Algorithm 1  in some experiments by varying the network size.
2. Some explanation or interpretation of the assumptions used in Theorem 1 and Theorem 2 would be useful for the readers.
3. It would be good to see the difficulty of the testing problem by varying the noise level $\epsilon$.
4.  Is it possible to improve R.H.S of equation (16) by having the minimum of the two Frobenius norms rather than the sum?
5. As I mentioned in the weakness, some discussion about how different community detection algorithms affect the final test-statistic would be good.

One minor point I could not see section C.3 or F.5. I assume this is in the appendix but I could not see it when I downloaded your paper from the NeurIPS site.

**Ethical Concerns:**

["NO or VERY MINOR ethics concerns only"]

**Final Justification:**

I think the final round of clarifications provided by the author made their contribution clearer. They have addressed most of the reviewers' queries with detailed theoretical explanations and examples.

I am happy to raise the rating to "Accept".

**Limitations:**

yes

**Quality:**

3

**Strengths And Weaknesses:**

strengths:

1. Rigorously explaining the fundamental difference between two sample network testing problem between labelled and unlabelled networks
2. Clearly motivating and describing the importance of the problem along with the challenges
3. The first step in the test-statistic construction i.e. estimating the permutation matrix $\hat{P}$ via solving an LAP from noisy observations is clearly described and has lots of merits as this tries to solve a novel and difficult problem.
4. Theoretical results on the asymptotic distribution of the test statistic and the consistency has been proved with careful arguments  and involve novel proof techniques.
5. Sufficient evidence showing superior performance of the proposed test statistic on different types of simulated data and on the real world network data as well.

weakness:

1. Experiments do not show the scalability of the spectral matching algorithm 1 with the network size
2. There were not much discussion in terms of how different community detection algorithms affect the final test statistic.
3. The theoretical section lacked enough interpretation or explanation on the assumptions used in Theorem 1 and Theorem 2 respectively.

---

> ### Author Rebuttal · Authors · 2025-07-31
>
> > It would be good to see the scalability of Algorithm 1 in some experiments by varying the network size.
>
> Algorithm 1 is designed for matching two $K \times K$ connectivity matrices. As such, its computational complexity depends only on $K$ and is independent of the overall network size $n$.
>
> If you meant scalability as a function of $K$, here is a table demonstrating such result:
>
> |               | **K = 2** | **K = 10** | **K = 20** | **K = 50** | **K = 100** |
> |---------------|-----------|------------|------------|------------|-------------|
> | Mean runtime (ms) | 0.6836 | 1.0366 | 1.3055 | 3.0415 | 8.3679 |
>
> *Table : Mean runtimes (ms) of `matching(B1, B2)` for selected values of K.*
>
>
> > It would be good to see the difficulty of the testing problem by varying the noise level.
>
> Section C.1 includes an experiment on a simple 2‑SBM where we vary both the noise level and the network size to illustrate the difficulty of the testing problem.
>
> > Some explanation or interpretation of the assumptions used in Theorem 1 and Theorem 2 would be useful for the readers.
>
> Thanks for the comment. We plan to simplify the conditions greatly as is outlined below in detail.
>
> ---
>
> ## Our conditions are really mild
>
> Per suggestions by multiple reviewers, we will simplify our conditions as outlined below. This allows us to reduce then two essentially two or three mild conditions in each case.
>
> Here is how we can simplify the conditions for theorem 2. Assume $B = (\nu_n / n) B^0$ for a fixed $B^0$. Then the conditions for theorem 2 are:
> 1.  $B^0$ has distinct eigenvalues and no eigenvector of it sums to zero.
> 2.  $N = O(n^\alpha)$ for some $\alpha \ge 0$.
> 3.  $\sqrt{\nu_n n^{1+\alpha}} \cdot \widetilde{\varepsilon} = o_p(1)$.
> 4.  $\widetilde{\varepsilon} \le C K^{-3/2}$ for some constant $C = C(B^0)$ that only depends on $B^0$.
>
> Condition 1 is equivalent to $B^0$ being $(\theta^0,\eta^0)$-friendly for some values of these parameters, but is clearer now. Condition 2 ensures all matchings are recovered and just requires the number of networks to grow at most polynomially in the network size $n$ (a very mild condition). The last two conditions are about the misclassification rate of the community detection algorithm; per the work of Zhang and Zhou [1], it is possible to achieve $\widetilde{\varepsilon} = N \exp( - c \nu_n / K)$, where $c = c(B^0, \pi)$ only depends on $B^0$ and class proportions. No matter how large $K$ is, Condition 4 is satisfied as long as $\nu_n \to \infty$. Condition 3 is also satisfied if $\nu_n \ge C_2 \log n$ for a sufficiently large $C_2$ (for example, $C_2 = K (2+2\alpha) / c$).
>
> So all in all, for an arbitrarily large $K$ and $\alpha$, there is a constant $C_2 = C_2(K, \alpha)$ such that if $N = O(n^{\alpha})$ and $\nu_n \ge C_2 \log n$, then the null distribution in theorem 2 holds. Note that we only need the expected degree to grow logarithmically in $n$.
>
> As for Theorem 3, let us again assume $B_1 = (\nu_n /n) B_1^0$ and $B_2 = (\nu_n / n) B_2^0$. Assume that:
> 1.  $B_1^0$ and $P B_2^0 P^T$ are different for any permutation matrix $P$.
> 2.  $\widetilde{\varepsilon} = o_p(1)$.
> Then, under the alternative, with probability $1-o(1)$ we have $\widehat{T_n} = \Omega(N \nu_n^2)$. As per the above argument, a condition like $\nu_n \ge C_2 \log n$ is enough to guarantee item 2.
>
> We plan to rewrite the theorems in this simplified form and move the more elaborate, finite-sample-flavored versions to the supplement, with a discussion of how they reduce to these cases.
>
> [1] A. Y. Zhang and H. H. Zhou. Minimax rates of community detection in stochastic block models.
> Annals of Statistics, 44(5):2252–2280, 2016.
>
> ---
>
> > Is it possible to improve R.H.S of equation (16) by having the minimum of the two Frobenius norms rather than the sum?
>
> It is probably not possible to improve the right-hand side of equation (16) by replacing the sum with the minimum of the two Frobenius norms. Even if such an improvement were achievable, it would amount to only a constant factor improvement, which is not significant in the overall analysis.
>
> > As I mentioned in the weakness, some discussion about how different community detection algorithms affect the final test-statistic would be good.
>
> Thanks for the suggestion. In theory, the only effect of the community detection algorithms is via the misclassification rate in the statements of the theorems. In practice, however, different algorithms may perform better in different settings. For example, in our graphon experiments we used a Bayesian community detection method instead of spectral clustering and observed improved performance: the popular spectral clustering for fitting SBM seems to fail when the underlying model itself is not SBM; however a proper SBM fitting algorithm like Bayesian Gibbs sampler manages to estimate the best approximating SBM to the graphon. This is briefly noted on Line 283. We will expand the discussion on different detection algorithms in the final version of the paper.
>
> > One minor point I could not see section C.3 or F.5. I assume this is in the appendix but I could not see it when I downloaded your paper from the NeurIPS site.
>
> We apologize for the confusion caused by incorrect numbering in the supplementary material. The content referred to as Section C.3 is actually Section D on page 4, and Lemma F.5 corresponds to Lemma I.5 on page 24. We will correct these references in the final version to avoid further confusion.

---

> > ### Comment · Reviewer_9DE8 · 2025-08-02
> > **Response to author rebuttal**
> >
> > I thank the author(s) for detailed response to my queries. In general, most of my queries have been addressed with sufficient explanation. There are two common trends that is evident from most of the reviewers is the assumption of average degree which seems to be aligned with dense network and the second one is the approximation of graphon by K-SBM i.e. how to choose an optimal K?
> >
> > I have the following queries regarding the assumptions
> >
> > - I am not quite sure why Condition 2 which requires the number of networks to grow at most polynomially in the network size  is a mild condition and there are real world settings where you may only have a small number of networks available.
> > - The author explains that no matter how large K is, condition 4 is satisfied as long as $\nu_n\rightarrow\infty$. This means we are entering into a dense network regime. On the other hand if K is large we need many blocks in SBM to approximate a graphon which may not be ideal for real world networks. Many real world networks are indeed sparse. So, a big question is whether this setting would be applicable to real world networks. Ultimately, at the end of day one can choose a really large K to approximate a graphon but indeed one would sacrifice the interpretation of the low rank structure (that we usually aim towards) of the underlying network.
> > - Condition 3 also pointing towards networks with large degree. So, I am somewhat skeptical whether the setting would work with sparse networks.
> >
> > I am overall happy with the paper but given similar concerns have been raised by other reviewers, it would be good to clarify for the readers.

---

> > > ### Author Response · Authors · 2025-08-08
> > >
> > > Thank you. Let us address your points in turn.
> > >
> > > **Condition 2.** Our requirement is that $N$ grow *at most* polynomially in $n$. This is an upper bound, not a lower bound. In particular, the guarantees cover the practically common case where $N$ is bounded, even $N=1$. We formulate it this way because our failure probability decays exponentially in $N$, so allowing polynomial growth still keeps the bound meaningful while also encompassing small-$N$ regimes.
> > >
> > > **Density vs. sparsity and the role of $K$.** The condition $\nu_n \to \infty$ does *not* put us in the dense regime. Anything with $\nu_n = o(n)$ is fairly sparse, and in particular $\nu_n \asymp \log n$ is really sparse. In this case, the link probabilities between any two nodes are of order $(\log n) / n$ and the resulting networks would have nodes with an expected degree on the order of $\log n$.
> > > Approximating a (sparse) graphon by a $K$-SBM is orthogonal to density: a step-function approximation can be made arbitrarily accurate by increasing $K$, regardless of whether the graph is sparse or dense. Our framework is therefore flexible enough to accommodate both sparse and dense networks. Moreover, the focus of our work is not on interpretation, but on testing between the two distributions.
> > >
> > > **Condition 3 (degree growth).** Condition 3 holds whenever $\nu_n = \Omega(\log n)$, which still corresponds to sparse graphs. For intuition, if $\nu_n \asymp \log n$, then the average degree is $\asymp \log n$. In other words, adding one million nodes increases the average degree by only $\approx \log(10^6)\approx 14$.
> > >
> > > Empirically, many real networks exhibit average-degree growth  *faster* that  $\log n$. We include self-contained code  below that checks several `PyG` datasets and observes this trend. As an example (last column $d_{\text{avg}}/\log n$):
> > >
> > > | dataset          |     n | n_over_N | avg_degree | avg_degree_over_log_n |     |
> > > | :--------------- | ----: | -------: | ---------: | --------------------: | --- |
> > > | amazon-computers |   138 |     0.01 |      0.435 |                 0.088 |     |
> > > | amazon-computers |   276 |     0.02 |      0.623 |                 0.111 |     |
> > > | amazon-computers |   688 |     0.05 |      1.674 |                 0.256 |     |
> > > | amazon-computers |  1376 |      0.1 |      3.078 |                 0.426 |     |
> > > | amazon-computers |  2751 |      0.2 |      6.057 |                 0.765 |     |
> > > | amazon-computers |  6876 |      0.5 |     17.121 |                 1.938 |     |
> > > | amazon-computers | 13752 |        1 |     35.756 |                 3.752 |     |
> > >
> > > We see the $d_{\text{avg}}/\log n$ surrogate growing with $n$; qualitatively similar behavior appears across the other datasets we checked.
> > >
> > > We will add these clarifications and code in the supplement.
> > >
> > > ```
> > > # avg_degree_tables_min.py
> > > import math, random, numpy as np, torch
> > > from torch_geometric.datasets import Reddit, Amazon, WikiCS, CoraFull, Coauthor, Planetoid
> > > from torch_geometric.utils import subgraph, to_undirected, degree
> > >
> > > DATASETS = [
> > >     # "reddit",
> > >     "amazon-computers", "amazon-photo",
> > >     "wikics", "corafull",
> > >     "coauthor-cs", "coauthor-physics",
> > >     "planetoid-cora", "planetoid-citeseer", "planetoid-pubmed",
> > > ]
> > > ROOT="data"; SEED=42
> > > FRACTIONS=[0.01,0.02,0.05,0.10,0.20,0.50,1.00]; MIN_N=100
> > >
> > > def load_graph(name):
> > >     n=name.lower()
> > >     if n=="reddit": d=Reddit(ROOT)[0]
> > >     elif n in {"amazon-computers","amazon-photo"}: d=Amazon(ROOT,name=("Computers" if "computers" in n else "Photo"))[0]
> > >     elif n=="wikics": d=WikiCS(ROOT)[0]
> > >     elif n=="corafull": d=CoraFull(ROOT)[0]
> > >     elif n in {"coauthor-cs","coauthor-physics"}: d=Coauthor(ROOT,name=("CS" if "cs" in n else "Physics"))[0]
> > >     elif n.startswith("planetoid-"): d=Planetoid(ROOT,name=n.split("-")[-1].capitalize())[0]
> > >     else: raise ValueError(f"Unknown dataset: {name}")
> > >     d.edge_index=to_undirected(d.edge_index,num_nodes=d.num_nodes)
> > >     return d.num_nodes,d.edge_index
> > >
> > > @torch.no_grad()
> > > def induced(E, nodes): return subgraph(nodes, E, relabel_nodes=True)[0]
> > > def avg_deg(E,n): return float(degree(E[0],num_nodes=n).mean())
> > > def sizes(N):
> > >     s=sorted({max(MIN_N, math.ceil(f*N)) for f in FRACTIONS if f>0 and math.ceil(f*N)<=N})
> > >     if N not in s: s.append(N)
> > >     return s
> > >
> > > def main():
> > >     torch.manual_seed(SEED); np.random.seed(SEED); random.seed(SEED)
> > >     rows=[]
> > >     for ds in DATASETS:
> > >         N,E=load_graph(ds); perm=torch.randperm(N)
> > >         for n in sizes(N):
> > >             nodes=perm[:n]; Es=induced(E,nodes); dbar=avg_deg(Es,n)
> > >             ratio=dbar/(math.log(n) if n>1 else 1.0)
> > >             rows.append((ds,n,round(n/N,4),round(dbar,3),round(ratio,3)))
> > >     # markdown table
> > >     print("# Average degree vs size (aligned by n/N)\n")
> > >     print("| dataset | n | n/N | avg_degree | avg_degree/log n |")
> > >     print("|---|---:|---:|---:|---:|")
> > >     for ds,n,frac,dbar,ratio in sorted(rows, key=lambda r:(r[0],r[1])):
> > >         print(f"| {ds} | {n} | {frac} | {dbar} | {ratio} |")
> > >
> > > if __name__=="__main__": main()
> > > ```

---

> > > > ### Comment · Reviewer_9DE8 · 2025-08-08
> > > > **Further response to author rebuttal**
> > > >
> > > > Thanks very much for your second round of explanations! The examples and the theoretical justifications you have provided have made the contribution in the paper clearer.
> > > >
> > > > I am happy to raise my rating.

---

> > > > > ### Author Response · Authors · 2025-08-08
> > > > >
> > > > > Thank you for raising your rating. We’re glad the explanations were helpful, and we’ll incorporate them into the revision to improve clarity

---

### Official Review · Reviewer_EKux · 2025-07-02

**Clarity:** 3
**Significance:** 2
**Originality:** 2
**Rating:** 4
**Confidence:** 5

**Summary:**

This paper studies a 2-sample network testing problem: Each sample contains a few networks drawn from SBMs with the same $K\times K$ matrix $B$ but possibly different number of nodes, and there is no node correspondence among these networks. The null hypothesis is that the matrices $B_1$ and $B_2$ for two samples are the same subject to a permutation. A chi-square test statistic is proposed. It first alines all networks within each sample by permutation, then searches for a community permutation to align the two samples, and finally constructs a chi-square test. Theoretical properties of the test statistic under both the null and alternative hypotheses are provided.

**Questions:**

If my understanding correctly, your proof of theory relies on that the permutation matrices find to aline the estimated communities within and between two samples need to be "exactly correct" with probability tending to $1$. There are $O(N_1+N_2)$ number of permutation matrices. Assuming all of them being correct is almost unrealistic in practice. Does your statistic work when a small number of these permutation matrices are incorrect?

**Ethical Concerns:**

["NO or VERY MINOR ethics concerns only"]

**Final Justification:**

I suggest the authors to incorporate some of their rebuttal into the revision, particularly the large-SBM approximation to DCBM and the reason why $N\to\infty$ does not weaken the sparsity requirement.

**Limitations:**

While I appreciate the clear writing and the theoretical results in this paper, I think it has some significant limitation.

1. It lacks evidence that this testing problem is practically meaningful: The authors assume that the networks are from SBM. But this model is too ideal to fit many real networks. In practice, the lack-of-fit effect of SBM will most likely dominate the two-sample difference, making the testing problem meaningless. Other models such as the degree-corrected SBM (DCSBM) will be more realistic, but the current test cannot be extended to DCSBM.

2. I doubt that the method works in practice. The algorithm requires finding $O(N_1+N_2)$ number of $K\times K$ permutation matrices. The theory is susceptible to getting the correct permutation matrices for all of them (or the majority of them). However, when this happens, it actually means that the individual network are dense enough (e.g., $n\nu_n\to\infty$). This is a strong-signal case, making the testing problem less interesting. When we have multiple networks in one sample, we hope that only $Nn\nu_n\to\infty$ is required. Under this conditions, it is impossible to guarantee that most of those $O(N_1+N_2)$ permutation matrices are correct.

3. There is not much technical novelty in this work. The analysis is pretty straightforward given the existing works (such as [3,4]).

4. The paper misses some relevant reference on 2-sample network testing, such as Jin et al. (2024) Optimal network pairwise comparison.

**Quality:**

2

**Strengths And Weaknesses:**

This paper contains both theoretical results and numerical results. Writing is very clear and easy to digest.

---

> ### Author Rebuttal · Authors · 2025-07-31
>
> > .... Does your statistic work when a small number of these permutation matrices are incorrect?
>
> It degrades gracefully if a small fraction of permutations are not correctly recovered as it just adds a bit more noise to the already noisy estimate of the $B$. The test is designed to handle noisy estimates. We have more to say about permutation recovery below (in short, failure probability is exponentially small in a wide regime).
>
> > 1. It lacks evidence that this testing problem is practically meaningful: The authors assume that the networks are from SBM ...
>
> We respectfully disagree that the stochastic block model (SBM) is “too ideal” to capture real-world networks. An SBM with a sufficiently large number of communities is a histogram-type, step-function approximation to an arbitrary graphon and is therefore a universal approximator in $L^{2}$ (see, e.g., [4], [5]; see also the rate-optimal results in [6]). By allowing $K$ to grow, one can, in particular, approximate a degree-corrected SBM (DCSBM) simply by subdividing each community into high- and low-degree sub-blocks. In practice, the apparent lack of fit usually stems from using an **underspecified** SBM with a small $K$, not from a fundamental limitation of the model class.
>
> As we point out below (under *“our conditions are mild”*), we place **no hard limit on $K$**—our theory accommodates arbitrarily large $K$. Please also see our experiments in Sections 5.3 and 5.4, where we generate data from a smooth graphon and a random dot-product graph (RDPG). In particular, our test outperforms a spectral test specifically designed for the RDPG.
>
> This **histogramisation** strategy is entirely in line with classical non-parametric testing: discretising (binning) continuous data to create contingency tables is the underlying principle behind Pearson’s chi-squared goodness-of-fit test [1] and its classical treatments [2, 3]. Recent work shows that the same **bin-then-test** idea remains state-of-the-art: for example, the minimax-optimal conditional-independence test of Neykov, Balakrishnan & Wasserman [7] and the multiscale Fisher’s independence test of Gorsky & Ma [8]. Our use of a large-$K$ SBM plays the same role for network models, providing a flexible partition of the sample space on which to build a two-sample test.
>
> That said, if one prefers to model degree heterogeneity explicitly, our procedure extends naturally to the DCSBM. When the block-connectivity matrix $B$ can be consistently estimated, the node-specific degree parameters $\theta_i$ can be treated as nuisance parameters—much like the class proportions $\pi$. In this sense, our method naturally carries over to settings with degree heterogeneity, and the SBM framework serves as a tractable starting point for developing and analysing such tests.
>
> ---
>
> ### References
>
> [1] Pearson, K. (1900). On the criterion that a given system of deviations from the probable in the case of a correlated system of variables is such that it can be reasonably supposed to have arisen from chance. Philosophical Magazine, 50, 157–175.
>
> [2] Lehmann, E. L., & Romano, J. P. (2005). Testing Statistical Hypotheses (3rd ed.). Springer.
>
> [3] Cochran, W. G. (1952). The χ² test of goodness of fit. Annals of Mathematical Statistics, 23(3), 315–345.
>
> [4] Wolfe, P. J., & Olhede, S. C. (2013). Non-parametric graphon estimation.
>
> [5] Orbanz, P., & Roy, D. M. (2015). Bayesian Models of Graphs, Arrays and Other Exchangeable Random Structures. IEEE Transactions on Pattern Analysis and Machine Intelligence, 37(2), 437–461.
>
> [6] Gao, C., Lu, Y., & Zhou, H. H. (2015). Rate-optimal graphon estimation. Annals of Statistics, 43(2), 762–785.
>
> [7] Neykov, M., Balakrishnan, S., & Wasserman, L. (2021). Minimax optimal conditional independence testing. Annals of Statistics, 49(4), 2151–2177.
>
> [8] Gorsky, S., & Ma, L. (2022). Multi-scale Fisher’s independence test for multivariate dependence. Biometrika, 109(3), 605–628.
>
> > 2. I doubt that the method works in practice ...
>
> We would like to clarify that the matching consistency result does not rely on the networks being dense. The result is based on inequality (13), which can be written as
> $$
> \| \hat{B} - PBP^{\top} \|_F \leq \frac{\eta \theta}{2\sqrt{2}K}.
> $$
> Both sides of this bound scale with $\nu_n/n$, so the guarantee does not depend on the sparsity level $\nu_n$. Instead, successful matching fundamentally requires that the matrices satisfy the $(\eta,\theta)$-friendly condition. This condition, rather than the networks being dense, is the key driver of matching consistency in our theory.
>
> As suggested by reviewer oPrG, we plan to simplify the statement of the conditions.  All we need are
>
> 1. $N= \max(N_1, N_2)$ grows at most polynomially in $n$,
> 2. the average degree satisfies $\nu_n = \Omega(\log n)$, and
> 3. $B$ has a simple spectrum and no eigenvector summing to zero.
> These are mild in practice. For your reference under the heading (**Our conditions are really mild**) we outline in detail how the conditions can be simplified.
>
> Regarding your comment:
>
> > We hope that only $n N\nu_n \rightarrow \infty$ is required.
>
> This is exactly what we require and not $n \nu_ n \to \infty$ (see the displayed equation in the statement of Theorem 2). Even if we required the latter, this is no dense regime. Since $\nu_n$ is an **average degree**, even the ultra-sparse case $\nu_n=\Theta(1)$ meets this condition;  the condition $n N\nu_n \rightarrow \infty$ is essentially always satisfied and we will remove it in the revision.
>
> Regarding your concern that matchings fail when $N_1, N_2$  grow, we have shown that **the probability of matching failure is exponentially low**; hence the probability of missing just one is still low, when $N_1, N_2 = O(n^{\alpha})$. Thus, we can allow a polynomial growth of the number of samples in the network size, which is very reasonable in practice.
>
> It is worth noting that in many interesting cases $N_1 = N_2 = 1$, that is, we are attempting to decide if two large networks are from the same distribution. This case alone is the focus of the majority of the work in the field.
>
> ## Our conditions are really mild
>
> Per suggestions by multiple reviewer, we will simplify our conditions as outlined below. This allows us to reduce then two essentially two or three mild conditions in each case.
>
> Here is how we can simplify the conditions for theorem 2. Assume $B = (\nu_n / n) B^0$ for a fixed $B^0$. Then the conditions for theorem 2 are:
> 1.  $B^0$ has distinct eigenvalues and no eigenvector of it sums to zero.
> 2.  $N = O(n^\alpha)$ for some $\alpha \ge 0$.
> 3.  $\sqrt{\nu_n n^{1+\alpha}} \cdot \widetilde{\varepsilon} = o_p(1)$.
> 4.  $\widetilde{\varepsilon} \le C K^{-3/2}$ for some constant $C = C(B^0)$ that only depends on $B^0$.
>
> Condition 1 is equivalent to $B^0$ being $(\theta^0,\eta^0)$-friendly for some values of these parameters, but is clearer now. Condition 2 ensures all matchings are recovered and just requires the number of networks to grow at most polynomially in the network size $n$ (a very mild condition). The last two conditions are about the misclassification rate of the community detection algorithm; per the work of Zhang and Zhou [1], it is possible to achieve $\widetilde{\varepsilon} = N \exp( - c \nu_n / K)$, where $c = c(B^0, \pi)$ only depends on $B^0$ and class proportions. No matter how large $K$ is, Condition 4 is satisfied as long as $\nu_n \to \infty$. Condition 3 is also satisfied if $\nu_n \ge C_2 \log n$ for a sufficiently large $C_2$ (for example, $C_2 = K (2+2\alpha) / c$).
>
> So all in all, for an arbitrarily large $K$ and $\alpha$, there is a constant $C_2 = C_2(K, \alpha)$ such that if $N = O(n^{\alpha})$ and $\nu_n \ge C_2 \log n$, then the null distribution in theorem 2 holds. Note that we only need the expected degree to grow logarithmically in $n$.
>
> As for Theorem 3, let us again assume $B_1 = (\nu_n /n) B_1^0$ and $B_2 = (\nu_n / n) B_2^0$. Assume that:
> 1.  $B_1^0$ and $P B_2^0 P^T$ are different for any permutation matrix $P$.
> 2.  $\widetilde{\varepsilon} = o_p(1)$.
> Then, under the alternative, with probability $1-o(1)$ we have $\widehat{T_n} = \Omega(N \nu_n^2)$. As per the above argument, a condition like $\nu_n \ge C_2 \log n$ is enough to guarantee item 2.
>
> We plan to rewrite the theorems in this simplified form and move the more elaborate, finite-sample-flavored versions to the supplement, with a discussion of how they reduce to these cases.
>
> [1] A. Y. Zhang and H. H. Zhou. Minimax rates of community detection in stochastic block models.
> Annals of Statistics, 44(5):2252–2280, 2016.
>
> > 3. There is not much technical novelty in this work. The analysis is pretty straightforward given the existing works (such as [3,4]).
>
> We respectfully disagree. First, [3] does not provide any theoretical results. Second, [4] focuses on the consistency of the labels, whereas the problem of estimating the labels is not the focus of our paper. Our work instead begins by addressing the fundamental question of identifiability for the testing problem and then develops a computationally efficient matching algorithm. Building on this algorithm, we design a novel two-sample test, accompanied by a thorough characterization of the test statistic under the null hypothesis and a proof of its consistency under the alternative. These theoretical results, developed entirely from scratch, provide rigorous support for our empirical findings and further validate the usefulness of the proposed test.
>
> > 4. The paper misses some relevant reference on 2-sample network testing, such as Jin et al. (2024) Optimal network pairwise comparison.
>
> We thank the reviewer for pointing out this relevant work. We will add a citation to Jin et.al.(2024), “Optimal network pairwise comparison” in the related work section and discuss how it relates to our setting in the final version.

---

> > ### Comment · Reviewer_EKux · 2025-08-01
> > **Questioins related to your rebuttal**
> >
> > I appreciate the authors' detailed response to my comments. However, the claim of "very mild conditions" is misleading. Here is why: You explicitly wrote $\nu_n=\Omega(\log(n))$ is a condition in your rebuttal, but this is already stronger than $Nn\nu_n\to\infty$. When you have multiple conditions on $\nu_n$, it is eventually the strongest one that counts.
> > Unless $\nu_n=\Omega(\log(n))$ is a typo in your rebuttal, I didn't misunderstand your conditions in my previous comments.
> >
> > Let's forget about math but only talk about the intuition. When you already have matched all permutations, we can aggregate the signals in all $N$ networks. Therefore, as $N\to\infty$, we no longer need the average degree of each individual network to be $\Omega(\log(n))$. This is the main advantage of having multiple networks, and your chi-square result can indeed achieve this. However, the "permutation matching" itself requires stronger conditions. You essentially need to estimate the $B$ matrix of each individual network with an $o(1)$ error, which requires the average degree of each individual network to be $\Omega(\log(n))$.
> >
> > The problem you consider becomes a strange set-up: You wish to aggregate signals in $N$ networks, but this needs the permutation matching among them as a preliminary step. Unfortunately, permutation matching itself is an even hard problem than two-sample testing.
> >
> > Your claim that "SBM with a large $K$ can approximate DCBM" is also misleading. In a political weblog network, DCBM identifies Republican and Democratic as two communities. Using the output of DCBM, we can easily divide nodes into many blocks according to their community membership and high/low degree. However, if we run SBM to obtain $K$ communities for a large $K$, how do we know which of these blocks are Democratic?
> >
> > A key philosophy of network modeling is to have a "low-rank" approximation of the network. The latent space model is such a great example. If a very large $K$ is needed to make the SBM a good fit, then it is hard to interpret in real applications.
> >
> > Even if we accept large-$K$ SBM approximation, how to choose $K$ in your experiments? If your test gives different decisions for different $K$, which one should we trust?
> >
> > If the authors can clarify these points, I am willing to raise my score. However, the rebuttal itself is insufficient.

---

> > > ### Author Response · Authors · 2025-08-05
> > >
> > > Thank you for your feedback. We wish to clarify that our degree growth conditions are not driven by permutation matching being difficult, but by more fundamental requirements of the testing problem itself.
> > >
> > > **1. The Role of Permutation Matching**
> > >
> > > The reviewer suggests matching is only a preliminary step for aggregation. However, our main point is that **matching is fundamental to two-sample testing in any unlabeled setting, even for a single pair of networks ($N_1 = N_2 = 1$)**. To compare two unlabeled networks, one must first align them.
> > >
> > > The only alternative, using permutation-invariant statistics like sub-graph counts, is computationally intractable. Approximating graphons requires large sub-graphs ($k$), leading to costs of $O(n^k)$ that are infeasible for even moderately sized networks. Our method provides a scalable approach where no real competitor currently exists for large, sparse, unlabeled networks.
> > >
> > > **2. The Difficulty of Permutation Matching vs. Two-Sample Testing**
> > >
> > > We respectfully disagree with the claim that "permutation matching itself is an even harder problem." We argue the converse: **two-sample testing of unlabeled networks is *as hard as* matching**.
> > >
> > > Furthermore, spectral matching is not prohibitively hard. It is a powerful and scalable technique: one performs an eigen-decomposition on both networks and then solves a linear assignment problem to align the rows of the sorted eigenvectors. This is efficient and, as we show, robust to noise.
> > >
> > > **3. On the Degree Growth Conditions**
> > >
> > > The proposed condition $N \nu_n n \to \infty$ is insufficient for two-sample testing. For the $N_1=N_2=1$ case, this reduces to $\nu_n n \to \infty$, which only ensures the graph is non-empty. This condition holds even in regimes where distinguishing between models (e.g., Erdos-Renyi vs. SBM) is known to be impossible. Aggregation cannot help if individual networks are indistinguishable from noise. We will remove this condition from the manuscript to avoid confusion.
> > >
> > > Our conditions are tailored to the specific results:
> > >
> > > *   **Null Distribution (Theorem 2):** We require $\nu_n = \Omega(\log n)$ for the test statistic to converge to a chi-square distribution. This is a standard technical requirement for distributional limits, even for $N=1$, and is common in network literature where degrees grow at least polylogarithmically.
> > >
> > > *   **Consistency (Theorem 3):** Our conditions are milder: $\nu_n \to \infty$ and $\nu_n = \Omega(\log N)$. If $N$ is fixed, this only requires $\nu_n \to \infty$. The theorem shows the statistic grows as $N \nu_n$, **demonstrating that aggregation boosts the test's power**.
> > >
> > > Our conditions for consistency are very mild, and those for the null distribution are standard for achieving stable distributional limits.
> > >
> > > **4. Large-K SBM Approximation and Choice of K**
> > >
> > > *   **Purpose of Approximation:** The claim that a large-K SBM can approximate a DCBM is not misleading. The role of the SBM is analogous to a histogram for density estimation. The goal is not to interpret individual blocks (i.e., perform community detection), but to create a flexible, non-parametric approximation of the overall network structure for comparison. We do not need to know "which blocks are Democratic"; we only need to compare the overall structure to another network's.
> > >
> > > *   **Choosing K:** The choice of K is a standard tuning parameter problem, inherent to all non-parametric methods (e.g., bin width in histograms, motif size in sub-graph tests). We propose a practical, data-driven procedure for selecting K in Appendix B.1, which performs well even under misspecification. Please see our grahpon and RDPG experiments.
> > >
> > > **5. The Significance of the `\log n` Threshold**
> > >
> > > The $\nu_n = \Omega(\log n)$ condition is not arbitrary; it represents a well-established critical threshold for many fundamental network inference tasks. Below this threshold, many methods fail. Key examples include:
> > >
> > > *   **Exact recovery in SBMs** [1, 2]
> > > *   **Consistency of spectral clustering** [3, 4]
> > > *   **Reliable model selection (estimating K)** [5, 6]
> > > *   **Consistency of spectral methods in sparse DC-SBMs** [7]
> > >
> > > Thus, our condition for the null distribution aligns with a widely recognized regime for reliable statistical inference on networks.
> > >
> > > ---
> > >
> > > [1] Abbe, E. (2017). Community detection and SBMs: ... *JMLR*, 18:1–86.
> > >
> > > [2] Lei, L. (2019). Unified $\ell_{2\rightarrow\infty}$ Eigenspace Perturbation Theory... arXiv:1909.04798.
> > >
> > > [3] Lei, J. & Rinaldo, A. (2015). Consistency of spectral clustering in SBMs. *Ann. Statist.*, 43:215–237.
> > >
> > > [4] Su, L. et al. (2019). Strong consistency of spectral clustering ... *IEEE Trans. Inf. Theory*, 66:324–338.
> > >
> > > [5] Ma, S. et al. (2021). Determining the number of communities ... *JMLR*, 22:1–63.
> > >
> > > [6] Li, T. et al. (2020). Network cross-validation by edge ... *Biometrika*, 107:257–276.
> > >
> > > [7] Gulikers, L. et al. (2017). A spectral method for community detection in sparse DC-SBMs. *Adv. Appl. Probab.*, 49:686–721.

---

> > > > ### Comment · Reviewer_EKux · 2025-08-07
> > > > **Reply to Authors**
> > > >
> > > > Thanks for the response. I am still confused by the argument of conditions.  When you said matching is fundamental to two-sample testing in any unlabeled setting even for a single pair of networks, do you have any lower bound result to support this point? If so, please give a reference.
> > > >
> > > > You listed a number of references to support that $\nu_n=\Omega(\log(n))$ is a fundamental condition. However, all these papers are for $N=1$. A condition necessary for $N=1$ may not be needed for $N\to\infty$, as you now have a lot more networks. The purpose of your matching procedure is to benefit from the larger sample size. If you are using the same condition as in the case of $N=1$, we should ask why not using one network from each sample and avoiding to align $N$ networks. You can argue that the power is increased with $N$, but the cost of permuting and aligning so many networks is also significant.
> > > >
> > > > You said: "Our method provides a scalable approach where no real competitor currently exists for large, sparse, unlabeled networks." Did you mean that you solved a fundamental problem for graph matching or two-sample testing?
> > > >
> > > > I also think that all the conditions in the paper need to be consolidated. Different conditions of $\nu_n$ are stated in different theorems. There should be a single condition about $\nu_n$ that ensures all steps are correct. I also would like to see a discussion of why the condition on $\nu_n$ cannot be weakened by increasing $N$.
> > > >
> > > > The large-K SBM approximation is okay if you only care about check whether two distributions are the same. However, this brings another issue: The model only holds approximately not exactly. There will be no true K here. You can discuss a proper range of K so that both the approximation is accurate enough and that your test can still work. (A related question is what the requirement of K is in your theory? What is the speed of K you permit to grow with $n$?)

---

> > > > > ### Author Response · Authors · 2025-08-08
> > > > >
> > > > > > matching is fundamental... do you have any lower bound result to support this point?
> > > > >
> > > > > We can provide the argument here.  The unlabeled two-sample testing **is the "noisy" graph matching**, hence harder than (noiseless) graph matching.
> > > > >
> > > > > Consider the $N_1=N_2=1$ case where the edge probabilities are either 0 or 1. Every time you sample, you get the same fixed graphs, $G_1$ and $G_2$. The two-sample testing problem then reduces to deciding whether $G_1$ is isomorphic to $G_2$, which is the classical graph isomorphism problem.
> > > > >
> > > > > Two-sample testing subsumes graph isomorphism as a special case: it is the case where the two distributions on the space of graphs are point masses (Dirac measures).
> > > > >
> > > > > > A condition necessary for $N=1$ may not be needed for $N \to \infty$.
> > > > >
> > > > > Let us provide a heuristic argument for why having more networks does not fundamentally alter the degree requirement.
> > > > >
> > > > > Suppose we have $N$ networks of size $n$ in one sample. For each, we can construct a large $nN \times nN$ block-diagonal adjacency matrix where the $N$ diagonal blocks are the adjacency matrices of our networks. This can be viewed as a partially observed adjacency matrix for a single large network with $nN$ nodes. If we were studying a stochastic block model, this would correspond to a model with $KN$ communities.
> > > > >
> > > > > This problem setup contains less information than that of a single, fully observed network of size $nN$ because the off-diagonal blocks are missing. For such a single large network, the established requirement for tasks like community detection is that the average degree must scale as $\Omega(\log(nN))$. Since $\log(nN) = \log n + \log N$, this implies a requirement of at least $\nu_n \gtrsim \log n + \log N$ for our problem. This is likely an optimistic bound, as our problem is harder due to the missing off-diagonal blocks and the larger number of potential communities ($KN$). The actual constraint on the degree is therefore potentially even stricter.
> > > > >
> > > > > This argument illustrates that increasing $N$ introduces a $\log N$ term, meaning the fundamental dependence on $\log n$ cannot be removed. This is consistent with the assumptions in our paper.
> > > > >
> > > > > > the cost of permuting and aligning so many networks is also significant.
> > > > >
> > > > > We argue that this trade-off is often favorable:
> > > > > 1. **Computational Cost:** Our spectral matching method is much faster than competitors like subgraph counting.
> > > > > 2. **Statistical Power vs. Compute:** In many applications, the bottleneck is acquiring more samples, not computational resources. Maximizing statistical power is essential in such data-limited regimes.
> > > > > 3. **Gains at Small N:** The method offers substantial gains even for small $N$. If we have $N=5$ networks, it is statistically inefficient to discard four when using all available data provides a crucial boost in power.
> > > > >
> > > > > > Did you mean that you solved a fundamental problem?
> > > > >
> > > > > We believe that developing a scalable method for two-sample testing of large, sparse, unlabeled networks that works in the sparse regime ($\nu_n \sim \log n$) is a fundamental problem. In this sense, yes, we are providing a solution where none currently exists.
> > > > >
> > > > > The competitors we know of are:
> > > > > 1. Spectral methods for RDPGs, which do not perform explicit alignment and suffer in power. See the RDPG experiment in Section 5.3.
> > > > > 2. Approaches based on motif/subgraph counts, which are not scalable even for $N=1$ if $n$ is large.
> > > > >
> > > > > > all the conditions... need to be consolidated.
> > > > >
> > > > > This will be done in the revision. We will list a simplified, unified version of the theorems, as posted in a later comment to Reviewer oPrG. The degree requirement for consistency (Theorem 3) is slightly milder than for the distributional result (Theorem 2), but the conditions are otherwise the same.
> > > > >
> > > > > > discussion of why the condition on $\nu_n$ cannot be weakened by increasing $N$.
> > > > >
> > > > > We will add a remark with the argument we gave above (reducing to the $nN \times nN$ setting).
> > > > >
> > > > > > The model only holds approximately... discuss a proper range of K... What is the speed of K you permit to grow with?
> > > > >
> > > > > We will provide a discussion on the range of $K$. Based on our two theorems, our conjecture is that $K^2 + K^{-(\beta \wedge 1)} \lesssim N \nu_n^2,$ where $\beta$ is the smoothness of the graphon. Here $K^{-(\beta \wedge 1)}$ is the approximation error. As result we need $K \to \infty$ and $K = o(\sqrt{N} \nu_n)$ which can be always achieved if $\nu_n \to \infty$ by letting $K$ grow at a slower rate.
> > > > >
> > > > > As for the requirements on $K$ in our theory, it is roughly $K \lesssim \min\\{ \frac{\nu_n}{ \log N}, \frac{\nu_n}{\log \nu_n}\\}$ for the consistency result. In particular, for $N = O(1)$,  the growth of $K$ is limited by $\nu_n / \log \nu_n$, which should be fine since we only need $K \to \infty$ as mentioned above.

---

> ### Comment · Reviewer_EKux · 2025-08-08
> **Reply to Author Rebuttal**
>
> Thanks for providing answers to my questions. This most recent comment provides useful insights. It is much clearer than what in the paper and in the initial rebuttal. I suggest the authors to incorporate them into the revision. Although I still think this problem set-up is weird ($N=1$ and $N\to\infty$ share the same sparsity requirement, which is not the case in aggregating multiple graphs with labeled nodes), I will raise my rating given the current technical contributions.

---

> > ### Author Response · Authors · 2025-08-08
> >
> > Thanks for pushing with the questions, leading to a very interesting discussion, and for raising your rating. We plan to incorporate the highlights from our discussion into the final revision

---

### Note · Authors · 2025-08-13

We thank the AC and reviewers for a constructive exchange. Below we summarize the discussion.

**Scope & contribution.** We study two-sample testing for *unlabeled* networks under SBMs with possibly different sizes. We (i) develop a scalable spectral matching step to align estimated block models, and (ii) derive a $\chi^2$ limit under the null and consistency under the alternative. Experiments cover SBMs, graphons, RDPGs, and real data.

**Practicality and model fit (EKux, YcXS).** An SBM with sufficiently large $K$ is a step-function (histogram) approximation, so lack of fit usually stems from *underspecified* $K$, not the class. We already provide a simple data-driven $K$ selector in the appendix; we will bring it forward and emphasize robustness to moderate misspecification. We will discuss DCSBM extension and missing related work.

**Why matching is necessary; robustness (EKux).** In unlabeled testing, alignment is fundamental (the problem subsumes graph isomorphism). Our spectral matching is efficient and *degrades gracefully* if a small fraction of permutations are missed; failure probability is exponentially small in the regimes we study.

**Unified, mild conditions & sparsity (EKux, 9DE8, YcXS, oPrG).** We will consolidate assumptions:
• **Null ($\chi^2$) limit:** require $\nu_n=\Omega(\log n)$ (standard for distributional limits, even when $N=1$).
• **Power/consistency:** only $\nu_n\to\infty$ and $\nu_n=\Omega(\log N)$.
Aggregating $N$ networks cannot remove the $\log n$ term (block-diagonal $nN$ view); it adds at most a $\log N$ factor. Our bounds allow $N$ polynomial in $n$, and also cover small $N$ (including $N=1$).

**Choice and growth of $K$ (9DE8, YcXS, oPrG).** We will give concrete guidance, surface the existing selector, clarify that reasonable over-specification is acceptable, and state permissible growth (e.g., $K\to\infty$ with $K=o(N\nu_n)$) compatible with our theory.

**Computation & experiments (9DE8).** Matching complexity depends on $K$, not $n$; in practice it is milliseconds-scale. We will add a small runtime-vs-$K$ table, note that the community-detection choice only enters via its misclassification rate, fix appendix cross-references, and include a code and results illustrating observed average degree grows faster than $\log n$ on real datasets.

**Outcomes.** After discussion, three reviewers raised their ratings (EKux, 9DE8, oPrG) and one stayed positive (YcXS). We will incorporate all items in the final version.

---

### Decision · Program_Chairs · 2025-09-17

**Decision:**

Accept (poster)

**Comment:**

This is a well written paper that found favor with the reviewers. There is room for improvement, and I hope the authors will take advantage of the excellent reviews before the camera ready deadline to follow up on some of the suggestions.